# Symmetric Mean-field Langevin Dynamics for Distributional Minimax Problems

**Juno Kim[1,2]***  **Kakei Yamamoto[3]**  **Kazusato Oko[1,2]**  **Zhuoran Yang[4]**  **Taiji Suzuki[1,2]**

[1]The University of Tokyo, Tokyo, Japan   [2]Center for Advanced Intelligence Project, RIKEN
[3]Massachusetts Institute of Technology, Cambridge, MA   [4]Yale University, New Haven, CT
*junokim@g.ecc.u-tokyo.ac.jp

## ABSTRACT

In this paper, we extend mean-field Langevin dynamics to minimax optimization over probability distributions for the first time with symmetric and provably convergent updates. We propose *mean-field Langevin averaged gradient* (MFL-AG), a single-loop algorithm that implements gradient descent ascent in the distribution spaces with a novel weighted averaging, and establish average-iterate convergence to the mixed Nash equilibrium. We also study both time and particle discretization regimes and prove a new uniform-in-time propagation of chaos result which accounts for the dependency of the particle interactions on all previous distributions. Furthermore, we propose *mean-field Langevin anchored best response* (MFL-ABR), a symmetric double-loop algorithm based on best response dynamics with linear last-iterate convergence. Finally, we study applications to zero-sum Markov games and conduct simulations demonstrating long-term optimality.

## 1 INTRODUCTION

The mean-field Langevin dynamics (MFLD) provides powerful theoretical tools to analyze optimization on the space of probability measures such as the training of two-layer neural networks (Mei et al., 2018; Chizat & Bach, 2018). The McKean-Vlasov stochastic process corresponds to the Wasserstein gradient flow minimizing an entropy-regularized convex functional, where the Gaussian noise encourages exploration and ensures global convergence (Hu et al., 2021; Chizat, 2022; Nitanda et al., 2022a). Langevin-based methods are especially attractive as they capture nonlinear aspects of learning as well as admit efficient particle discretizations. However, it remains unclear how to extend beyond single-objective optimization problems in a principled manner.

In this work, we develop MFLD for distributional minimax optimization problems. Denote by $\mathcal{P}_2(\mathcal{X}), \mathcal{P}_2(\mathcal{Y})$ the spaces of probability measures of finite variance on $\mathcal{X}, \mathcal{Y}$ with fixed base measures $\rho^\mu, \rho^\nu$. We consider the entropy-regularized saddle point problem for a convex-concave functional $\mathcal{L} : \mathcal{P}_2(\mathcal{X}) \times \mathcal{P}_2(\mathcal{Y}) \to \mathbb{R}$ with regularization strength or temperature $\lambda > 0$,[1]

$$\min_{\mu \in \mathcal{P}_2(\mathcal{X})} \max_{\nu \in \mathcal{P}_2(\mathcal{Y})} \mathcal{L}_\lambda(\mu, \nu), \quad \mathcal{L}_\lambda(\mu, \nu) := \mathcal{L}(\mu, \nu) + \lambda \operatorname{KL}(\mu \| \rho^\mu) - \lambda \operatorname{KL}(\nu \| \rho^\nu). \tag{1}$$

This formulation encompasses all objectives of the form $\mathcal{L}(\mu, \nu) = \iint Q(x, y) \mu(\mathrm{d}x) \nu(\mathrm{d}y)$ for generic nonconvex-nonconcave potentials $Q$. Such problems naturally arise for example in training generative adversarial networks (Goodfellow et al., 2020; Arjovsky et al., 2017; Hsieh et al., 2019), robust learning (Madry et al., 2018; Sinha et al., 2018) or solving zero-sum games in reinforcement learning (Daskalakis & Panageas, 2019; Domingo-Enrich et al., 2020; Zeng et al., 2022).

One is immediately led to consider *mean-field Langevin descent ascent* (MFL-DA) dynamics, the coupled distribution-dependent stochastic processes which seek to simultaneously minimize $\mathcal{L}$ over $\mu$ and maximize over $\nu$ (see Appendix A.2 for definitions of functional derivative and convexity):

$$\mathrm{d}X_t = \left( -\nabla_x \tfrac{\delta \mathcal{L}}{\delta \mu}(\mu_t, \nu_t)(X_t) + \lambda \nabla_x \log \rho^\mu(X_t) \right) \mathrm{d}t + \sqrt{2\lambda}\, \mathrm{d}W_t^\mu, \quad \mu_t = \operatorname{Law}(X_t),$$

$$\mathrm{d}Y_t = \left( \nabla_y \tfrac{\delta \mathcal{L}}{\delta \nu}(\mu_t, \nu_t)(Y_t) + \lambda \nabla_y \log \rho^\nu(Y_t) \right) \mathrm{d}t + \sqrt{2\lambda}\, \mathrm{d}W_t^\nu, \quad \nu_t = \operatorname{Law}(Y_t),$$

---

[1]Throughout the paper, sub/superscripts such as $\rho^\mu, \rho^\nu$ differentiate quantities related to the min and max variables, and do *not* indicate dependency on the distributions $\mu, \nu$. Our results are easily extended to different temperatures for each variable. We will also present many results for $\mu$ and omit the analogous statement for $\nu$.

where $W_t^\mu, W_t^\nu$ are independent Brownian motions. Descent ascent methods are more challenging to analyze compared to their single optimization counterparts; it is known that simultaneous updates may display cyclic or divergent behavior even for the simplest matrix games (Daskalakis & Panageas, 2019). For finite strategy spaces, a vigorous line of research has established convergence guarantees by employing optimistic or extragradient update rules; see Cen et al. (2023); Zeng et al. (2022) for an overview of recent literature and applications to Markov games.

Unfortunately, the convergence of MFL-DA is to the best of our knowledge still an open problem, and mean-field minimax dynamics remains largely unexplored. Existing results fail to establish convergence guarantees (Domingo-Enrich et al., 2020) or only give proofs for near-static flows where one strategy updates extremely or even infinitely quickly compared to the other (Ma & Ying, 2021; Lu, 2022). These works also impose the unrealistic assumption that $\mathcal{X}, \mathcal{Y}$ are both compact Riemannian manifolds without boundary. In contrast, we allow $\mathcal{X}, \mathcal{Y}$ to be Euclidean spaces.

Another fundamental consideration when implementing mean-field dynamics is to account for the errors arising from time discretization and particle approximation in a non-asymptotic manner, the latter referred to as *propagation of chaos* (Sznitman, 1991). Prior works generally give error bounds that blow up exponentially as training progresses (Mei et al., 2018; De Bortoli et al., 2020); uniform-in-time results were proven in the single optimization case only recently by Chen et al. (2022); Suzuki et al. (2023). Hence we are faced with the following research question:

> *Can we develop symmetric MFLD algorithms for distributional minimax problems with global convergence guarantees, and further provide uniform-in-time control over discretization errors?*

## 1.1 SUMMARY OF CONTRIBUTIONS

We address the above problem by proposing *mean-field Langevin averaged gradient*, a symmetric single-loop algorithm which takes inspiration from dual averaging methods and replaces the MFL-DA drift with the historical weighted average. We prove average-iterate convergence to the mixed Nash equilibrium. We also study both time and particle discretization and establish a new uniform-in-time propagation of chaos result. The analysis is greatly complicated by the dependence of the interactions on all previous distributions and the techniques developed are of independent interest.

In addition, we propose a symmetric double-loop algorithm, *mean-field Langevin anchored best response*, which realizes the best-response flow suggested in Lascu et al. (2023) via an inner loop running Langevin dynamics. We show that the outer loop updates enjoy last-iterate linear convergence to the mixed Nash equilibrium. Furthermore, we apply our theory to zero-sum Markov games and propose a two-step iterative scheme that finds the regularized Markov perfect equilibrium. Finally, we numerically demonstrate the superior optimality of both algorithms compared to MFL-DA.

## 2 PROBLEM SETTING AND ASSUMPTIONS

Denote by $\mathcal{P}_2(\mathbb{R}^d)$ the space of probability measures on $\mathbb{R}^d$ equipped with the Borel $\sigma$-algebra with finite second moment. Let $\mathcal{X} = \mathbb{R}^{d_\mathcal{X}}, \mathcal{Y} = \mathbb{R}^{d_\mathcal{Y}}$ and $\mathcal{L} : \mathcal{P}_2(\mathcal{X}) \times \mathcal{P}_2(\mathcal{Y}) \to \mathbb{R}$ be a weakly convex-concave functional. Our objective is to find the mixed Nash equilibrium (MNE) solving (1). Entropic regularization is frequently adopted in minimax optimization to account for imperfect information and ensure good convergence properties (McKelvey & Palfrey, 1995; Sokota et al., 2023).

We proceed to state our assumptions which are standard in the MFLD literature (Suzuki et al., 2023).

**Assumption 1** (Regularity of $\rho^\mu, \rho^\nu$). *We assume that $\rho^\mu = \exp(-U^\mu)$ and $\rho^\nu = \exp(-U^\nu)$ for $r_\mu$- and $r_\nu$-strongly convex potentials $U^\mu : \mathcal{X} \to \mathbb{R}$ and $U^\nu : \mathcal{Y} \to \mathbb{R}$, respectively. Furthermore, $\nabla_x U^\mu$ and $\nabla_y U^\nu$ are $R_\mu$- and $R_\nu$-Lipschitz, repsectively, and $\nabla_x U^\mu(0) = \nabla_y U^\nu(0) = 0$.*

**Assumption 2** (Regularity of $\mathcal{L}$ for MFL-AG). *We assume $\mathcal{L}$ is convex-concave and admits $C^1$ functional derivatives $\frac{\delta\mathcal{L}}{\delta\mu}, \frac{\delta\mathcal{L}}{\delta\nu}$ at any $(\mu,\nu)$, whose gradients are uniformly bounded, and Lipschitz continuous with respect to the input and $\mu, \nu$. That is, there exist constants $K_\mu, L_\mu, M_\mu > 0$ such that $\|\nabla_x \frac{\delta\mathcal{L}}{\delta\mu}(\mu,\nu)(x)\| \leq M_\mu$ and*

$$\left\| \nabla_x \frac{\delta\mathcal{L}}{\delta\mu}(\mu,\nu)(x) - \nabla_x \frac{\delta\mathcal{L}}{\delta\mu}(\mu',\nu')(x') \right\| \leq K_\mu \|x - x'\| + L_\mu(W_1(\mu,\mu') + W_1(\nu,\nu')) \quad (2)$$

*for all $x, x'$, $\mu$, and $\nu$. The same properties hold for $\nabla_y \frac{\delta\mathcal{L}}{\delta\nu}$ with $K_\nu, L_\nu, M_\nu > 0$.*

Assumption 2 implies in particular that $\frac{\delta\mathcal{L}}{\delta\mu}$ is $M_\mu$-Lipschitz and $\mu \mapsto \mathcal{L}(\mu, \nu)$ is $M_\mu$-Lipschitz in $W_1$. To present our results at full generality, we do *not* require boundedness of the functional derivatives $\frac{\delta\mathcal{L}}{\delta\mu}$ and $\frac{\delta\mathcal{L}}{\delta\nu}$, which would nevertheless simplify some arguments and improve for instance the log-Sobolev constants via the Holley-Stroock argument (Proposition A.4).

The KL regularization is enough to assure existence and uniqueness of the MNE via an application of the Kakutani fixed-point theorem (Conforti et al., 2020); see Appendix A.2 for the proof.

**Proposition 2.1** (Existence and uniqueness of MNE). *Under Assumptions 1 and 2, the solution* $(\mu^*, \nu^*)$ *to* (1) *uniquely exists and satisfies the first-order equations*

$$\mu^* \propto \rho^\mu \exp\left(-\frac{1}{\lambda}\frac{\delta\mathcal{L}}{\delta\mu}(\mu^*,\nu^*)\right), \quad \nu^* \propto \rho^\nu \exp\left(\frac{1}{\lambda}\frac{\delta\mathcal{L}}{\delta\nu}(\mu^*,\nu^*)\right). \tag{3}$$

The suboptimality of any given pair $(\mu, \nu)$ is quantified via the *Nikaidô-Isoda (NI) error* (Nikaidô & Isoda, 1955),

$$\mathrm{NI}(\mu,\nu) := \max_{\nu'\in\mathcal{P}_2(\mathcal{Y})} \mathcal{L}_\lambda(\mu,\nu') - \min_{\mu'\in\mathcal{P}_2(\mathcal{X})} \mathcal{L}_\lambda(\mu',\nu).$$

From the discussion in the proof of Proposition 2.1, it follows that $\mathrm{NI}(\mu, \nu) \geq 0$ and $\mathrm{NI}(\mu, \nu) = 0$ if and only if $\mu = \mu^*, \nu = \nu^*$. A pair $(\mu, \nu)$ satisfying $\mathrm{NI}(\mu, \nu) \leq \epsilon$ is called an $\epsilon$-MNE. As is usual in both discrete (Cen et al., 2021; Wei et al., 2021) and continuous (Lu, 2022; Lascu et al., 2023) minimax settings, our main goal is to prove convergence of the NI error along the proposed algorithms, which also implies convergence to the MNE in relative entropy (Lemma 3.5).

Cen et al. (2023) also point out that the MNE serves to approximate the MNE of the unregularized objective $\mathcal{L}$ as $\lambda \to 0$. However, $\mathcal{L}$ may not possess an MNE at all, e.g. for some bilinear objectives. Domingo-Enrich et al. (2020); Lu (2022) bypass this issue by assuming $\mathcal{X}, \mathcal{Y}$ are compact manifolds without boundary, in which case existence is guaranteed by Glicksberg's theorem. Alternatively, we may restrict the initialization and solution space to $\mathrm{KL}(\mu\|\rho^\mu) \leq R$, $\mathrm{KL}(\nu\|\rho^\nu) \leq R$ for some large radius $R$. Furthermore, if $\mathcal{L}$ does possess an MNE, it is possible to adopt the $\lambda_t = \Theta(1/\log t)$ cooling schedule studied in Lu (2022) for which our results can be modified to ensure $O(1/\log t)$ convergence to the *unregularized* MNE. Nonetheless, our focus is on the regularized problem $\mathcal{L}_\lambda$.

## 3 MEAN-FIELD LANGEVIN AVERAGED GRADIENT

### 3.1 PROPOSED METHOD

The main obstruction to proving convergence of MFL-DA is the complicated dependency of the proximal Gibbs distribution $\widehat{\mu}$ for $\mu$ on the opposing policy $\nu$. Motivated by dual averaging methods (Nesterov, 2009; Xiao, 2009; Nitanda et al., 2022b), our idea is simply to take the average of the drift over time so that the slowdown of the rolling average will ensure convergence of the KL gap.

We propose the *mean-field Langevin averaged gradient* (MFL-AG) flow with a weighting scheme $(\beta_t)_{t\geq 0}$ and temperature $\lambda > 0$ as the coupled pair of history-dependent McKean–Vlasov processes

$$\begin{aligned}
\mathrm{d}X_t &= -\left(\frac{1}{B_t}\int_0^t \beta_s \nabla_x \frac{\delta\mathcal{L}}{\delta\mu}(\mu_s,\nu_s)(X_t)\,\mathrm{d}s + \lambda\nabla_x U^\mu(X_t)\right)\mathrm{d}t + \sqrt{2\lambda}\,\mathrm{d}W_t^\mu, \\
\mathrm{d}Y_t &= \left(\frac{1}{B_t}\int_0^t \beta_s \nabla_y \frac{\delta\mathcal{L}}{\delta\nu}(\mu_s,\nu_s)(Y_t)\,\mathrm{d}s - \lambda\nabla_y U^\nu(Y_t)\right)\mathrm{d}t + \sqrt{2\lambda}\,\mathrm{d}W_t^\nu,
\end{aligned} \tag{4}$$

where $\mu_t = \mathrm{Law}(X_t)$, $\nu_t = \mathrm{Law}(Y_t)$ and $W_t^\mu$, $W_t^\nu$ are independent Brownian motions on $\mathcal{X}$ and $\mathcal{Y}$, respectively. The corresponding particle algorithm is studied in Section 3.3.

By *weighting scheme* we mean any integrable function $\beta = (\beta_t) : \mathbb{R}_{\geq 0} \to \mathbb{R}_{>0}$ where the normalizing weight $B_t = \int_0^t \beta_s\,\mathrm{d}s$ satisfies $B_t \to \infty$ and $\beta_t/B_t \to 0$ as $t \to \infty$. These conditions are roughly equivalent to $\widetilde{\Omega}(1/t) \leq \beta_t < \widetilde{O}(e^t)$ and ensure that the most recent update continues to influence the rolling average, but at an ever-decreasing rate. We will often substitute $\beta_t = t^r$ for a fixed exponent $r$ to obtain explicit convergence rates.

The dependence on previous distributions $(\mu_s, \nu_s)_{s\leq t}$ serves as a major point of departure from most existing works on mean-field dynamics. Nevertheless, existence and uniqueness of the flow (4) can be verified by extending the classical contraction argument of Sznitman (1991). The proof can be found in Appendix B.1.

**Proposition 3.1** (Well-definedness of MFL-AG flow). *Under Assumptions 1 and 2, the MFL-AG flow $(X_t, Y_t)$ (4) with continuous sample paths uniquely exists for all $t \in [0, \infty)$ for any initial distribution $\mu_0 \in \mathcal{P}_2(\mathcal{X}), \nu_0 \in \mathcal{P}_2(\mathcal{Y})$.*

The Fokker-Planck equations corresponding to the system (4) can be formulated as

$$\partial_t \mu_t = \nabla_x \cdot \left( \frac{\mu_t}{B_t} \int_0^t \beta_s \nabla_x \frac{\delta \mathcal{L}}{\delta \mu}(\mu_s, \nu_s) \, \mathrm{d}s + \lambda \mu_t \nabla_x U^\mu \right) + \lambda \Delta_x \mu_t = \lambda \nabla_x \cdot \left( \mu_t \nabla_x \log \frac{\mu_t}{\widehat{\mu}_t} \right),$$

$$\partial_t \nu_t = -\nabla_y \cdot \left( \frac{\nu_t}{B_t} \int_0^t \beta_s \nabla_y \frac{\delta \mathcal{L}}{\delta \nu}(\mu_s, \nu_s) \, \mathrm{d}s - \lambda \nu_t \nabla_y U^\nu \right) + \lambda \Delta_y \nu_t = \lambda \nabla_y \cdot \left( \nu_t \nabla_y \log \frac{\nu_t}{\widehat{\nu}_t} \right),$$

where $\widehat{\mu}_t, \widehat{\nu}_t$ are the MFL-AG proximal distributions given as

$$\widehat{\mu}_t \propto \rho^\mu \exp \left( -\frac{1}{\lambda B_t} \int_0^t \beta_s \frac{\delta \mathcal{L}}{\delta \mu}(\mu_s, \nu_s) \, \mathrm{d}s \right), \quad \widehat{\nu}_t \propto \rho^\nu \exp \left( \frac{1}{\lambda B_t} \int_0^t \beta_s \frac{\delta \mathcal{L}}{\delta \nu}(\mu_s, \nu_s) \, \mathrm{d}s \right) \quad (5)$$

which are well-defined due to the strong convexity of $U^\mu, U^\nu$ and Assumption 2.

MFL-AG is similar in spirit to fictitious play methods (Brown, 1951) in the two-player zero-sum game setting with $\beta_t \equiv 1$, where each player assumes their opponent has a stationary strategy and optimizes based on the average behavior of the opponent; the ideal fictitious play algorithm would perform the update $\mu_{t+1} = \widehat{\mu}_t$. If $\beta_t$ is increasing, the algorithm can be considered to undervalue older information which is more suitable for non-stationary environments. However, such methods require exact computation of the optimal response at every step which is generally unfeasible. In contrast, the MFL-AG policies continuously flow towards their response policies at any given time.

As usual, $\widehat{\mu}_t, \widehat{\nu}_t$ satisfy a log-Sobolev inequality which is crucial to controlling the mean-field flows. The mild dependency $\alpha_\mu = \Omega(1/d_\mathcal{X})$ is the only manifestation of dimensional dependence in our results, and can be avoided in cases where the Holley-Strook argument applies. See Appendix A.1 for details.

**Proposition 3.2.** *Let the probability measure $\mu \propto \rho^\mu \exp(-\lambda^{-1} h) \in \mathcal{P}_2(\mathcal{X})$ with $\|h\|_{\mathrm{Lip}} \leq M_\mu$. Then under Assumption 1, $\mu$ satisfies the log-Sobolev and Talagrand's inequalities with constant*

$$\alpha_\mu \geq \frac{r_\mu}{2} e^{-\frac{4M_\mu^2}{r_\mu \lambda^2} \sqrt{\frac{2d_\mathcal{X}}{\pi}}} \vee \left( \frac{4}{r_\mu} + \left( \frac{M_\mu}{r_\mu \lambda} + \sqrt{\frac{2}{r_\mu}} \right)^2 \left( 2 + \frac{d_\mathcal{X}}{2} \log \frac{e^2 R_\mu}{r_\mu} + \frac{4M_\mu^2}{r_\mu \lambda^2} \right) e^{\frac{M_\mu^2}{2r_\mu \lambda^2}} \right)^{-1}.$$

## 3.2 CONTINUOUS-TIME CONVERGENCE

We begin by studying the properties of the flow (4). At each time $t$ the policies evolve towards the proximal distributions, and the deceleration of the rolling average allows the flow to catch up with $\widehat{\mu}_t, \widehat{\nu}_t$; this observation plays a key part in further analyses. Note that we state many results for only the min policy $\mu$ and omit the analogous statement for $\nu$.

**Proposition 3.3** (Proximal convergence of MFL-AG flow). *Under Assumptions 1 and 2, for the weighting scheme $\beta_t = t^r$ with a fixed exponent $r > -1$ the proximal KL gap is bounded as*

$$\mathrm{KL}(\mu_t \| \widehat{\mu}_t) \leq \frac{2(r+1)^2 M_\mu^2}{\alpha_\mu^3 \lambda^4 t^2} + O(t^{-3}).$$

See Appendix B.2 for the proof. It is then clear that if MFL-AG converges, it must converge to the MNE (3) by setting $\mu_\infty = \widehat{\mu}_\infty, \nu_\infty = \widehat{\nu}_\infty$.

For ordinary MFLD, KL gap convergence of the above type is generally enough to show absolute convergence through entropy sandwich inequalities, see e.g. Nitanda et al. (2022a); Lu (2022). In our case, however, the relative entropy no longer quantifies the optimality gap at $(\mu_t, \nu_t)$ since the proximal distributions are no longer 'state functions' and depend on the entire history in (5). Nevertheless, we are able to obtain our first main result, average-iterate convergence of MFL-AG. Our approach, detailed in Appendix B.3, extends conjugate function arguments from dual averaging to the minimax setting and also leverages the preceding $O(1/t^2)$ KL gap convergence.

**Theorem 3.4** (Average-iterate convergence of MFL-AG flow). *Denote the weighted average of the MFL-AG distributions up to time $t$ as $\bar{\mu}_t = \frac{1}{B_t} \int_0^t \beta_s \mu_s \, \mathrm{d}s, \bar{\nu}_t = \frac{1}{B_t} \int_0^t \beta_s \nu_s \, \mathrm{d}s$. Then under*

---

**Algorithm 1** Mean-field Langevin Averaged Gradient

---

**Require:** temperature $\lambda$, max epochs $K$, learning rate $\eta$, number of particles $N$, exponent $r$
**Initialization:** $\overline{\mathscr{X}}_K, \overline{\mathscr{Y}}_K \leftarrow \varnothing, \mathscr{X}_1, \mathscr{Y}_1$
    **for** $k = 1, \cdots, K - 1$ **do**
        For all particles $i = 1, \cdots, N$ sample $\xi_k^{\mu,i} \sim \mathcal{N}(0, \mathrm{I}_{d_{\mathcal{X}}}), \xi_k^{\nu,i} \sim \mathcal{N}(0, \mathrm{I}_{d_{\mathcal{Y}}})$ and update
        $X_{k+1}^i \leftarrow X_k^i - \frac{\eta}{B_k} \sum_{j=1}^k \beta_j \nabla_x \frac{\delta\mathcal{L}}{\delta\mu}(\mu_{\mathscr{X}_j}, \nu_{\mathscr{Y}_j})(X_k^i) - \lambda\eta\nabla_x U^\mu(X_k^i) + \sqrt{2\lambda\eta}\xi_k^{\mu,i}$
        $Y_{k+1}^i \leftarrow Y_k^i + \frac{\eta}{B_k} \sum_{j=1}^k \beta_j \nabla_y \frac{\delta\mathcal{L}}{\delta\nu}(\mu_{\mathscr{X}_j}, \nu_{\mathscr{Y}_j})(Y_k^i) - \lambda\eta\nabla_y U^\nu(Y_k^i) + \sqrt{2\lambda\eta}\xi_k^{\nu,i}$
    **end for**
    **for** $k = 1, \cdots, K$ **do**
        Sample $\lfloor \beta_k N / B_K \rceil$ particles from $\mathscr{X}_k, \mathscr{Y}_k$ and concatenate with $\overline{\mathscr{X}}_K, \overline{\mathscr{Y}}_K$, resp.
    **end for**
    **return** $\overline{\mathscr{X}}_K, \overline{\mathscr{Y}}_K$

---

*Assumptions 1 and 2, for the weighting scheme $\beta_t = t^r$ with fixed exponent $r > 0$, the NI error of the averaged pair $\bar{\mu}_t, \bar{\nu}_t$ converges with rate*

$$\mathrm{NI}(\bar{\mu}_t, \bar{\nu}_t) \leq \left( \frac{M_\mu^2}{\alpha_\mu^2} + \frac{M_\nu^2}{\alpha_\nu^2} \right) \frac{4(r+1)^2}{r\lambda^2 t} + O(t^{-2}),$$

*and the leading term is optimized when $\beta_t = t$. For the unweighted averaging scheme $\beta_t \equiv 1$,*

$$\mathrm{NI}(\bar{\mu}_t, \bar{\nu}_t) \leq \left( \frac{M_\mu^2}{\alpha_\mu^2} + \frac{M_\nu^2}{\alpha_\nu^2} \right) \frac{4 \log t}{\lambda^2 t} + O(t^{-1}).$$

In light of Lemma 3.5 (proved in Appendix A.2), Theorem 3.4 immediately implies convergence with the same rate of $(\bar{\mu}_t, \bar{\nu}_t)$ in relative entropy to the MNE.

**Lemma 3.5** (Entropy sandwich lower bound). *For any $\mu \in \mathcal{P}_2(\mathcal{X})$ and $\nu \in \mathcal{P}_2(\mathcal{Y})$ it holds that*

$$\mathrm{KL}(\mu\|\mu^*) + \mathrm{KL}(\nu\|\nu^*) \leq \lambda^{-1} \mathrm{NI}(\mu, \nu).$$

The weighting exponent $r$ can be thought of as a hyperparameter controlling the following trade-off. A larger $r$ tends to give more weight to recent information, which leads to a faster-moving average and slower convergence of the proximal gap (Proposition 3.3). However, it also allows for faster convergence of the weighted average to the MNE. The rate is optimized when $r = 1$, which is in agreement with works such as Tao et al. (2021) on dual averaging and Guo et al. (2020) on stochastic gradient descent which incorporate averaging with increasing weights $\beta_t \propto t$ to obtain improved rates ($\sim 1/t$) compared to the unweighted averages ($\sim \log t/t$).

## 3.3 TIME AND SPACE DISCRETIZATION

We now summarize our discretization analysis of MFL-AG developed throughout Appendix C. Our study incorporates both a discrete time step $\eta$ for the Langevin flow and particle approximations for the laws $\mu, \nu$. Denote ordered sets of $N$ particles by $\mathscr{X} = (X^i)_{i=1}^N \in \mathcal{X}^N$, $\mathscr{Y} = (Y^i)_{i=1}^N \in \mathcal{Y}^N$ and the corresponding empirical distributions by $\mu_{\mathscr{X}} = \frac{1}{N} \sum_{i=1}^N \delta_{X^i}, \nu_{\mathscr{Y}} = \frac{1}{N} \sum_{i=1}^N \delta_{Y^i}$. The update $\mathscr{X}_{k+1}, \mathscr{Y}_{k+1}$ will depend on the full history $(\mathscr{X}_{1:k}, \mathscr{Y}_{1:k})$, where $\mathscr{X}_1$ and $\mathscr{Y}_1$ are sampled independently from initial distributions $\mu^\circ \in \mathcal{P}_2(\mathcal{X})$ and $\nu^\circ \in \mathcal{P}_2(\mathcal{Y})$, respectively.

In order to implement gradient averaging, the integral in (4) must be replaced by the discrete-time average with respect to a sequence of weights $(\beta_k)_{k\in\mathbb{N}}$; the cumulative weights are denoted as $B_k = \sum_{j=1}^k \beta_j$. Moreover, the final average of $\mu_{\mathscr{X}_1}, \cdots, \mu_{\mathscr{X}_K}$ may be computed by randomly sampling $\beta_k N/B_K$ particles from each set $\mathscr{X}_k$ and concatenating. See Algorithm 1 for details.

The propagation of chaos framework recently developed in Chen et al. (2022); Suzuki et al. (2023) relies on a lifted proximal distribution $\widehat{\mu}^{(N)}$ on the configuration space $\mathcal{X}^N$. By integrating out the conditioning on the previous step in the continuity equation, this is used to elegantly control the evolution of the joint distribution $\mu^{(N)}$ of the $N$ particles. In our case, however, the dependency on the full history $(\mathscr{X}_{1:k}, \mathscr{Y}_{1:k})$ cannot be integrated out consistently and must be retained:

$$\widehat{\mu}_k^{(N)}(\mathscr{X}) \propto \rho^{\mu\otimes N}(\mathscr{X}) \exp\left( -\frac{N}{\lambda B_k} \int_{\mathcal{X}} \sum_{j=1}^k \beta_j \frac{\delta\mathcal{L}}{\delta\mu}(\mu_{\mathscr{X}_j}, \nu_{\mathscr{Y}_j})\mu_{\mathscr{X}}(\mathrm{d}x) \right).$$

This renders the KL gap argument with $\mu^{(N)}$ inaccessible and we must work step-by-step with the atomic measures $\mu_{\mathscr{X}_k}, \nu_{\mathscr{Y}_k}$, which further complicates matters as we cannot directly utilize metrics involving $\mu_{\mathscr{X}_k}$ in order to avoid the curse of dimensionality. Instead, we prove and exploit the following uniform law of large numbers (Appendix C.3).

**Proposition 3.6.** *Let $F : \mathcal{P}_2(\mathcal{X}) \times \mathcal{P}_2(\mathcal{Y}) \times \mathcal{X} \to \mathbb{R}$, $(\mu, \nu, x) \mapsto F(\mu, \nu)(x)$ be a functional such that $F(\mu, \nu)$ is $M_\mu$-Lipschitz on $\mathcal{X}$ and further satisfies*

$$\|F(\mu, \nu) - F(\mu', \nu')\|_{\mathrm{Lip}} \leq L_\mu(W_1(\mu, \mu') + W_1(\nu, \nu')).$$

*If $\eta \leq \bar{\eta} := \frac{r_\mu \lambda}{2(L_\mu + \lambda R_\mu)^2} \wedge \frac{r_\mu}{4\lambda R_\mu^2} \wedge \frac{r_\nu \lambda}{2(L_\nu + \lambda R_\nu)^2} \wedge \frac{r_\nu}{4\lambda R_\nu^2}$ and the weight sequence $\beta_k = k^r$ for $r \geq 0$, then for all integers $k, N$ it holds that*

$$\mathbb{E}_{\mathscr{X}_{1:k}, \mathscr{Y}_{1:k}} \left[ \int_{\mathcal{X}} F(\mu_{\mathscr{X}_k}, \nu_{\mathscr{Y}_k})(\mu_{\mathscr{X}_k} - \Pi \widehat{\mu}_{k-1}^{(N)})(\mathrm{d}x) \right] \leq \frac{r+1}{k} C_1(\eta) + C_2\sqrt{\eta} + \frac{C_3}{\sqrt{N}}. \quad (6)$$

*The same bound also holds for the max policy $\nu$. The constants $C_2, C_3$ only depend on problem quantities (including the LSI constants) with at most polynomial order, while the function $C_1$ depends on problem quantities and $\eta$.*

Here, $\Pi$ denotes the average of the $N$ pushforward operators along the coordinate projection maps $\mathscr{X} \mapsto X^i$. The main idea of the proof is to *look backwards in time*: close enough so that the dynamics is nearly particle-independent due to the slowdown of the averaged gradient, but far enough to assure exponential convergence to the approximate stationary distribution. Furthermore, the $W_1$-Lipschitz leave-one-out argument in Step 3 shows that the $O(1/\sqrt{N})$ rate is optimal.

We finally present our main discretization error bound; the proof is presented in Appendix C.5.

**Theorem 3.7** (Convergence of discretized MFL-AG). *Denote the averaged empirical distributions as $\mu_{\overline{\mathscr{X}}_k} = \frac{1}{B_k} \sum_{j=1}^{k} \beta_j \mu_{\mathscr{X}_j}$, $\nu_{\overline{\mathscr{Y}}_k} = \frac{1}{B_k} \sum_{j=1}^{k} \beta_j \nu_{\mathscr{Y}_j}$. If $\eta \leq \bar{\eta}$ and $\beta_k = k^r$ with $r > 0$, the MFL-AG discrete update satisfies for all $K, N$,*

$$W_1^2(\mathbb{E}[\mu_{\overline{\mathscr{X}}_K}], \mu^*) + W_1^2(\mathbb{E}[\nu_{\overline{\mathscr{Y}}_K}], \nu^*) \leq \frac{(r+1)^2}{rK} \widetilde{C}_1(\eta) + \widetilde{C}_2\sqrt{\eta} + \frac{\widetilde{C}_3}{\sqrt{N}}$$

*with similar constants as in Proposition 3.6. If $r = 0$, the first term is replaced by $O(\log K / K)$.*

Hence the errors arising from time and particle discretization are separately bounded as $O(\sqrt{\eta})$ and $O(1/\sqrt{N})$. An unfortunate byproduct of the perturbation analysis is a roughly $\eta^{-1/\alpha_\mu}$ order dependency in the constant $C_1(\eta)$; nonetheless, the convergence in time is $O(1/K)$ for any fixed $\eta$. In particular, for any specified error $\epsilon > 0$ we can take $\eta = O(\epsilon^2)$ and $N = O(\epsilon^{-2})$ as well as $K = O(\epsilon^{-1/\alpha_\mu \wedge \alpha_\nu})$ so that $W_1^2(\mathbb{E}[\mu_{\overline{\mathscr{X}}_K}], \mu^*) + W_1^2(\mathbb{E}[\nu_{\overline{\mathscr{Y}}_K}], \nu^*) < \epsilon$.

We remark that the squared Wasserstein distance is a natural measure of optimality consistent with the continuous-time rate obtained in Theorem 3.4 in view of Lemma 3.5. Note that Theorem 3.7 quantifies the bias of the MFL-AG outputs, but does not tell us anything about the variance. In Appendix C.6, we give a bound for the expected distance $\mathbb{E}[W_1(\mu_{\overline{\mathscr{X}}_k}, \mu^*) + W_1(\nu_{\overline{\mathscr{Y}}_k}, \nu^*)]$ and also discuss why the curse of dimensionality is unavoidable in this setting.

## 4 MEAN-FIELD LANGEVIN ANCHORED BEST RESPONSE

### 4.1 PROPOSED METHOD

Our second proposal builds upon the *mean-field best response* (MF-BR) flow recently proposed in Lascu et al. (2023). There, the authors prove that the strategies $(\mu_t, \nu_t)_{t \geq 0}$ given by the linear flow

$$\mathrm{d}\mu_t(x) = \beta(\widehat{\mu}_t(x) - \mu_t(x)) \, \mathrm{d}t, \quad \mathrm{d}\nu_t(x) = \beta(\widehat{\nu}_t(x) - \nu_t(x)) \, \mathrm{d}t,$$

with speed $\beta > 0$ converge exponentially to the unique MNE, where $\widehat{\mu}_t \propto \rho^\mu \exp\left(-\frac{1}{\lambda} \frac{\delta \mathcal{L}}{\delta \mu}(\mu_t, \nu_t)\right)$, $\widehat{\nu}_t \propto \rho^\nu \exp\left(\frac{1}{\lambda} \frac{\delta \mathcal{L}}{\delta \nu}(\nu_t, \nu_t)\right)$ are the best response proximal distributions, so called because they are the optimal responses against the current policies of all players (rather than the historical average

in MFL-AG). However, a major weakness of MF-BR is that the flow is not directly realizable by a particle algorithm.

We therefore propose the *mean-field Langevin anchored best response* (MFL-ABR) process by incorporating an inner loop running Langevin dynamics, decoupled by anchoring the gradient at the output $(\mu_k, \nu_k)$ of the previous outer loop:

$$X_0^\dagger \sim \rho^\mu, \quad \mathrm{d}X_t^\dagger = -\left(\nabla_x \frac{\delta \mathcal{L}}{\delta \mu}(\mu_k, \nu_k)(X_t^\dagger) + \lambda \nabla_x U^\mu(X_t^\dagger)\right) \mathrm{d}t + \sqrt{2\lambda}\, \mathrm{d}W_t^\mu, \quad 0 \le t \le \tau,$$

and similarly for $Y_t^\dagger$. The outputs at time $\tau$, denoted by $\mu_{k,\tau}^\dagger = \mathrm{Law}(X_\tau^\dagger), \nu_{k,\tau}^\dagger = \mathrm{Law}(Y_\tau^\dagger)$ serve as approximations of the best response proximal distributions (replacing time $t$ with the discrete index $k$). The outer loop then performs the discretized MF-BR update,

$$\mu_{k+1} = (1-\beta)\mu_k + \beta\mu_{k,\tau}^\dagger, \quad \nu_{k+1} = (1-\beta)\nu_k + \beta\nu_{k,\tau}^\dagger,$$

where $\mu_0 = \rho^\mu, \nu_0 = \rho^\nu$. The flow can be immediately realized by a simple particle algorithm; see Algorithm 2 in the appendix. A similar method for single convex optimization was also recently implemented in Chen et al. (2023) but without any theoretical guarantees.

## 4.2 CONTINUOUS-TIME CONVERGENCE

To analyze the convergence of MFL-ABR, we require the following alternative assumptions for $\mathcal{L}$ which are taken from Lascu et al. (2023).

**Assumption 3** (Regularity of $\mathcal{L}$ for MFL-ABR). *We assume that $\mathcal{L}$ is convex-concave and admits $C^1$ functional derivatives which are uniformly bounded as $\|\frac{\delta \mathcal{L}}{\delta \mu}(\mu, \nu)\|_\infty \le C_\mu$, $\|\frac{\delta \mathcal{L}}{\delta \nu}(\mu, \nu)\|_\infty \le C_\nu$ for constants $C_\mu, C_\nu > 0$. Furthermore, $L$ admits second order functional derivatives which are uniformly bounded as $\|\frac{\delta^2 \mathcal{L}}{\delta \mu^2}\|_\infty \le C_{\mu\mu}$, $\|\frac{\delta^2 \mathcal{L}}{\delta \mu \delta \nu}\|_\infty \le C_{\mu\nu}$, $\|\frac{\delta^2 \mathcal{L}}{\delta \nu^2}\|_\infty \le C_{\nu\nu}$ and symmetric in the sense that $\frac{\delta^2 \mathcal{L}}{\delta \mu \delta \nu}(\mu, \nu, x, y) = \frac{\delta^2 \mathcal{L}}{\delta \nu \delta \mu}(\mu, \nu, y, x)$ for all $\mu, \nu$ and $x \in \mathcal{X}, y \in \mathcal{Y}$.*

Existence and uniqueness of the MNE still hold under this assumption as proved in Lascu et al. (2023). Also, $\widehat{\mu}_t, \widehat{\nu}_t$ both satisfy the LSI with constant $\alpha = r_\mu \exp\left(-\frac{4C_\mu}{\lambda}\right) \wedge r_\nu \exp\left(-\frac{4C_\nu}{\lambda}\right)$ by the Holley-Stroock argument; we take the minimum since it dominates the overall convergence rate.

We now present the overall convergence result for MFL-ABR. The proof, given in Appendix D.2, is a combination of a time-discretized version of the argument in Lascu et al. (2023) for the outer loop and a TV distance perturbation analysis for the inner loop developed in Appendix D.1.

**Theorem 4.1** (Convergence of MFL-ABR). *The NI error of the MFL-ABR outer loop output after $k$ steps is bounded for a constant $C$ as*

$$\mathrm{NI}(\mu_k, \nu_k) \le 2(C_\mu + C_\nu)\exp(-\beta k) + 12\lambda^{-\frac{1}{2}}(C_\mu^{\frac{3}{2}} + C_\nu^{\frac{3}{2}})\exp(-\alpha\lambda\tau) + C\beta.$$

Hence we achieve linear convergence in the outer loop iteration, with a uniform-in-time inner loop error linearly converging in $\tau$ and time discretization error proportional to $\beta$. It follows that an $\epsilon$-MNE may be obtained in $k = O(\frac{1}{\epsilon}\log\frac{1}{\epsilon})$ outer loop iterations with $\beta = O(\epsilon)$ and $\tau = O(\log\frac{1}{\epsilon})$.

We do not give a discrete-particle analysis of MFL-ABR and instead remark that discretization of the fixed-drift inner loop is trivial, while Theorem 4.1 already covers the outer-loop error due to finite $\tau$ and nonzero $\beta$. The remaining element is particle discretization analysis of the outer loop momentum sampling which we feel strays from the scope of this work.

## 5 APPLICATIONS TO ZERO-SUM MARKOV GAMES

### 5.1 BILINEAR PROBLEMS

We briefly discuss the case when $\mathcal{L}$ is bilinear, that is $\mathcal{L}(\mu, \nu) = \iint Q(x,y)\mu(\mathrm{d}x)\nu(\mathrm{d}y)$ for a $C^1$ reward $Q : \mathcal{X} \times \mathcal{Y} \to \mathbb{R}$. Assumption 2 is easily verified under the conditions $\|\nabla_x Q\|_\infty \le Q_x$ and $\nabla_x Q$ is $L_x^i$-Lipschitz in each coordinate $i = 1, \cdots, d_\mathcal{X}$ by taking $M_\mu = Q_x, K_\mu = L_\mu = \|L_x\|_2$, while Assumption 3 holds if $Q$ is uniformly bounded. The averaged gradient in (4) is then equal to

$$\frac{1}{B_t}\int_0^t \beta_s \nabla_x \frac{\delta \mathcal{L}}{\delta \mu}(\mu_s, \nu_s)(X_t)\,\mathrm{d}s + \lambda \nabla_x U^\mu(X_t) = \int_\mathcal{Y} \nabla_x Q(X_t, y)\bar{\nu}_t(\mathrm{d}y) + \lambda \nabla_x U^\mu(X_t);$$

the drift only depends on the history through the average distributions $\bar{\mu}_t, \bar{\nu}_t$. Therefore, instead of storing and iterating over all previous states which could be computationally prohibitive, we only require the rolling averages to be stored and updated alongside the primary dynamics. In the discrete case, this means that we store the length $N$ arrays $\overline{\mathscr{X}}, \overline{\mathscr{Y}}$ alongside $\mathscr{X}_k, \mathscr{Y}_k$ and perform

$$X_{k+1}^i \leftarrow X_k^i - \frac{\eta}{N} \sum_{m=1}^N \nabla_x Q(X_k^i, \overline{Y}^m) - \lambda\eta\nabla_x U^\mu(X_k^i) + \sqrt{2\lambda\eta}\xi_k^{\mu,i},$$
$$Y_{k+1}^i \leftarrow Y_k^i + \frac{\eta}{N} \sum_{m=1}^N \nabla_y Q(\overline{X}^m, Y_k^i) - \lambda\eta\nabla_y U^\nu(Y_k^i) + \sqrt{2\lambda\eta}\xi_k^{\nu,i}.$$

We then discard $\lfloor \beta_{k+1} N / B_{k+1} \rfloor$ particles from $\overline{\mathscr{X}}, \overline{\mathscr{Y}}$ and replace with random samples drawn from $\mathscr{X}_{k+1}, \mathscr{Y}_{k+1}$, respectively. After $K$ steps, the arrays $\overline{\mathscr{X}}, \overline{\mathscr{Y}}$ are returned.

Thus, both algorithms only require 4 arrays to be stored and updated (the inner and outer states for MFL-ABR), incurring no significant computational cost compared to MFL-DA (2 arrays).

## 5.2 ZERO-SUM MARKOV GAMES

In this section we outline an application to policy optimization in Markov games. We consider the two-player zero-sum discounted Markov game defined by the tuple $\mathfrak{M} = (\mathcal{S}, \mathcal{X}, \mathcal{Y}, P, r, \gamma)$ with continuous action spaces $\mathcal{X} = \mathbb{R}^{d_\mathcal{X}}, \mathcal{Y} = \mathbb{R}^{d_\mathcal{Y}}$, finite state space $\mathcal{S}$, reward $r : \mathcal{S} \times \mathcal{X} \times \mathcal{Y} \to \mathbb{R}$, transition kernel $P : \mathcal{S} \times \mathcal{X} \times \mathcal{Y} \to \mathcal{P}(\mathcal{S})$ and discount factor $\gamma \in [0, 1)$. The strategies of the min and max players are represented by $\mu = \mu(s) = \mu(\cdot|s) : \mathcal{S} \to \mathcal{P}_2(\mathcal{X})$ and $\nu : \mathcal{S} \to \mathcal{P}_2(\mathcal{Y})$.

The regularized value and $Q$-functions are defined for all $s \in \mathcal{S}$ as

$$V_\lambda^{\mu,\nu}(s) = \mathbb{E}\left[\sum_{t=0}^\infty \gamma^t \left(r(s_t, x_t, y_t) + \lambda\log\frac{\mu(x_t|s_t)}{\rho^\mu(x_t)} - \lambda\log\frac{\nu(y_t|s_t)}{\rho^\nu(y_t)}\right)\Bigg| s_0 = s\right],$$
$$Q_\lambda^{\mu,\nu}(x, y|s) = r(s, x, y) + \gamma\mathbb{E}_{s'\sim P(\cdot|s,x,y)}[V_\lambda^{\mu,\nu}(s')],$$

where the expectation is taken over all trajectories $s_0, x_0, y_0, s_1, \cdots$ generated by $x_k \sim \mu(\cdot|s_k)$, $y_k \sim \nu(\cdot|s_k)$ and $s_{k+1} \sim P(\cdot|s_k, x_k, y_k)$. Our goal is to find the MNE which solves the distributional minimax problem $\min_{\mu:\mathcal{S}\to\mathcal{P}_2(\mathcal{X})} \max_{\nu:\mathcal{S}\to\mathcal{P}_2(\mathcal{Y})} V_\lambda^{\mu,\nu}(s)$ for all states simultaneously; a detailed introduction to the topic can be found in e.g. Sutton & Barto (2018); Cen et al. (2023). For zero-sum Markov games, the MNE is also called the regularized Markov perfect equilibrium.

To this end, we propose the following two-step iterative scheme. For simplicity, we only consider the continuous-time MFLD and assume full knowledge of game quantities as well as the existence and uniqueness of the MNE $(\mu^*, \nu^*)$ which is known for finite Markov games (Shapley, 1953).

**Step 1** (Minimax dynamics). Given $Q^{(k)}$, run MFL-AG or MFL-ABR for each state $s \in \mathcal{S}$ for sufficient time to obtain an $\epsilon_\mathcal{L}$-MNE $(\mu^{(k)}(s), \nu^{(k)}(s))$ for the regularized minimax problem

$$\mathcal{L}_\lambda(\mu, \nu; Q^{(k)}(s)) := \iint_{\mathcal{X}\times\mathcal{Y}} Q^{(k)}(x, y|s)\mu(\mathrm{d}x)\nu(\mathrm{d}y) + \lambda\,\mathrm{KL}(\mu\|\rho^\mu) - \lambda\,\mathrm{KL}(\nu\|\rho^\nu).$$

**Step 2** (Approximate value iteration). For each $s$, set $V^{(k+1)}(s) = \mathcal{L}_\lambda(\mu^{(k)}(s), \nu^{(k)}(s); Q^{(k)}(s))$ and update the $Q$-function by letting $Q^{(k+1)} = Q(\cdot, \cdot|s)$ satisfying

$$\left|Q(x, y|s) - r(s, x, y) - \gamma\mathbb{E}_{s'\sim P(\cdot|s,x,y)}[V^{(k+1)}(s')]\right| \le \epsilon_Q,$$

where $\epsilon_Q > 0$ quantifies a model error. In practice, $Q^{(k+1)}$ can be obtained by any offline RL algorithm with function approximation, e.g. a deep neural network, as long as the sup norm of Bellman error to the update is bounded. Moreover, we assume the gradients $\nabla_x Q, \nabla_y Q$ are bounded and Lipschitz and can be queried freely.

With this scheme, we are guaranteed convergence to the MNE. The proof is identical to the discrete strategy case (Cen et al., 2021, Theorem 3) and is included in Appendix A.3 for completeness.

**Proposition 5.1.** *The above scheme linearly converges to the optimal value function as*

$$\|V^{(k)} - V^*\|_\infty \le \gamma^k\|V^{(0)} - V^*\|_\infty + \frac{\epsilon_\mathcal{L} + \epsilon_Q}{1 - \gamma}.$$

This proposition shows that our two-step algorithm finds the Markov perfect equilibrium at a linear rate of convergence up to a sum of the optimization error for learning the MNE of the inner problem, and the Bellman error for estimating the $Q$-functions.

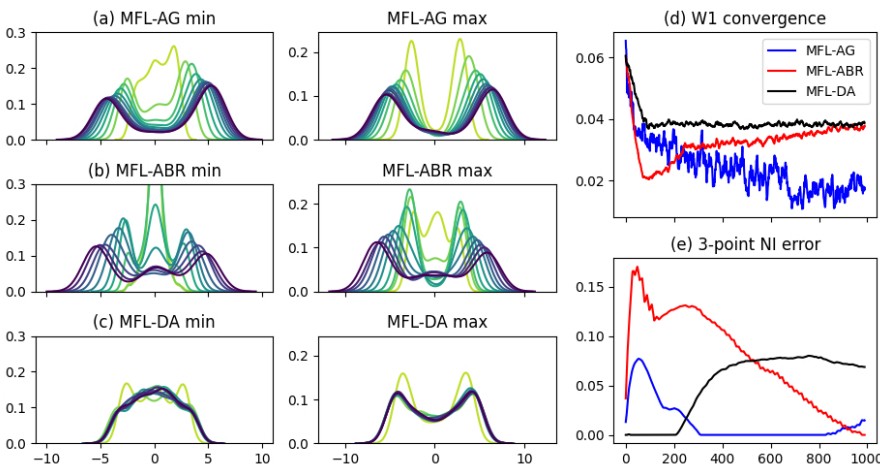

Figure 1: Density evolution of (a) MFL-AG, (b) MFL-ABR, and (c) MFL-DA every 100 epochs. (d) Convergence speed measured in $W_1$ distance. (e) Optimality comparison via 3-point NI error.

## 6  NUMERICAL EXPERIMENTS

We test our proposed algorithms and compare against ordinary descent ascent dynamics in a simulated setting. We consider $d_{\mathcal{X}} = d_{\mathcal{Y}} = 1$ and optimize the bilinear objective

$$\mathcal{L}(\mu, \nu) = \iint Q(x,y)\mu(\mathrm{d}x)\nu(\mathrm{d}y), \quad Q(x,y) = (1 + e^{-(x-y)^2})^{-1}.$$

The sigmoid nonlinearity introduces nontrivial interactions between the min and max policies. We also take regularizers $\rho^\mu = \rho^\nu = \mathcal{N}(0,1)$ and $\lambda = 0.01$. Both MFL-AG with $r = 1$ and MFL-DA are run with 1,000 particles for 1,000 epochs with learning rate $\eta = 0.3$. MFL-ABR is run with 1,000 particles for 50 outer loop iterations with 20 inner iterations per loop and $\eta = 0.3, \beta = 0.15$. We implement the rolling average update for MFL-AG in Section 5.1 and a 'warm start' scheme for MFL-ABR where the inner loop is not re-initialized for stability. We report the results in Figure 1.

Figure 1(a)-(c) show kernel density plots of the evolving min and max policies $\mu_{\mathscr{X}_k}, \nu_{\mathscr{Y}_k}$ for each algorithm per every 100 epochs. MFL-AG and MFL-ABR converge to similar solutions while MFL-DA converges to a different distribution much more rapidly. Figure 1(d) plots convergence speed by computing the sum of the empirical Wasserstein distances $W_1(\mu_{\mathscr{X}_k}, \mu_{\mathscr{X}_{k+1}}) + W_1(\nu_{\mathscr{Y}_k}, \nu_{\mathscr{Y}_{k+1}})$.

To compare the optimality of the outputs $(\mathscr{X}^i, \mathscr{Y}^i)$ $(i = 0, 1, 2)$ of the three algorithms, we use the *3-point NI error* $\mathrm{NI}^i := \max_j \mathcal{L}_\lambda(\mu_{\mathscr{X}^i}, \nu_{\mathscr{Y}^j}) - \min_j \mathcal{L}_\lambda(\mu_{\mathscr{X}^j}, \nu_{\mathscr{Y}^i})$ which measures relative optimality analogous to a $3 \times 3$ payoff matrix. The values are reported in Figure 1(e). While the MFL-DA output is initially the desirable strategy due to its rapid convergence, MFL-AG gradually optimizes and soon dominates MFL-DA with zero error, which is later followed by MFL-ABR. We therefore conclude MFL-AG and MFL-ABR can substantially outperform ordinary descent ascent despite the slower convergence rates.

## 7  CONCLUSION

In this paper, we developed the first symmetric mean-field Langevin dynamics for entropy-regularized minimax problems with global convergence guarantees. We proposed the single-loop MFL-AG algorithm and proved average-iterate convergence to the MNE. We also established a new uniform-in-time analysis of propagation of chaos that accounts for dependence on history using novel perturbative techniques. Furthermore, we proposed the double-loop MFL-ABR algorithm and proved time-discretized linear convergence of the outer loop.

Our work represents early steps towards an understanding of mean-field dynamics for multiple learning agents and opens up further avenues of investigation. Some interesting directions are developing a single-loop symmetric algorithm with last-iterate convergence, studying nonconvex-nonconcave parametrizations or applications to multi-agent reinforcement learning.

## ACKNOWLEDGMENTS

JK was partially supported by JST CREST (JPMJCR2015) and Toshiba Corporation. KO was partially supported by JST ACT-X (JPMJAX23C4). TS was partially supported by JSPS KAKENHI (20H00576) and JST CREST (JPMJCR2115).

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

TABLE OF CONTENTS

---

**Algorithm 2** Mean-field Langevin Anchored Best Response

---

**Require:** temperature $\lambda$, outer loop iteration $K$, inner loop iteration $L$, learning rate $\eta$, number of particles $N$, exponent $r$

**Initialization:** $\mathscr{X}_0 \sim \rho^\mu$, $\mathscr{Y}_0 \sim \rho^\nu$

  **for** $k = 0, \cdots, K-1$ **do**

    Sample $\mathscr{X}_0^\dagger \sim \rho^\mu$, $\mathscr{Y}_0^\dagger \sim \rho^\nu$

    **for** $\ell = 0, \cdots, L-1$ **do**

      For all particles $i = 1, \cdots, N$ sample $\xi_\ell^{\mu,i} \sim \mathcal{N}(0, \mathrm{I}_{d_\mathcal{X}})$, $\xi_\ell^{\nu,i} \sim \mathcal{N}(0, \mathrm{I}_{d_\mathcal{Y}})$ and update

      $X_{\ell+1}^{\dagger i} \leftarrow X_\ell^{\dagger i} - \eta \nabla_x \frac{\delta\mathcal{L}}{\delta\mu}(\mu_{\mathscr{X}_k}, \nu_{\mathscr{Y}_k})(X_\ell^{\dagger i}) - \lambda\eta \nabla_x U^\mu(X_\ell^{\dagger i}) + \sqrt{2\lambda\eta}\xi_\ell^{\mu,i}$

      $Y_{\ell+1}^{\dagger i} \leftarrow Y_\ell^{\dagger i} + \eta \nabla_y \frac{\delta\mathcal{L}}{\delta\nu}(\mu_{\mathscr{X}_k}, \nu_{\mathscr{Y}_k})(Y_\ell^{\dagger i}) - \lambda\eta \nabla_y U^\nu(Y_\ell^{\dagger i}) + \sqrt{2\lambda\eta}\xi_\ell^{\nu,i}$

    **end for**

    Discard $\lfloor\beta N\rfloor$ particles from $\mathscr{X}_k, \mathscr{Y}_k$ and replace with random samples from $\mathscr{X}_L^\dagger, \mathscr{Y}_L^\dagger$, resp.

  **end for**

  **return** $\mathscr{X}_K, \mathscr{Y}_K$

---

# APPENDIX

## A PRELIMINARIES

### A.1 OPTIMAL TRANSPORT

We begin by introducing basic concepts and inequalities from optimal transport theory that will be useful in analyzing the behavior of Langevin dynamics.

**Definition A.1** (*p*-Wasserstein metric). For $p \in [1, \infty)$, let $\mathcal{P}_p(\mathbb{R}^d)$ be the space of probability measures on $\mathbb{R}^d$ with finite $p$th moment. The $p$-Wasserstein distance between $\mu, \nu \in \mathcal{P}_p(\mathbb{R}^d)$ is defined as

$$W_p(\mu, \nu) = \left(\inf_{\gamma \in \Pi(\mu,\nu)} \int_{\mathbb{R}^d} \|x - y\|^p \, \mathrm{d}\gamma(x,y)\right)^{\frac{1}{p}}$$

where $\Pi(\mu, \nu)$ denotes the set of joint distributions on $\mathbb{R}^d \times \mathbb{R}^d$ with marginal laws $\mu$ and $\nu$ on the first and second factors, respectively. By Kantorovich-Rubinstein duality, the metric $W_1$ can also be written as

$$W_1(\mu, \nu) = \sup_{\|f\|_{\mathrm{Lip}} \leq 1} \int_{\mathbb{R}^d} f \, \mathrm{d}\mu - \int_{\mathbb{R}^d} f \, \mathrm{d}\nu.$$

**Definition A.2** (Log-Sobolev inequality). A probability measure $\nu \in \mathcal{P}_2(\mathbb{R}^d)$ is said to satisfy the logarithmic Sobolev inequality (LSI) with constant $\alpha > 0$ if for any smooth function $f : \mathbb{R}^d \to \mathbb{R}$,

$$\mathrm{Ent}_\nu(f^2) := \mathbb{E}_\nu[f^2 \log f^2] - \mathbb{E}_\nu[f^2] \log \mathbb{E}_\nu[f^2] \leq \frac{2}{\alpha} \mathbb{E}_\nu[\|\nabla f\|_2^2].$$

For any measure $\mu \in \mathcal{P}_2(\mathbb{R}^d)$ absolutely continuous with respect to $\nu$, the LSI implies that KL divergence is upper bounded by the relative Fisher information,

$$\mathrm{KL}(\mu\|\nu) \leq \frac{1}{2\alpha} \mathbb{E}_\mu\left[\left\|\nabla \log \frac{\mathrm{d}\mu}{\mathrm{d}\nu}\right\|_2^2\right].$$

**Proposition A.3** (Bakry & Émery, 1985). *If $f : \mathbb{R}^d \to \mathbb{R}$ is a function such that $\nabla^2 f \succeq \alpha I_d$, the probability density $p \propto \exp(-f)$ satisfies the LSI with constant $\alpha$.*

**Proposition A.4** (Holley & Stroock, 1987). *Let $p$ be a density on $\mathbb{R}^d$ satisfying the LSI with constant $\alpha$. For a bounded function $B : \mathbb{R}^d \to \mathbb{R}$, the perturbed distribution*

$$p_B(x)dx = \frac{\exp(B(x))p(x)}{\mathbb{E}_p[\exp(B(x))]} \, \mathrm{d}x$$

*also satisfies the LSI with constant $\alpha/\exp(4\|B\|_\infty)$.*

**Definition A.5** (Poincaré and Talagrand's inequalities). A probability measure $\nu \in \mathcal{P}_2(\mathbb{R}^d)$ is said to satisfy the Poincaré inequality with constant $\alpha > 0$ if for any smooth function $f : \mathbb{R}^d \to \mathbb{R}$,

$$\text{Var}_\nu(f) := \mathbb{E}_\nu[f^2] - (\mathbb{E}_\nu[f])^2 \leq \frac{1}{\alpha} \mathbb{E}_\nu[\|\nabla f\|_2^2].$$

Moreover, $\nu$ is said to satisfy Talagrand's inequality with constant $\alpha > 0$ if for any measure $\mu \in \mathcal{P}_2(\mathbb{R}^d)$ absolutely continuous with respect to $\nu$, the 2-Wasserstein distance is upper bounded as

$$\frac{\alpha}{2} W_2^2(\mu, \nu) \leq \text{KL}(\mu \| \nu).$$

If $\nu$ satisfies the LSI with constant $\alpha$, then it satisfies the Poincaré inequality with the same constant. We also have the following implication.

**Theorem A.6** (Otto & Villani, 2000). *If a probability measure $\nu \in \mathcal{P}_2(\mathbb{R}^d)$ satisfies the LSI with constant $\alpha$, then it satisfies Talagrand's inequality with the same constant.*

**Proof of Proposition 3.2.** We take the stronger of the two bounds in Lemma 2.1 of Bardet et al. (2018) and Theorem 2.7 of Cattiaux & Guillin (2022); the latter removes the exponential dependency on $d_\mathcal{X}$ in exchange for more complicated polynomial terms. See Lemma 6 of Suzuki et al. (2023) for more details.

## A.2 Mixed Nash Equilibrium

**Definition A.7** (functional derivative). Let $F$ be a functional on $\mathcal{P}_2(\mathbb{R}^d)$. The functional derivative $\frac{\delta F}{\delta \mu}$ at $\mu \in \mathcal{P}_2(\mathbb{R}^d)$ is defined as a functional $\mathcal{P}_2(\mathbb{R}^d) \times \mathbb{R}^d \to \mathbb{R}$ satisfying for all $\nu \in \mathcal{P}_2(\mathbb{R}^d)$,

$$\left. \frac{\mathrm{d}}{\mathrm{d}\epsilon} F(\mu + \epsilon(\nu - \mu)) \right|_{\epsilon=0} = \int_{\mathbb{R}^d} \frac{\delta F}{\delta \mu}(\mu)(x)(\nu - \mu)(\mathrm{d}x).$$

As the functional derivative is defined up to additive constants, we impose the additional condition $\int_{\mathbb{R}^d} \frac{\delta F}{\delta \mu}(\mu) \, \mathrm{d}\mu = 0$. Furthermore, $F$ is defined to be convex if its satisfies the convexity condition for all $\nu \in \mathcal{P}_2(\mathbb{R}^d)$:

$$F(\nu) \geq F(\mu) + \int_{\mathbb{R}^d} \frac{\delta F}{\delta \mu}(\mu)(x)(\nu - \mu)(\mathrm{d}x).$$

Finally, $F$ is defined to be concave if $-F$ is convex.

**Proof of Proposition 2.1.** Recall that the 2-Wasserstein distance is finite and metrizes weak convergence on $\mathcal{P}_2(\mathbb{R}^d)$ (Villani, 2009, Theorem 6.9). Also, the divergence $\mu \mapsto \text{KL}(\mu \| \rho^\mu)$ is proper and lower semi-continuous with respect to the weak topology (Lanzetti et al., 2022). Furthermore, $\rho^\mu$ satisfies Talagrand's inequality with constant $r_\mu$ by Theorem A.6 so that the map $\mu \mapsto \mathcal{L}_\lambda(\mu, \nu)$ is strongly convex. Hence the minimizer of $\mu \mapsto \mathcal{L}_\lambda(\mu, \nu)$ is unique, and similarly the maximizer of $\nu \mapsto \mathcal{L}_\lambda(\mu, \nu)$ is unique. Existence of the MNE is now guaranteed by Theorem 3.6 in Conforti et al. (2020) by verifying Assumption 2.1 and conditions (i)-(iii).

For uniqueness, suppose to the contrary that $(\mu^*, \nu^*)$, $(\tilde{\mu}^*, \tilde{\nu}^*)$ are two distinct solutions of (1). The optimality conditions yield the chain of strict inequalities

$$\mathcal{L}_\lambda(\mu^*, \nu^*) > \mathcal{L}_\lambda(\mu^*, \tilde{\nu}^*) > \mathcal{L}_\lambda(\tilde{\mu}^*, \tilde{\nu}^*) > \mathcal{L}_\lambda(\tilde{\mu}^*, \nu^*) > \mathcal{L}_\lambda(\mu^*, \nu^*),$$

a contradiction. Finally, the first-order conditions follow from Corollary 3.3 in Conforti et al. (2020), adjusting the base measures as to be different for $\mu, \nu$. □

**Proof of Lemma 3.5.** By convex-concavity of $\mathcal{L}$ and the first-order condition (3),

$$\text{NI}(\mu, \nu) \geq \mathcal{L}_\lambda(\mu, \nu^*) - \mathcal{L}_\lambda(\mu^*, \nu)$$

$$\geq \int_{\mathcal{X}} \frac{\delta \mathcal{L}}{\delta \mu}(\mu^*, \nu^*)(\mu - \mu^*)(\mathrm{d}x) + \lambda \text{KL}(\mu \| \rho^\mu) - \lambda \text{KL}(\nu^* \| \rho^\nu)$$

$$- \int_{\mathcal{Y}} \frac{\delta \mathcal{L}}{\delta \nu}(\mu^*, \nu^*)(\nu - \nu^*)(\mathrm{d}y) - \lambda \text{KL}(\mu^* \| \rho^\mu) + \lambda \text{KL}(\nu \| \rho^\nu)$$

$$= -\int_{\mathcal{X}} \lambda \log \frac{\mu^*}{\rho^\mu}(\mu - \mu^*)(\mathrm{d}x) + \lambda \operatorname{KL}(\mu\|\rho^\mu) - \lambda \operatorname{KL}(\nu^*\|\rho^\nu)$$

$$- \int_{\mathcal{Y}} \lambda \log \frac{\nu^*}{\rho^\nu}(\nu - \nu^*)(\mathrm{d}y) - \lambda \operatorname{KL}(\mu^*\|\rho^\mu) + \lambda \operatorname{KL}(\nu\|\rho^\nu)$$

$$= \lambda \operatorname{KL}(\mu\|\mu^*) + \lambda \operatorname{KL}(\nu\|\nu^*).$$

$\square$

### A.3 PROOF OF PROPOSITION 5.1

We use the bound $|\mathcal{L}_\lambda(\mu, \nu) - \mathcal{L}_\lambda(\mu^*, \nu^*)| \le \operatorname{NI}(\mu, \nu)$ which can be shown by the following string of inequalities,

$$\mathcal{L}_\lambda(\mu, \nu) - \mathcal{L}_\lambda(\mu^*, \nu^*) \le \max_{\nu'} \mathcal{L}_\lambda(\mu, \nu') - \mathcal{L}_\lambda(\mu^*, \nu) \le \max_{\nu'} \mathcal{L}_\lambda(\mu, \nu') - \min_{\mu'} \mathcal{L}_\lambda(\mu', \nu),$$

$$\mathcal{L}_\lambda(\mu, \nu) - \mathcal{L}_\lambda(\mu^*, \nu^*) \ge \min_{\mu'} \mathcal{L}_\lambda(\mu', \nu) - \mathcal{L}_\lambda(\mu, \nu^*) \ge \min_{\mu'} \mathcal{L}_\lambda(\mu', \nu) - \max_{\nu'} \mathcal{L}_\lambda(\mu, \nu').$$

Denoting the ideal minimax update in Step 1 as

$$\widetilde{V}^{(k+1)}(s) = \min_{\mu \in \mathcal{P}_2(\mathcal{X})} \max_{\nu \in \mathcal{P}_2(\mathcal{Y})} \mathcal{L}_\lambda(\mu, \nu; Q^{(k)}(s)),$$

this implies

$$|V^{(k+1)}(s) - \widetilde{V}^{(k+1)}(s)| \le \operatorname{NI}(\mu^{(k)}(s), \nu^{(k)}(s)) \le \epsilon_\mathcal{L}.$$

Now denote the ideal value iteration in Step 2 as

$$\widetilde{Q}^{(k)}(s) = r(s, x, y) + \gamma \mathbb{E}_{s' \sim P(\cdot|s,x,y)}[V^{(k)}(s)]$$

and note that the optimal value and $Q$-functions $V^* = V_\lambda^{\mu^*, \nu^*}$, $Q^* = Q_\lambda^{\mu^*, \nu^*}$ satisfy the Bellman equation

$$Q^*(x, y|s) = r(s, x, y) + \gamma \mathbb{E}_{s' \sim P(\cdot|s,x,y)}[V^*(s')].$$

Hence we bound

$$\|V^{(k+1)} - V^*\|_\infty \le \epsilon_\mathcal{L} + \|\widetilde{V}^{(k+1)} - V^*\|_\infty$$

$$\le \epsilon_\mathcal{L} + \sup_{\mu,\nu,s} \big| \mathcal{L}_\lambda(\mu, \nu; Q^{(k)}(s)) - \mathcal{L}_\lambda(\mu, \nu; Q^*(s)) \big|$$

$$\le \epsilon_\mathcal{L} + \|Q^{(k)} - Q^*\|_\infty$$

$$\le \epsilon_\mathcal{L} + \|Q^{(k)} - \widetilde{Q}^{(k)}\|_\infty + \|\widetilde{Q}^{(k)} - Q^*\|_\infty$$

$$\le \epsilon_\mathcal{L} + \epsilon_Q + \gamma\|V^{(k)} - V^*\|_\infty.$$

Therefore by Gronwall's lemma we conclude that

$$\|V^{(k)} - V^*\|_\infty \le \gamma^k \|V^{(0)} - V^*\|_\infty + \frac{\epsilon_\mathcal{L} + \epsilon_Q}{1 - \gamma}.$$

$\square$

## B CONVERGENCE ANALYSIS OF MFL-AG

### B.1 PROOF OF PROPOSITION 3.1

Some definitions are in order. Denote by $C_{\mathcal{X},T} = C([0, T], \mathcal{X})$ the space of continuous sample paths on $\mathcal{X}$ and by $\mathcal{M}(C_{\mathcal{X},T})$ the space of probability measures on $C_{\mathcal{X},T}$. We define two versions of the *lifted* 1-Wasserstein distance on $\mathcal{M}(C_{\mathcal{X},T})$ as

$$\widetilde{W}_{1,T}(\mu, \mu') = \inf_\gamma \int \sup_{t \le T} \|\omega(t) - \omega'(t)\| \, \mathrm{d}\gamma(\omega, \omega') \wedge 1,$$

$$W_{1,T}(\mu, \mu') = \inf_\gamma \int \sup_{t \le T} \|\omega(t) - \omega'(t)\| \wedge 1 \, \mathrm{d}\gamma(\omega, \omega')$$

where the infimum runs over all couplings $\gamma \in \mathcal{M}(C_{\mathcal{X},T} \times C_{\mathcal{X},T})$ with marginal laws $\mu, \mu'$. The inner truncated metric $W_{1,T}$ is complete, nondecreasing in $T$ and metrizes the weak topology on $\mathcal{M}(C_{\mathcal{X},T})$ (Dobrushin, 1970); the outer truncation $\widetilde{W}_{1,T}$ serves to upper bound $W_{1,T}$. We repeat the construction for $\mathcal{Y}$ and extend $W_{1,T}, \widetilde{W}_{1,T}$ to the product space $\mathcal{M}(C_{\mathcal{X},T}) \times \mathcal{M}(C_{\mathcal{Y},T})$ as

$$W_{1,T}((\mu,\nu),(\mu',\nu')) = W_{1,T}(\mu,\mu') + W_{1,T}(\nu,\nu'),$$

etc. Now define $\Phi : \mathcal{M}(C_{\mathcal{X},T}) \times \mathcal{M}(C_{\mathcal{Y},T}) \to \mathcal{M}(C_{\mathcal{X},T}) \times \mathcal{M}(C_{\mathcal{Y},T})$ as the map which associates to the pair $(\mu, \nu)$ the laws of the stochastic processes $(X_t)_{t \leq T}, (Y_t)_{t \leq T}$,

$$X_t = X_0 - \int_0^t \frac{1}{B_s} \int_0^s \beta_r \nabla_x \frac{\delta \mathcal{L}}{\delta \mu}(\mu_r, \nu_r)(X_s) \, dr + \lambda \nabla_x U^\mu(X_s) \, ds + \sqrt{2\lambda} W_t^\mu,$$

$$Y_t = Y_0 + \int_0^t \frac{1}{B_s} \int_0^s \beta_r \nabla_y \frac{\delta \mathcal{L}}{\delta \nu}(\mu_r, \nu_r)(Y_s) \, dr - \lambda \nabla_y U^\nu(Y_s) \, ds + \sqrt{2\lambda} W_t^\nu$$

for $0 \leq t \leq T$, where $\mu_t, \nu_t$ denote the marginal distributions of $\mu, \nu$ at time $t$ and in particular $\mu_0, \nu_0$ are the prescribed initial distributions. A solution to (4) then corresponds precisely to a fixed point of $\Phi$.

**Lemma B.1.** *There exists a constant $C_T > 0$ so that for any $0 \leq t \leq T$,*

$$\widetilde{W}_{1,t}(\Phi(\mu,\nu), \Phi(\mu',\nu')) \leq C_T \int_0^t \widetilde{W}_{1,s}((\mu,\nu),(\mu',\nu')) \, ds.$$

*Proof.* First note that for any $0 \leq s \leq t \leq T$,

$$\widetilde{W}_{1,t}(\mu,\mu') \geq \inf_\gamma \int \|\omega(s) - \omega'(s)\| \, d\gamma(\omega, \omega') \wedge 1 \geq W_1(\mu_s, \mu_s') \wedge 1.$$

Let $(X_t')_{t \leq T}, (Y_t')_{t \leq T}$ denote the synchronous processes

$$X_t' = X_0 - \int_0^t \frac{1}{B_s} \int_0^s \beta_r \nabla_x \frac{\delta \mathcal{L}}{\delta \mu}(\mu_r', \nu_r')(X_s') \, dr + \lambda \nabla_x U^\mu(X_s') \, ds + \sqrt{2\lambda} W_t^\mu,$$

$$Y_t' = Y_0 + \int_0^t \frac{1}{B_s} \int_0^s \beta_r \nabla_y \frac{\delta \mathcal{L}}{\delta \nu}(\mu_r', \nu_r')(Y_s') \, dr - \lambda \nabla_y U^\nu(Y_s') \, ds + \sqrt{2\lambda} W_t^\nu$$

corresponding to another pair of distributions $(\mu', \nu')$. Then by Assumption 2,

$$\sup_{s \leq t} \|X_s - X_s'\|$$

$$\leq \int_0^t \sup_{r \leq s} \left\| \nabla_x \frac{\delta \mathcal{L}}{\delta \mu}(\mu_r, \nu_r)(X_s) - \nabla_x \frac{\delta \mathcal{L}}{\delta \mu}(\mu_r', \nu_r')(X_s') \right\| + \lambda \|\nabla_x U^\mu(X_s) - \nabla_x U^\mu(X_s')\| \, ds$$

$$\leq \int_0^t (K_\mu + \lambda R_\mu) \|X_s - X_s'\| + \sup_{r \leq s} L_\mu (W_1(\mu_r, \mu_r') + W_1(\nu_r, \nu_r')) \wedge 2M_\mu \, ds$$

$$\leq (K_\mu + \lambda R_\mu) \int_0^t \|X_s - X_s'\| \, ds + (L_\mu \vee 2M_\mu) \int_0^t \sup_{r \leq s} W_1(\mu_r, \mu_r') \wedge 1 + W_1(\nu_r, \nu_r') \wedge 1 \, ds.$$

Thus by Gronwall's lemma we obtain

$$\sup_{s \leq t} \|X_s - X_s'\| \leq (L_\mu \vee 2M_\mu) e^{(K_\mu + \lambda R_\mu)T} \int_0^t \sup_{r \leq s} W_1(\mu_r, \mu_r') \wedge 1 + W_1(\nu_r, \nu_r') \wedge 1 \, ds.$$

Then defining the constant $C_T = (L_\mu \vee 2M_\mu) e^{(K_\mu + \lambda R_\mu)T} + (L_\nu \vee 2M_\nu) e^{(K_\nu + \lambda R_\nu)T}$, by taking the joint distribution coupling of $(X_t)_{t \leq T}$ and $(X_t')_{t \leq T}$ we have

$$\widetilde{W}_{1,t}(\Phi(\mu,\nu), \Phi(\mu',\nu')) \leq C_T \int_0^t \sup_{r \leq s} W_1(\mu_r, \mu_r') \wedge 1 + W_1(\nu_r, \nu_r') \wedge 1 \, ds,$$

which proves the lemma. $\qquad\square$

We now use the contraction property to prove Proposition 3.1. Starting at any $(\mu, \nu)$ and recursively applying Lemma B.1, we have

$$
\widetilde{W}_{1,T}(\Phi^{k+1}(\mu, \nu), \Phi^k(\mu, \nu)) \leq C_T^k \int_0^T \int_0^{t_1} \cdots \int_0^{t_{k-1}} \widetilde{W}_{1,t_k}(\Phi(\mu, \nu), (\mu, \nu)) \, \mathrm{d}t_k \cdots \mathrm{d}t_2 \, \mathrm{d}t_1
$$

$$
\leq \frac{C_T^k T^k}{k!} \widetilde{W}_{1,T}(\Phi(\mu, \nu), (\mu, \nu)),
$$

so that $\widetilde{W}_{1,T}(\Phi^{k+1}(\mu, \nu), \Phi^k(\mu, \nu)) \to 0$ as $k \to \infty$. Since $\widetilde{W}_{1,T}$ upper bounds $W_{1,T}$, the sequence $(\Phi^k(\mu, \nu))_{k \geq 0}$ is Cauchy and therefore converges to a fixed point of $\Phi$ due to the completeness of $\mathcal{M}(C_{\mathcal{X},T}) \times \mathcal{M}(C_{\mathcal{Y},T})$ with respect to $W_{1,T}$. Similarly, recursively applying Lemma B.1 to two fixed points $(\mu, \nu), (\mu', \nu')$ yields

$$
W_{1,T}((\mu, \nu), (\mu', \nu')) \leq \widetilde{W}_{1,T}((\mu, \nu), (\mu', \nu')) \leq \frac{C_T^k T^k}{k!} \widetilde{W}_{1,T}((\mu, \nu), (\mu', \nu')) \to 0,
$$

hence the fixed point is unique. Finally, truncating the obtained flows $((\mu_t)_{t \leq T}, (\nu_t)_{t \leq T})$ at time $T' < T$ must again yield the fixed point in $\mathcal{M}(C_{\mathcal{X},T'}) \times \mathcal{M}(C_{\mathcal{Y},T'})$ so that we may consistently extend the flows to all time $t \in [0, \infty)$.

## B.2 PROOF OF PROPOSITION 3.3

Write the normalization factor for $\widehat{\mu}_t$ as

$$
Z_t^\mu = \int_{\mathcal{X}} \exp\left( -\frac{1}{\lambda B_t} \int_0^t \beta_s \frac{\delta \mathcal{L}}{\delta \mu}(\mu_s, \nu_s) \, \mathrm{d}s \right) \rho^\mu(\mathrm{d}x).
$$

We first compute the time derivative of the proximal distribution,

$$
\partial_t \log \widehat{\mu}_t = -\partial_t \log Z_t^\mu - \frac{\beta_t}{\lambda B_t} \frac{\delta \mathcal{L}}{\delta \mu}(\mu_t, \nu_t) + \frac{\beta_t}{\lambda B_t^2} \int_0^t \beta_s \frac{\delta \mathcal{L}}{\delta \mu}(\mu_s, \nu_s) \, \mathrm{d}s
$$

$$
= \int_{\mathcal{X}} \left( \frac{\beta_t}{\lambda B_t} \frac{\delta \mathcal{L}}{\delta \mu}(\mu_t, \nu_t) - \frac{\beta_t}{\lambda B_t^2} \int_0^t \beta_s \frac{\delta \mathcal{L}}{\delta \mu}(\mu_s, \nu_s) \, \mathrm{d}s \right) \widehat{\mu}_t(\mathrm{d}\tilde{x})
$$

$$
- \frac{\beta_t}{\lambda B_t} \frac{\delta \mathcal{L}}{\delta \mu}(\mu_t, \nu_t) + \frac{\beta_t}{\lambda B_t^2} \int_0^t \beta_s \frac{\delta \mathcal{L}}{\delta \mu}(\mu_s, \nu_s) \, \mathrm{d}s.
$$

Roughly speaking, the proximal evolution speed is $O(\beta_t/B_t)$ which converges to zero as new information is continually downscaled. However, the maximum total displacement is $O(\log B_t) \to \infty$, ensuring that the algorithm does not prematurely stop before reaching equilibrium.

The time derivative of the KL gap can then be controlled by translating back into KL distance as

$$
\partial_t \mathrm{KL}(\mu_t \| \widehat{\mu}_t) = \int_{\mathcal{X}} \left( \log \frac{\mu_t}{\widehat{\mu}_t} \right) \partial_t \mu_t(\mathrm{d}x) - \int_{\mathcal{X}} (\partial_t \log \widehat{\mu}_t) \, \mu_t(\mathrm{d}x)
$$

$$
= -\lambda \int_{\mathcal{X}} \left\| \nabla_x \log \frac{\mu_t}{\widehat{\mu}_t} \right\|_2^2 \mu_t(\mathrm{d}x)
$$

$$
+ \frac{\beta_t}{\lambda B_t} \int_{\mathcal{X}} \left( \frac{\delta \mathcal{L}}{\delta \mu}(\mu_t, \nu_t) - \frac{1}{B_t} \int_0^t \beta_s \frac{\delta \mathcal{L}}{\delta \mu}(\mu_s, \nu_s) \, \mathrm{d}s \right) (\mu_t - \widehat{\mu}_t)(\mathrm{d}x)
$$

$$
\leq -2\alpha\lambda \cdot \mathrm{KL}(\mu_t \| \widehat{\mu}_t) + \frac{2M_\mu \beta_t}{\lambda B_t} W_1(\mu_t, \widehat{\mu}_t)
$$

by Proposition 3.2. The Wasserstein term is further bounded via Talagrand's inequality as

$$
W_1(\mu_t, \widehat{\mu}_t) \leq W_2(\mu_t, \widehat{\mu}_t) \leq \sqrt{\frac{2}{\alpha_\mu} \mathrm{KL}(\mu_t, \widehat{\mu}_t)}.
$$

Hence

$$
\partial_t \sqrt{\mathrm{KL}(\mu_t \| \widehat{\mu}_t)} \leq -\alpha_\mu \lambda \sqrt{\mathrm{KL}(\mu_t \| \widehat{\mu}_t)} + \frac{M_\mu \beta_t}{\lambda B_t} \sqrt{\frac{2}{\alpha_\mu}}
$$

and using an integrating factor, we conclude (starting from an arbitrary small but positive time $t_0$ to avoid potential singularities at $t = 0$)

$$\exp(\alpha_\mu \lambda t)\sqrt{\mathrm{KL}(\mu_t \| \widehat{\mu}_t)} \le \frac{M_\mu}{\lambda}\sqrt{\frac{2}{\alpha_\mu}}\int_{t_0}^t \frac{\beta_s}{B_s}\exp(\alpha_\mu \lambda s)\,\mathrm{d}s + \exp(\alpha_\mu \lambda t_0)\sqrt{\mathrm{KL}(\mu_{t_0}\|\widehat{\mu}_{t_0})}.$$

In particular, for the weight scheme $\beta_t = t^r$ with $r > -1$, by employing the asymptotic expansion of the exponential integral (Wong, 1989, Section I.4)

$$\mathrm{Ei}(z) = \int_{-\infty}^z \frac{\exp(t)}{t}\,\mathrm{d}t = \frac{\exp(z)}{z}\left(\sum_{k=0}^n \frac{k!}{z^k} + O(|z|^{-(n+1)})\right)$$

we conclude that

$$\mathrm{KL}(\mu_t\|\widehat{\mu}_t) \le \exp(-2\alpha_\mu\lambda t)\left(\frac{(r+1)M_\mu}{\lambda}\sqrt{\frac{2}{\alpha_\mu}}\,\mathrm{Ei}(\alpha_\mu\lambda t) + \mathrm{const.}\right)^2$$

$$\le \frac{2(r+1)^2 M_\mu^2}{\alpha_\mu^3 \lambda^4 t^2} + O(t^{-3}).$$

We also show a boundedness result which guarantees that the flow is in a sense well-behaved.

**Lemma B.2.** *The MFL-AG flow* $(\mu_t, \nu_t)$ *satisfies for all* $t \ge 0$,

$$\mathrm{KL}(\mu_t\|\rho^\mu) \le \mathrm{KL}(\mu_0\|\rho^\mu) \vee \frac{M_\mu^2}{2r_\mu\lambda^2} \quad \textit{and} \quad \mathrm{KL}(\nu_t\|\rho^\nu) \le \mathrm{KL}(\nu_0\|\rho^\nu) \vee \frac{M_\nu^2}{2r_\nu\lambda^2}.$$

*Proof.* The density $\rho^\mu$ satisfies the LSI with constant $r_\mu$ by Proposition A.3 so that we may derive

$$\begin{aligned}
\partial_t\,\mathrm{KL}(\mu_t\|\rho^\mu) &= \int_{\mathcal{X}}\left(\log\frac{\mu_t}{\rho^\mu}\right)\partial_t\mu_t(\mathrm{d}x)\\
&= -\lambda\int_{\mathcal{X}}\nabla_x\log\frac{\mu_t}{\rho^\mu}\cdot\nabla_x\log\frac{\mu_t}{\widehat{\mu}_t}\mu_t(\mathrm{d}x)\\
&= -\lambda\int_{\mathcal{X}}\left\|\nabla_x\log\frac{\mu_t}{\rho^\mu}\right\|_2^2\mu_t(\mathrm{d}x) + \lambda\int_{\mathcal{X}}\nabla_x\log\frac{\mu_t}{\rho^\mu}\cdot\nabla_x\log\frac{\widehat{\mu}_t}{\rho^\mu}\mu_t(\mathrm{d}x)\\
&\le -\frac{\lambda}{2}\int_{\mathcal{X}}\left\|\nabla_x\log\frac{\mu_t}{\rho^\mu}\right\|_2^2\mu_t(\mathrm{d}x) + \frac{\lambda}{2}\int_{\mathcal{X}}\left\|\nabla_x\log\frac{\widehat{\mu}_t}{\rho^\mu}\right\|_2^2\mu_t(\mathrm{d}x)\\
&\le -r_\mu\lambda\cdot\mathrm{KL}(\mu_t\|\rho^\mu) + \frac{M_\mu^2}{2\lambda}.
\end{aligned}$$

The assertion is then proved by Gronwall's inequality. $\qquad\square$

### B.3  PROOF OF THEOREM 3.4

We first introduce two conjugate-type auxiliary functionals and state some properties.

**Lemma B.3.** *Given Lipschitz functions* $\zeta_\mu : \mathcal{X} \to \mathbb{R}$, $\zeta_\nu : \mathcal{Y} \to \mathbb{R}$, *for the pair of probability measures* $\mu \in \mathcal{P}_2(\mathcal{X})$, $\nu \in \mathcal{P}_2(\mathcal{Y})$ *define the time-dependent functional*

$$J_t(\mu,\nu|\zeta^\mu,\zeta^\nu) = -\int_{\mathcal{X}}\zeta^\mu(\mu-\rho^\mu)(\mathrm{d}x) + \int_{\mathcal{Y}}\zeta^\nu(\nu-\rho^\nu)(\mathrm{d}y) - \lambda B_t(\mathrm{KL}(\mu\|\rho^\mu) + \mathrm{KL}(\nu\|\rho^\nu)).$$

*Then the maximum*

$$\widehat{J}_t(\zeta^\mu,\zeta^\nu) = \max_{\mu\in\mathcal{P}_2(\mathcal{X})}\max_{\nu\in\mathcal{P}_2(\mathcal{Y})} J_t(\mu,\nu|\zeta^\mu,\zeta^\nu)$$

*exists for all* $t > 0$ *and is uniquely attained by the pair of probability distributions defined as* $\widehat{\mu}_t(\zeta^\mu) \propto \exp(-(\lambda B_t)^{-1}\zeta^\mu - U^\mu)$ *and* $\widehat{\nu}_t(\zeta^\nu) \propto \exp((\lambda B_t)^{-1}\zeta^\nu - U^\nu)$.

*Proof.* Since $J_t(\mu, \nu | \zeta^\mu, \zeta^\nu)$ decomposes into terms depending only on $\mu$ and $\nu$, respectively, the proof is similar to that of Proposition 2.1. That is, $\mu \mapsto \mathrm{KL}(\mu \| \rho^\mu)$ is lower semi-continuous and strongly convex with respect to the 2-Wasserstein metric by Talagrand's inequality for $\rho^\mu$ so that combined with any linear functional,

$$\underset{\mu \in \mathcal{P}_2(\mathcal{X})}{\arg \max} \, \zeta^\mu(\mu - \rho^\mu)(\mathrm{d}x) - \lambda B_t \cdot \mathrm{KL}(\mu \| \rho^\mu)$$

has a unique maximizer $\widehat{\mu}_t(\zeta^\mu)$ which moreover is given by the stated first-order condition. $\qquad\square$

The following properties are direct extensions of standard conjugacy results in convex analysis, see e.g. Hiriart-Urruty & Lemaréchal (2004), Section E.

**Lemma B.4.** *The functional* $\widehat{J}_t(\zeta^\mu, \zeta^\nu)$ *satisfies the following properties.*

(i) *$\widehat{J}_t$ is nonnegative and convex in both arguments.*

(ii) *$\widehat{J}_t$ admits functional derivatives at any $(\zeta^\mu, \zeta^\nu)$ which are given as*

$$\frac{\delta \widehat{J}_t}{\delta \zeta^\mu}(\zeta^\mu, \zeta^\nu) = -\widehat{\mu}_t(\zeta^\mu) + \rho^\mu, \quad \frac{\delta \widehat{J}_t}{\delta \zeta^\nu}(\zeta^\mu, \zeta^\nu) = \widehat{\nu}_t(\zeta^\nu) - \rho^\nu.$$

(iii) *The derivative with respect to time is bounded as*

$$\partial_t \widehat{J}_t(\zeta^\mu, \zeta^\nu) \le -\lambda \beta_t (\mathrm{KL}(\widehat{\mu}_t(\zeta^\mu) \| \rho^\mu) + \mathrm{KL}(\widehat{\nu}_t(\zeta^\nu) \| \rho^\nu)).$$

*Proof.* (i) Note that $\widehat{J}_t \ge 0$ by taking $\mu = \rho^\mu, \nu = \rho^\nu$, and $\widehat{J}_t$ is convex in both $\zeta^\mu, \zeta^\nu$ as it is a pointwise maximum of affine functionals.

(ii) Due to the explicit dependency of $\widehat{\mu}_t(\zeta^\mu)$ on $\zeta^\mu$, $\widehat{J}_t(\zeta^\mu, \zeta^\nu) = J_t(\widehat{\mu}_t(\zeta^\mu), \widehat{\mu}_t(\zeta^\nu) | \zeta^\mu, \zeta^\nu)$ admits a functional derivative with respect to $\zeta^\mu$ and

$$\frac{\delta \widehat{J}_t}{\delta \zeta^\mu}(\zeta^\mu, \zeta^\nu) = -\widehat{\mu}_t(\zeta^\mu) + \rho^\mu - \int_{\mathcal{X}} \left( \zeta^\mu + \lambda B_t \log \frac{\widehat{\mu}(\zeta^\mu)}{\rho^\mu} \right) \frac{\delta}{\delta \zeta^\mu} \widehat{\mu}_t(\zeta^\mu)(\mathrm{d}x) = -\widehat{\mu}_t(\zeta^\mu) + \rho^\mu.$$

(iii) The time derivative of $\widehat{J}_t$ exists due to the differentiability of $(B_t)_{t \ge 0}$. For any $t' > t$,

$$\begin{aligned}
\widehat{J}_{t'}(\zeta^\mu, \zeta^\nu) &= J_{t'}(\widehat{\mu}_{t'}(\zeta^\mu), \widehat{\nu}_{t'}(\zeta^\nu) | \zeta^\mu, \zeta^\nu) \\
&= J_t(\widehat{\mu}_{t'}(\zeta^\mu), \widehat{\nu}_{t'}(\zeta^\nu) | \zeta^\mu, \zeta^\nu) - \lambda(B_{t'} - B_t)(\mathrm{KL}(\widehat{\mu}_{t'}(\zeta^\mu) \| \rho^\mu) + \mathrm{KL}(\widehat{\nu}_{t'}(\zeta^\nu) \| \rho^\nu)) \\
&\le \widehat{J}_t(\zeta^\mu, \zeta^\nu) - \lambda(B_{t'} - B_t)(\mathrm{KL}(\widehat{\mu}_{t'}(\zeta^\mu) \| \rho^\mu) + \mathrm{KL}(\widehat{\nu}_{t'}(\zeta^\nu) \| \rho^\nu))
\end{aligned}$$

by the maximality of $\widehat{J}_t$, thus taking the limit $t' \downarrow t$ yields the stated inequality. $\qquad\square$

We proceed to the proof of Theorem 3.4. Denote the unnormalized aggregate derivatives as

$$\delta_t^\mu = \int_0^t \beta_s \frac{\delta \mathcal{L}}{\delta \mu}(\mu_s, \nu_s) \, \mathrm{d}s, \quad \delta_t^\nu = \int_0^t \beta_s \frac{\delta \mathcal{L}}{\delta \nu}(\mu_s, \nu_s) \, \mathrm{d}s$$

which are Lipschitz due to Assumption 2. Then by Lemma B.4,

$$\begin{aligned}
&\frac{\mathrm{d}}{\mathrm{d}t} \widehat{J}_t(\delta_t^\mu, \delta_t^\nu) \\
&= \int_{\mathcal{X}} \partial_t \delta_t^\mu \frac{\delta \widehat{J}_t}{\delta \zeta^\mu}(\delta_t^\mu, \delta_t^\nu)(\mathrm{d}x) + \int_{\mathcal{Y}} \partial_t \delta_t^\nu \frac{\delta \widehat{J}_t}{\delta \zeta^\nu}(\delta_t^\mu, \delta_t^\nu)(\mathrm{d}y) + (\partial_t \widehat{J}_t)(\delta_t^\mu, \delta_t^\nu) \\
&\le \beta_t \int_{\mathcal{X}} \frac{\delta \mathcal{L}}{\delta \mu}(\mu_t, \nu_t)(-\widehat{\mu}_t(\delta_t^\mu) + \rho^\mu)(\mathrm{d}x) + \beta_t \int_{\mathcal{Y}} \frac{\delta \mathcal{L}}{\delta \nu}(\mu_t, \nu_t)(\widehat{\nu}_t(\delta_t^\nu) - \rho^\nu)(\mathrm{d}y) \\
&\quad - \lambda \beta_t (\mathrm{KL}(\widehat{\mu}_t(\delta_t^\mu) \| \rho^\mu) + \mathrm{KL}(\widehat{\nu}_t(\delta_t^\nu) \| \rho^\nu)).
\end{aligned}$$

The NI error of the averaged distributions can now be bounded,

$$
\mathrm{NI}(\bar{\mu}_t, \bar{\nu}_t)
$$
$$
= \max_{\mu,\nu} \mathcal{L}_\lambda(\bar{\mu}_t, \nu) - \mathcal{L}_\lambda(\mu, \bar{\nu}_t)
$$
$$
\leq \max_{\mu,\nu} \frac{1}{B_t} \int_0^t \beta_s (\mathcal{L}_\lambda(\mu_s, \nu) - \mathcal{L}_\lambda(\mu, \nu_s)) \, \mathrm{d}s
$$
$$
\leq \max_{\mu,\nu} \frac{1}{B_t} \int_0^t \beta_s \bigg( \int_{\mathcal{Y}} \frac{\delta \mathcal{L}}{\delta \nu}(\mu_s, \nu_s)(\nu - \nu_s)(\mathrm{d}y) - \int_{\mathcal{X}} \frac{\delta \mathcal{L}}{\delta \mu}(\mu_s, \nu_s)(\mu - \mu_s)(\mathrm{d}x)
$$
$$
+ \lambda(\mathrm{KL}(\mu_s\|\rho^\mu) - \mathrm{KL}(\nu\|\rho^\nu) - \mathrm{KL}(\mu\|\rho^\mu) + \mathrm{KL}(\nu_s\|\rho^\nu)) \bigg) \, \mathrm{d}s
$$
$$
= \frac{1}{B_t} \max_{\mu,\nu} \bigg( \int_{\mathcal{Y}} \delta_t^\nu (\nu - \rho^\nu)(\mathrm{d}y) - \int_{\mathcal{X}} \delta_t^\mu (\mu - \rho^\mu)(\mathrm{d}x) - \lambda B_t (\mathrm{KL}(\mu\|\rho^\mu) + \mathrm{KL}(\nu\|\rho^\nu)) \bigg)
$$
$$
+ \frac{1}{B_t} \int_0^t \beta_s \bigg( \int_{\mathcal{Y}} \frac{\delta \mathcal{L}}{\delta \nu}(\mu_s, \nu_s)(\rho^\nu - \nu_s)(\mathrm{d}y) - \int_{\mathcal{X}} \frac{\delta \mathcal{L}}{\delta \mu}(\mu_s, \nu_s)(\rho^\mu - \mu_s)(\mathrm{d}x)
$$
$$
+ \lambda(\mathrm{KL}(\mu_s\|\rho^\mu) + \mathrm{KL}(\nu_s\|\rho^\nu)) \bigg) \, \mathrm{d}s,
$$

where we have used the convex-concavity of $\mathcal{L}_\lambda$ and $\mathcal{L}$ in succession. By extracting the terms corresponding to the auxiliary functional $\widehat{J}_t$, we are able to apply Lemma B.4(iii) and obtain that

$$
\frac{1}{B_t} \bigg[ \widehat{J}_t(\delta_t^\mu, \delta_t^\nu) + \int_0^t \beta_s \bigg( \int_{\mathcal{Y}} \frac{\delta \mathcal{L}}{\delta \nu}(\mu_s, \nu_s)(\rho^\nu - \nu_s)(\mathrm{d}y) - \int_{\mathcal{X}} \frac{\delta \mathcal{L}}{\delta \mu}(\mu_s, \nu_s)(\rho^\mu - \mu_s)(\mathrm{d}x)
$$
$$
+ \lambda(\mathrm{KL}(\mu_s\|\rho^\mu) + \mathrm{KL}(\nu_s\|\rho^\nu)) \bigg) \, \mathrm{d}s \bigg]
$$
$$
\leq \frac{1}{B_t} \bigg[ \int_0^t \bigg( -\lambda \beta_s (\mathrm{KL}(\widehat{\mu}_s(\delta_t^\mu)\|\rho^\mu) + \mathrm{KL}(\widehat{\nu}_s(\delta_t^\nu)\|\rho^\nu))
$$
$$
+ \beta_s \int_{\mathcal{X}} \frac{\delta \mathcal{L}}{\delta \mu}(\mu_s, \nu_s)(-\widehat{\mu}_s(\delta_s^\mu) + \rho^\mu)(\mathrm{d}x) + \beta_s \int_{\mathcal{Y}} \frac{\delta \mathcal{L}}{\delta \nu}(\mu_s, \nu_s)(\widehat{\nu}_s(\delta_s^\nu) - \rho^\nu)(\mathrm{d}y) \bigg) \, \mathrm{d}s
$$
$$
+ \int_0^t \beta_s \bigg( \int_{\mathcal{Y}} \frac{\delta \mathcal{L}}{\delta \nu}(\mu_s, \nu_s)(\rho^\nu - \nu_s)(\mathrm{d}y) - \int_{\mathcal{X}} \frac{\delta \mathcal{L}}{\delta \mu}(\mu_s, \nu_s)(\rho^\mu - \mu_s)(\mathrm{d}x)
$$
$$
+ \lambda(\mathrm{KL}(\mu_s\|\rho^\mu) + \mathrm{KL}(\nu_s\|\rho^\nu)) \bigg) \, \mathrm{d}s \bigg]
$$
$$
= \frac{1}{B_t} \int_0^t \beta_s \bigg( \lambda(\mathrm{KL}(\mu_s\|\rho^\mu) - \mathrm{KL}(\widehat{\mu}_s\|\rho^\mu) + \mathrm{KL}(\nu_s\|\rho^\mu) - \mathrm{KL}(\widehat{\nu}_s\|\rho^\nu))
$$
$$
+ \int_{\mathcal{X}} \frac{\delta \mathcal{L}}{\delta \mu}(\mu_s, \nu_s)(\mu_s - \widehat{\mu}_s)(\mathrm{d}x) - \int_{\mathcal{Y}} \frac{\delta \mathcal{L}}{\delta \nu}(\mu_s, \nu_s)(\nu_s - \widehat{\nu}_s)(\mathrm{d}y) \bigg) \, \mathrm{d}s
$$
$$
= \frac{1}{B_t} \int_0^t \beta_s \bigg( \lambda \int_{\mathcal{X}} \log \frac{\widehat{\mu}_s}{\rho^\mu}(\mu_s - \widehat{\mu}_s)(\mathrm{d}x) + \lambda \int_{\mathcal{X}} \log \frac{\mu_s}{\widehat{\mu}_s} \mu_s(\mathrm{d}x)
$$
$$
+ \lambda \int_{\mathcal{Y}} \log \frac{\widehat{\nu}_s}{\rho^\nu}(\nu_s - \widehat{\nu}_s)(\mathrm{d}y) + \lambda \int_{\mathcal{Y}} \log \frac{\nu_s}{\widehat{\nu}_s} \nu_s(\mathrm{d}y)
$$
$$
+ \int_{\mathcal{X}} \frac{\delta \mathcal{L}}{\delta \mu}(\mu_s, \nu_s)(\mu_s - \widehat{\mu}_s)(\mathrm{d}x) - \int_{\mathcal{Y}} \frac{\delta \mathcal{L}}{\delta \nu}(\mu_s, \nu_s)(\nu_s - \widehat{\nu}_s)(\mathrm{d}y) \bigg) \, \mathrm{d}s
$$
$$
= \frac{1}{B_t} \int_0^t \beta_s \bigg[ \int_{\mathcal{X}} \bigg( \frac{\delta \mathcal{L}}{\delta \mu}(\mu_s, \nu_s) - \frac{1}{B_s} \int_0^s \beta_r \frac{\delta \mathcal{L}}{\delta \mu}(\mu_r, \nu_r) \, \mathrm{d}r \bigg)(\mu_s - \widehat{\mu}_s)(\mathrm{d}x)
$$
$$
- \int_{\mathcal{Y}} \bigg( \frac{\delta \mathcal{L}}{\delta \nu}(\mu_s, \nu_s) - \frac{1}{B_s} \int_0^s \beta_r \frac{\delta \mathcal{L}}{\delta \nu}(\mu_r, \nu_r) \, \mathrm{d}r \bigg)(\nu_s - \widehat{\nu}_s)(\mathrm{d}y)
$$
$$
+ \lambda \int_{\mathcal{X}} \log \frac{\mu_s}{\widehat{\mu}_s} \mu_s(\mathrm{d}x) + \lambda \int_{\mathcal{Y}} \log \frac{\nu_s}{\widehat{\nu}_s} \nu_s(\mathrm{d}y) \bigg] \, \mathrm{d}s.
$$

By Proposition 3.3 and Talagrand's inequality, we can therefore bound

$$\mathrm{NI}(\bar{\mu}_t, \bar{\nu}_t)$$

$$\leq \frac{1}{B_t} \int_0^t \beta_s (2M_\mu W_1(\mu_s, \widehat{\mu}_s) + 2M_\nu W_1(\nu_s, \widehat{\nu}_s) + \lambda \,\mathrm{KL}(\mu_s \| \widehat{\mu}_s) + \lambda \,\mathrm{KL}(\nu_s \| \widehat{\nu}_s)) \, \mathrm{d}s$$

$$\leq \frac{2}{B_t} \int_0^t \beta_s \left( M_\mu \sqrt{\frac{2}{\alpha_\mu} \,\mathrm{KL}(\mu_s \| \widehat{\mu}_s)} + M_\nu \sqrt{\frac{2}{\alpha_\mu} \,\mathrm{KL}(\nu_s \| \widehat{\nu}_s)} \right) \mathrm{d}s$$

$$\quad + \frac{\lambda}{B_t} \int_0^t \beta_s (\mathrm{KL}(\mu_s \| \widehat{\mu}_s) + \mathrm{KL}(\nu_s \| \widehat{\nu}_s)) \, \mathrm{d}s$$

$$\leq \left( \frac{M_\mu^2}{\alpha_\mu^2} + \frac{M_\nu^2}{\alpha_\nu^2} \right) \frac{4(r+1)}{\lambda^2 B_t} \int_{t_0}^t \frac{\beta_s}{s} \left( 1 + O(s^{-1}) \right) \mathrm{d}s.$$

In particular, for $\beta_t = t^r$ with $r > 0$, we obtain the convergence rate

$$\mathrm{NI}(\bar{\mu}_t, \bar{\nu}_t) \leq \left( \frac{M_\mu^2}{\alpha_\mu^2} + \frac{M_\nu^2}{\alpha_\nu^2} \right) \frac{4(r+1)^2}{r \lambda^2 t} + O(t^{-2})$$

whose leading term is optimized when $r = 1$. For $\beta_t = 1$, we obtain the slightly slower rate

$$\mathrm{NI}(\bar{\mu}_t, \bar{\nu}_t) \leq \left( \frac{M_\mu^2}{\alpha_\mu^2} + \frac{M_\nu^2}{\alpha_\nu^2} \right) \frac{4 \log t}{\lambda^2 t} + O(t^{-1}).$$

We remark that for decreasing $\beta_t$, the integral tends to converge so that the normalizing $B_t^{-1}$ term dominates, leading to significantly slower convergence. For example, if $\beta_t \sim t^r$ for $-1 < r < 0$ the rate is $O(t^{-1-r})$; if $\beta_t \sim t^{-1}$, the rate is $O(\frac{1}{\log t})$. □

## C  TIME AND SPACE DISCRETIZATION

### C.1  GRADIENT STOPPED PROCESS

Denote $\mathscr{X}_k = (X_k^i)_{i=1}^N$, $\mathscr{Y}_k = (Y_k^i)_{i=1}^N$ and $\mu_{\mathscr{X}_k} = \frac{1}{N} \sum_{i=1}^N \delta_{X_k^i}$, $\nu_{\mathscr{Y}_k} = \frac{1}{N} \sum_{i=1}^N \delta_{Y_k^i}$. That is, the subscript $k$ denotes the number of steps while superscript $i$ denotes the $i$th particle. We also write $(\mathscr{X}, \mathscr{Y})_{1:k} := (\mathscr{X}_{1:k}, \mathscr{Y}_{1:k})$ for notational simplicity.

We analyze the following MFL-AG $N$-particle update for all $i = 1, \cdots, N$,

$$X_{k+1}^i = X_k^i - \frac{\eta}{B_k} \sum_{j=1}^k \beta_j \nabla_x \frac{\delta \mathcal{L}}{\delta \mu}(\mu_{\mathscr{X}_j}, \nu_{\mathscr{Y}_j})(X_k^i) - \lambda \eta \nabla_x U^\mu(X_k^i) + \sqrt{2\lambda \eta} \xi_k^{\mu, i},$$

$$Y_{k+1}^i = Y_k^i + \frac{\eta}{B_k} \sum_{j=1}^k \beta_j \nabla_y \frac{\delta \mathcal{L}}{\delta \nu}(\mu_{\mathscr{X}_j}, \nu_{\mathscr{Y}_j})(Y_k^i) - \lambda \eta \nabla_y U^\nu(Y_k^i) + \sqrt{2\lambda \eta} \xi_k^{\nu, i},$$

$$(7)$$

where $\xi_k^{\mu, i}, \xi_k^{\nu, i}$ are i.i.d. standard Gaussian and the initial values $\mathscr{X}_1, \mathscr{Y}_1$ are sampled from initial distributions $\mu_0 \in \mathcal{P}_2(\mathcal{X})$, $\nu_0 \in \mathcal{P}_2(\mathcal{Y})$. We write the history-dependent averaged drift function as

$$\mathfrak{b}_k^\mu = \mathfrak{b}_k^\mu(\cdot | (\mathscr{X}, \mathscr{Y})_{1:k}) = -\frac{1}{B_k} \sum_{j=1}^k \beta_j \nabla_x \frac{\delta \mathcal{L}}{\delta \mu}(\mu_{\mathscr{X}_j}, \nu_{\mathscr{Y}_j}) - \lambda \nabla_x U^\mu$$

and similarly for $\mathfrak{b}_k^\nu$. The history-dependent $N$-particle proximal distributions are defined on the configuration spaces $\mathcal{X}^N, \mathcal{Y}^N$ as the product distributions

$$\widehat{\mu}_k^{(N)}(\mathscr{X}) \propto \rho^{\mu \otimes N}(\mathscr{X}) \exp \left( -\frac{N}{\lambda B_k} \int_{\mathcal{X}} \sum_{j=1}^k \beta_j \frac{\delta \mathcal{L}}{\delta \mu}(\mu_{\mathscr{X}_j}, \nu_{\mathscr{Y}_j}) \mu_{\mathscr{X}}(\mathrm{d}x) \right),$$

$$\widehat{\nu}_k^{(N)}(\mathscr{Y}) \propto \rho^{\nu \otimes N}(\mathscr{Y}) \exp \left( \frac{N}{\lambda B_k} \int_{\mathcal{Y}} \sum_{j=1}^k \beta_j \frac{\delta \mathcal{L}}{\delta \nu}(\mu_{\mathscr{X}_j}, \nu_{\mathscr{Y}_j}) \nu_{\mathscr{Y}}(\mathrm{d}y) \right).$$

We substitute $\beta_k = k^r$ with $r \in \mathbb{R}_{\geq 0}$ whenever necessary to simplify the calculations, although similar results may be derived for any well-behaved sequence of weights.

The following lemma quantifies the sequential evolution of the averaged drift.

**Lemma C.1.** *For any pair of integers $k > \ell$ we have $\|\mathfrak{b}_k^\mu - \mathfrak{b}_\ell^\mu\|_\infty \leq 2\left(1 - \frac{B_\ell}{B_k}\right) M_\mu$.*

*Proof.* For any $x \in \mathcal{X}$,

$$
\begin{aligned}
\|\mathfrak{b}_k^\mu(x) - \mathfrak{b}_\ell^\mu(x)\| &= \left\| -\frac{1}{B_k} \sum_{j=1}^{k} \beta_j \nabla_x \frac{\delta \mathcal{L}}{\delta \mu}(\mu_{\mathscr{X}_j}, \nu_{\mathscr{Y}_j})(x) + \frac{1}{B_\ell} \sum_{j=1}^{\ell} \beta_j \nabla_x \frac{\delta \mathcal{L}}{\delta \mu}(\mu_{\mathscr{X}_j}, \nu_{\mathscr{Y}_j})(x) \right\| \\
&= \left\| \frac{B_k - B_\ell}{B_\ell B_k} \sum_{j=1}^{\ell} \beta_j \nabla_x \frac{\delta \mathcal{L}}{\delta \mu}(\mu_{\mathscr{X}_j}, \nu_{\mathscr{Y}_j})(x) - \frac{1}{B_k} \sum_{j=\ell+1}^{k} \beta_j \nabla_x \frac{\delta \mathcal{L}}{\delta \mu}(\mu_{\mathscr{X}_j}, \nu_{\mathscr{Y}_j})(x) \right\| \\
&\leq 2\left(1 - \frac{B_\ell}{B_k}\right) M_\mu,
\end{aligned}
$$

yielding the assertion. $\qquad\square$

*The gradient-stopped process.* For given integers $k > \ell$, consider the following synchronous modification of the MFL-AG update with the drift stopped at time $k - \ell$,

$$
\widetilde{X}_{j+1}^i = \widetilde{X}_j^i + \eta\, \mathfrak{b}_{j \wedge (k-\ell)}^\mu(\widetilde{X}_j^i) + \sqrt{2\lambda\eta}\,\xi_j^{\mu,i}, \quad \widetilde{Y}_{j+1}^i = \widetilde{Y}_j^i + \eta\, \mathfrak{b}_{j \wedge (k-\ell)}^\nu(\widetilde{Y}_j^i) + \sqrt{2\lambda\eta}\,\xi_j^{\nu,i}.
$$

The initializations $\widetilde{\mathscr{X}_1}, \widetilde{\mathscr{Y}_1}$ and the random vectors $\xi_j^{\mu,i}, \xi_j^{\nu,i}$ are to be shared with the original process so that $(\widetilde{\mathscr{X}}, \widetilde{\mathscr{Y}})_{1:k-\ell+1} = (\mathscr{X}, \mathscr{Y})_{1:k-\ell+1}$. We will study this process alongside the original in order to facilitate short-term perturbation analyses.

**Lemma C.2.** *If $\eta \leq \frac{r_\mu}{4\lambda R_\mu^2}$, the second moments of the particles $X_k^i$ and $\widetilde{X}_k^i$ are uniformly bounded for all $k \geq 1$ as*

$$
\mathbb{E}[\|X_k^i\|^2],\ \mathbb{E}[\|\widetilde{X}_k^i\|^2] \leq \mathbb{E}[\|X_1^i\|^2] + \mathfrak{s}^\mu, \quad \mathfrak{s}^\mu := \frac{2}{r_\mu}\left(\frac{M_\mu^2}{r_\mu \lambda^2} + \lambda\eta M_\mu^2 + d_\mathcal{X}\right).
$$

*Proof.* From the update rule (7),

$$
\begin{aligned}
\mathbb{E}[\|X_{k+1}^i\|^2] \\
= \mathbb{E}[\|X_k^i\|^2] - 2\eta\left\langle X_k^i, \frac{1}{B_k}\sum_{j=1}^{k}\beta_j \nabla_x \frac{\delta\mathcal{L}}{\delta\mu}(\mu_{\mathscr{X}_j},\nu_{\mathscr{Y}_j})(X_k^i) + \lambda\nabla_x U^\mu(X_k^i)\right\rangle \\
+ \eta^2\left\|\frac{1}{B_k}\sum_{j=1}^{k}\beta_j \nabla_x \frac{\delta\mathcal{L}}{\delta\mu}(\mu_{\mathscr{X}_j},\nu_{\mathscr{Y}_j})(X_k^i) + \lambda\nabla_x U^\mu(X_k^i)\right\|^2 + 2\lambda\eta d_\mathcal{X} \\
\leq \mathbb{E}[\|X_k^i\|^2] + 2\eta M_\mu \mathbb{E}[\|X_k^i\|] - 2\lambda\eta r_\mu \mathbb{E}[\|X_k^i\|^2] \\
+ 2\lambda^2\eta^2 M_\mu^2 + 2\lambda^2\eta^2 R_\mu^2 \mathbb{E}[\|X_k^i\|^2] + 2\lambda\eta d_\mathcal{X} \\
\leq (1 - \lambda\eta r_\mu)\mathbb{E}[\|X_k^i\|^2] + \frac{2\eta M_\mu^2}{r_\mu\lambda} + 2\lambda^2\eta^2 M_\mu^2 + 2\lambda\eta d_\mathcal{X},
\end{aligned}
$$

where we have used $\mathbb{E}[\|X_k^i\|] \leq \frac{r_\mu\lambda}{4M_\mu}\mathbb{E}[\|X_k^i\|^2] + \frac{M_\mu}{r_\mu\lambda}$ and $\eta \leq \frac{r_\mu}{4\lambda R_\mu^2}$. The statement now follows from induction. The same logic can be applied to $\mathbb{E}[\|\widetilde{X}_k^i\|^2]$. $\qquad\square$

**Lemma C.3.** *If $\eta \leq \frac{r_\mu\lambda}{2(L_\mu + \lambda R_\mu)^2}$, the Wasserstein error between the original and gradient-stopped process at time $k > \ell$ is bounded as*

$$
W_2(\mu_{\mathscr{X}_k}, \mu_{\widetilde{\mathscr{X}_k}}) \leq \frac{r+1}{k-\ell+1}\mathfrak{w}_\ell^\mu,
$$

*where*

$$(\mathfrak{w}_\ell^\mu)^2 := \left(2\eta + \frac{1}{r_\mu\lambda} \vee \frac{1}{2L_\mu}\right)\frac{M_\mu^2(1+2\eta L_\mu)^2((1+2\eta L_\mu)^\ell - 1)}{\eta^2 L_\mu^3}.$$

*Proof.* Decomposing the difference at each step $j > k - \ell$ as

$$X_{j+1}^i - \widetilde{X}_{j+1}^i = X_j^i - \widetilde{X}_j^i + \eta(\mathfrak{b}_j^\mu(X_j^i) - \mathfrak{b}_j^\mu(\widetilde{X}_j^i)) + \eta(\mathfrak{b}_j^\mu(\widetilde{X}_j^i) - \mathfrak{b}_{k-\ell}^\mu(\widetilde{X}_j^i)),$$

we expand to obtain

$$\|X_{j+1}^i - \widetilde{X}_{j+1}^i\|^2$$

$$\leq \|X_j^i - \widetilde{X}_j^i\|^2 + 2\eta\langle X_j^i - \widetilde{X}_j^i, \mathfrak{b}_j^\mu(X_j^i) - \mathfrak{b}_j^\mu(\widetilde{X}_j^i)\rangle + 2\eta\|X_j^i - \widetilde{X}_j^i\| \cdot \|\mathfrak{b}_j^\mu - \mathfrak{b}_{k-\ell}^\mu\|_\infty$$

$$+ 2\eta^2\|\mathfrak{b}_j^\mu(X_j^i) - \mathfrak{b}_j^\mu(\widetilde{X}_j^i)\|^2 + 2\eta^2\|\mathfrak{b}_j^\mu - \mathfrak{b}_{k-\ell}^\mu\|_\infty^2$$

$$\leq \|X_j^i - \widetilde{X}_j^i\|^2 + 2\eta(L_\mu - \lambda r_\mu)\|X_j^i - \widetilde{X}_j^i\|^2 + 4\eta\left(1 - \frac{B_{k-\ell}}{B_j}\right)M_\mu\|X_j^i - \widetilde{X}_j^i\|$$

$$+ 2\eta^2(L_\mu + \lambda R_\mu)^2\|X_j^i - \widetilde{X}_j^i\|^2 + 8\eta^2\left(1 - \frac{B_{k-\ell}}{B_j}\right)^2 M_\mu^2$$

$$\leq (1 + 2\eta L_\mu)\|X_j^i - \widetilde{X}_j^i\|^2 + \left(\frac{4\eta M_\mu^2}{r_\mu\lambda} + 8\eta^2 M_\mu^2\right)\left(1 - \frac{B_{k-\ell}}{B_j}\right)^2.$$

Starting from $X_{k-\ell}^i - \widetilde{X}_{k-\ell}^i = 0$ and iterating,

$$\|X_k^i - \widetilde{X}_k^i\|^2 \leq \left(\frac{4\eta M_\mu^2}{r_\mu\lambda} + 8\eta^2 M_\mu^2\right)\sum_{j=k-\ell+1}^{k-1}(1+2\eta L_\mu)^{k-j-1}\left(1 - \frac{B_{k-\ell}}{B_j}\right)^2, \quad k \geq \ell + 2. \tag{8}$$

Now noting that with $\beta_j = j^r$

$$1 - \frac{B_{k-\ell}}{B_j} \leq \frac{(j - k + \ell)j^r}{\int_0^j z^r\, dz} = (r+1)\left(1 - \frac{k-\ell}{j}\right) \leq \frac{(r+1)(j-k+\ell)}{k-\ell+1},$$

setting $\theta = (1 + 2\eta L_\mu)^{-1}$ we can explicitly compute

$$\sum_{j=k-\ell+1}^{k-1}(j - k + \ell)^2(1+2\eta L_\mu)^{k-j-1} = \theta^{1-\ell}\sum_{j=1}^{\ell-1}j^2\theta^j$$

$$= \frac{\theta^{2-\ell}}{(1-\theta)^3}(-(\ell-1)^2\theta^{\ell+1} + (2\ell^2 - 2\ell - 1)\theta^\ell - \ell^2\theta^{\ell-1} + 3 - \theta)$$

$$\leq \frac{\theta}{(1-\theta)^3}\left(\frac{3-\theta}{\theta^{\ell-1}} - 2\right) \leq \frac{2\theta}{(1-\theta)^3}(\theta^{-\ell} - 1).$$

Plugging back into (8) gives

$$\|X_k^i - \widetilde{X}_k^i\|^2 \leq \left(\frac{4M_\mu^2}{r_\mu\lambda} + 8\eta M_\mu^2\right)\frac{(r+1)^2(1+2\eta L_\mu)^2}{4\eta^2 L_\mu^3}\frac{(1+2\eta L_\mu)^\ell - 1}{(k-\ell+1)^2}$$

$$\leq (r+1)^2 M_\mu^2\left(2\eta + \frac{1}{r_\mu\lambda} \vee \frac{1}{2L_\mu}\right)\frac{(1+2\eta L_\mu)^2}{\eta^2 L_\mu^3}\frac{(1+2\eta L_\mu)^\ell - 1}{(k-\ell+1)^2}$$

uniformly for all $i \in [N]$. Note that the $(2L_\mu)^{-1}$ term is added to simplify later analyses and is generally vacuous. Finally, taking $W_2^2(\mu_{\mathscr{X}_k}, \mu_{\widetilde{\mathscr{X}_k}}) \leq \frac{1}{N}\|\mathscr{X}_k - \widetilde{\mathscr{X}_k}\|^2$ yields the desired bound. $\square$

The calculations for the two above two results are similar but the bounds are fundamentally different. In Lemma C.2 we rely on the long-distance dissipative nature of $\mathfrak{b}_k^\mu$ to prove a uniform-in-time guarantee, while in Lemma C.3 we forego the contraction to isolate the $1 - \frac{B_\ell}{B_k}$ factor and obtain tight short-term error bounds.

The leave-one-out error of the modified process can also be characterized as follows. We remark that the arguments in Lemmas C.2 and C.4 are identical to that in Suzuki et al. (2023).

**Lemma C.4.** *Denote the set of $N-1$ particles $(\widetilde{X}_k^1, \cdots, \widetilde{X}_k^{i-1}, \widetilde{X}_k^{i+1}, \cdots, \widetilde{X}_k^N)$ as $\mathscr{X}_k^{-i}$. If $\eta \leq \frac{r_\mu}{4\lambda R_\mu^2}$, the $W_2$ distance between $\mu_{\widetilde{\mathscr{X}}_k}$ and $\mu_{\widetilde{\mathscr{X}}_k^{-i}}$ at time $k > \ell$ can be bounded on average as*

$$\mathbb{E}_{\widetilde{\mathscr{X}}_k|(\mathscr{X},\mathscr{Y})_{1:k-\ell}}\left[W_2^2(\mu_{\widetilde{\mathscr{X}}_k}, \mu_{\widetilde{\mathscr{X}}_k^{-i}})\right] \leq \frac{4\mathfrak{s}^\mu}{N} + \frac{2}{N(N-1)}\sum_{j\neq i}\|X_{k-\ell}^j\|^2 + \frac{2}{N}\|X_{k-\ell}^j\|^2.$$

*Proof.* Similarly to Lemma C.2 but starting from time $k-\ell$, it can be shown that

$$\mathbb{E}_{\widetilde{\mathscr{X}}_k|(\mathscr{X},\mathscr{Y})_{1:k-\ell}}[\|\widetilde{X}_k^j\|^2] \leq \|X_{k-\ell}^j\|^2 \vee \mathfrak{s}^\mu, \quad j \in [N],$$

which will be useful in the sequel. Then taking the coupling $\sum_{j\neq i}\frac{1}{N}\delta_{(\widetilde{X}_k^j, \widetilde{X}_k^j)} + \frac{1}{N(N-1)}\delta_{(\widetilde{X}_k^i, \widetilde{X}_k^j)}$ for $\mu_{\widetilde{\mathscr{X}}_k}, \mu_{\widetilde{\mathscr{X}}_k^{-i}}$ gives

$$\mathbb{E}_{\widetilde{\mathscr{X}}_k|(\mathscr{X},\mathscr{Y})_{1:k-\ell}}\left[W_2^2(\mu_{\widetilde{\mathscr{X}}_k}, \mu_{\widetilde{\mathscr{X}}_k^{-i}})\right] \leq \mathbb{E}_{\widetilde{\mathscr{X}}_k|(\mathscr{X},\mathscr{Y})_{1:k-\ell}}\left[\frac{1}{N(N-1)}\sum_{j\neq i}\|\widetilde{X}_k^j - \widetilde{X}_k^i\|^2\right]$$

$$\leq \mathbb{E}_{\widetilde{\mathscr{X}}_k|(\mathscr{X},\mathscr{Y})_{1:k-\ell}}\left[\frac{2}{N(N-1)}\sum_{j\neq i}\|\widetilde{X}_k^j\|^2 + \frac{2}{N}\|\widetilde{X}_k^i\|^2\right]$$

$$\leq \frac{4\mathfrak{s}^\mu}{N} + \frac{2}{N(N-1)}\sum_{j\neq i}\|X_{k-\ell}^j\|^2 + \frac{2}{N}\|X_{k-\ell}^j\|^2.$$

The same bound holds for the original process. $\qquad\square$

### C.2 PROXIMAL PUSHFORWARD BOUNDS

For a measure $\mu^{(N)}$ on $\mathcal{X}^{(N)}$, denote by $\Pi$ the average of the pushforward operators $\Pi_\sharp^i$ along the projections $\mathscr{X} \mapsto X^i$ with the defining property

$$\int_{\mathcal{X}} f(x)\Pi\mu^{(N)}(\mathrm{d}x) = \int_{\mathcal{X}^N}\Pi^* f(\mathscr{X})\mu^{(N)}(\mathrm{d}\mathscr{X}) = \int_{\mathcal{X}^N}\frac{1}{N}\sum_{i=1}^N f(X^i)\mu^{(N)}(\mathrm{d}\mathscr{X})$$

for any integrable function $f : \mathcal{X} \to \mathbb{R}$. We immediately see that

$$\Pi\widehat{\mu}_k^{(N)} = \rho^\mu \exp\left(-\frac{1}{\lambda B_k}\int_{\mathcal{X}}\sum_{j=1}^k \beta_j \frac{\delta\mathcal{L}}{\delta\mu}(\mu_{\mathscr{X}_j}, \nu_{\mathscr{Y}_j})\right)$$

is the stationary distribution of the continuous-time Itô diffusion $\mathrm{d}Z_t = \mathfrak{b}_k^\mu(Z_t)\,\mathrm{d}t + \sqrt{2\lambda}\,\mathrm{d}W_t^\mu$, which entails the following uniform moment bound.

**Lemma C.5.** *The unnormalized second moment $\int_{\mathcal{X}}\|x\|^2\,\Pi\widehat{\mu}_k^{(N)}(\mathrm{d}x)$ is bounded above for any integer $k$ by $\mathfrak{q}^\mu := r_\mu^{-2}\lambda^{-2}M_\mu^2 + 2r_\mu^{-1}d_\mathcal{X}$.*

We also denote $\mathfrak{p}^\mu := \frac{1}{N}\sum_{i=1}^N \mathbb{E}[\|X_1^i\|^2] < \infty$.

*Proof.* We may compute for the initialization $Z_0 = 0$,

$$\frac{\mathrm{d}}{\mathrm{d}t}\mathbb{E}[\|Z_t\|^2] = 2\mathbb{E}\left[\langle Z_t, \mathfrak{b}_k^\mu(Z_t)\rangle\right] + 2\lambda d_\mathcal{X}$$

$$\leq 2M_\mu\mathbb{E}[\|Z_t\|] - 2r_\mu\lambda\mathbb{E}[\|Z_t\|^2] + 2\lambda d_\mathcal{X}$$

$$\leq -r_\mu\lambda\mathbb{E}[\|Z_t\|^2] + \frac{M_\mu^2}{r_\mu\lambda} + 2\lambda d_\mathcal{X},$$

which yields the bound in the infinite-time limit by Gronwall's lemma. $\qquad\square$

In particular, $\Pi\,\widehat{\mu}_{k-\ell}^{(N)}$ is the approximate stationary distribution of each independent particle of the gradient stopped process after time $k-\ell$ and enjoys an exponential convergence guarantee up to an $O(\eta)$ discretization error term.

**Proposition C.6.** *Assuming $\eta \le \frac{r_\mu}{4\lambda R_\mu^2}$, the KL gap from $\tilde{\mu}_k^i = \mathrm{Law}(\widetilde{X}_k^i|(\mathscr{X},\mathscr{Y})_{1:k-\ell})$ to $\Pi\,\widehat{\mu}_{k-\ell}^{(N)}$ of the gradient stopped process satisfies*

$$\mathrm{KL}(\tilde{\mu}_k^i\|\Pi\,\widehat{\mu}_{k-\ell}^{(N)}) \le \left(1 + \frac{3\exp(-(\ell-1)\alpha_\mu\lambda\eta)}{2\eta^2(L_\mu+\lambda R_\mu)^2}\right)(\mathfrak{K}^\mu\|X_{k-\ell}^i\|^2 + \mathfrak{L}^\mu),$$

*where*

$$\mathfrak{K}^\mu := \frac{\eta^2 R_\mu^2(L_\mu+\lambda R_\mu)^2}{\alpha_\mu}, \quad \mathfrak{L}^\mu := \frac{\eta(L_\mu+\lambda R_\mu)^2}{\alpha_\mu\lambda^2}\left(\eta M_\mu^2 + \lambda^2\eta R_\mu^2\mathfrak{s}^\mu + \lambda d_\mathcal{X}\right)$$

*are both of order $O(\eta)$.*

Hence, choosing

$$\ell = \ell^\mu := \frac{1}{\alpha_\mu\lambda\eta}\left\lceil \log\frac{3}{2\eta^2(L_\mu+\lambda R_\mu)^2}\right\rceil + 1 \tag{9}$$

guarantees that

$$W_2(\tilde{\mu}_k^i\|\Pi\,\widehat{\mu}_{k-\ell}^{(N)}) \le \sqrt{\frac{4}{\alpha_\mu}(\mathfrak{K}^\mu\|X_{k-\ell}^i\|^2 + \mathfrak{L}^\mu)}$$

for any integer $k > \ell$.

*Proof.* We emulate the one-step analysis in Nitanda et al. (2022a) whilst keeping the history $(\mathscr{X},\mathscr{Y})_{1:k-\ell}$ fixed; this dependence is omitted here for notational clarity. For $j \ge k-\ell$, denote by $\mu_t^\dagger$ the law of the process

$$\mathrm{d}X_t^\dagger = \mathfrak{b}_{k-\ell}^\mu(\widetilde{X}_j^i)\,\mathrm{d}t + \sqrt{2\lambda}\,\mathrm{d}W_t^\dagger, \quad 0 \le t \le \eta$$

with $X_0^\dagger = \widetilde{X}_j^i$ so that $X_\eta^\dagger \overset{d}{=} \widetilde{X}_{j+1}^i$. We overload notation and denote both conditional and joint distributions involving $X_t^\dagger$ by $\mu_t^\dagger$. The evolution of $\mu_t^\dagger$ is governed by the conditional Fokker-Planck equation

$$\partial_t\mu_t^\dagger(X_t^\dagger|\widetilde{X}_j^i) = -\nabla_x\cdot\left(\mu_t^\dagger(X_t^\dagger|\widetilde{X}_j^i)\,\mathfrak{b}_{k-\ell}^\mu(\widetilde{X}_j^i)\right) + \lambda\Delta_x\mu_t^\dagger(X_t^\dagger|\widetilde{X}_j^i).$$

Integrating out $\widetilde{X}_j^i$,

$$\begin{aligned}
\partial_t\mu_t^\dagger(X_t^\dagger) &= \int_\mathcal{X} -\nabla_x\cdot\left(\mu_t^\dagger(X_t^\dagger,\widetilde{X}_j^i)\,\mathfrak{b}_{k-\ell}^\mu(\widetilde{X}_j^i)\right)(\mathrm{d}\widetilde{X}_j^i) + \lambda\Delta_x\mu_t^\dagger(X_t^\dagger) \\
&= \nabla_x\cdot\left(\mu_t^\dagger(X_t^\dagger)\left(-\mathbb{E}_{\widetilde{X}_j^i|X_t^\dagger}\left[\mathfrak{b}_{k-\ell}^\mu(\widetilde{X}_j^i)\right] + \lambda\nabla_x\log\mu_t^\dagger(X_t^\dagger)\right)\right) \\
&= \lambda\nabla_x\cdot\left(\mu_t^\dagger(X_t^\dagger)\nabla_x\log\frac{\mu_t^\dagger}{\Pi\,\widehat{\mu}_{k-\ell}^{(N)}}(X_t^\dagger)\right) \\
&\quad + \nabla_x\cdot\left(\mu_t^\dagger(X_t^\dagger)\left(\mathfrak{b}_{k-\ell}^\mu(X_t^\dagger) - \mathbb{E}_{\widetilde{X}_j^i|X_t^\dagger}\left[\mathfrak{b}_{k-\ell}^\mu(\widetilde{X}_j^i)\right]\right)\right).
\end{aligned}$$

Hence the proximal KL gap from $\mu_t^\dagger$ to $\Pi\,\widehat{\mu}_{k-\ell}^{(N)}$ satisfies

$$\begin{aligned}
\partial_t\,\mathrm{KL}(\mu_t^\dagger\|\Pi\,\widehat{\mu}_{k-\ell}^{(N)}) &= \int_\mathcal{X}\log\frac{\mu_t^\dagger}{\Pi\,\widehat{\mu}_{k-\ell}^{(N)}}(\partial_t\mu_t^\dagger)(\mathrm{d}X_t^\dagger) \\
&= -\lambda\int_\mathcal{X}\left\|\nabla_x\log\frac{\mu_t^\dagger}{\Pi\,\widehat{\mu}_{k-\ell}^{(N)}}\right\|^2\mu_t^\dagger(\mathrm{d}X_t^\dagger) \\
&\quad - \iint_{\mathcal{X}\times\mathcal{X}}\log\frac{\mu_t^\dagger}{\Pi\,\widehat{\mu}_{k-\ell}^{(N)}}\cdot\left(\mathfrak{b}_{k-\ell}^\mu(X_t^\dagger) - \mathfrak{b}_{k-\ell}^\mu(\widetilde{X}_j^i)\right)\mu_t^\dagger(\mathrm{d}X_t^\dagger\,\mathrm{d}\widetilde{X}_j^i)
\end{aligned}$$

$$\leq -\frac{\lambda}{2}\int_{\mathcal{X}}\left\|\nabla_x\log\frac{\mu_t^\dagger}{\Pi\,\widehat{\mu}_{k-\ell}^{(N)}}\right\|^2\mu_t^\dagger(\mathrm{d}X_t^\dagger)+\frac{(L_\mu+\lambda R_\mu)^2}{2\lambda}\iint_{\mathcal{X}\times\mathcal{X}}\|X_t^\dagger-\widetilde{X}_j^i\|^2\mu_t^\dagger(\mathrm{d}X_t^\dagger\,\mathrm{d}\widetilde{X}_j^i)$$

$$\leq -\alpha_\mu\lambda\cdot\mathrm{KL}(\mu_t^\dagger\|\Pi\,\widehat{\mu}_{k-\ell}^{(N)})+\frac{(L_\mu+\lambda R_\mu)^2}{2\lambda}\int_{\mathcal{X}}\mathbb{E}_{\xi^\dagger}\left[\left\|\mathfrak{b}_{k-\ell}^\mu(\widetilde{X}_j^i)t+\sqrt{2\lambda t}\xi^\dagger\right\|^2\right]\tilde{\mu}_j^i(\mathrm{d}\widetilde{X}_j^i)$$

where $\xi^\dagger\sim\mathcal{N}(0,\mathrm{I}_{d_{\mathcal{X}}})$ and we have used the LSI for $\Pi\,\widehat{\mu}_{k-\ell}^{(N)}$. The second term is further bounded as

$$\mathbb{E}_{\xi^\dagger}\left[\left\|\mathfrak{b}_{k-\ell}^\mu(\widetilde{X}_j^i)t+\sqrt{2\lambda t}\xi^\dagger\right\|^2\right]\leq\eta^2\,\mathbb{E}_{\widetilde{X}_j^i|X_{1:k-\ell}}\left[\left\|\mathfrak{b}_{k-\ell}^\mu(\widetilde{X}_j^i)\right\|^2\right]+2\lambda\eta d_{\mathcal{X}}$$

$$\leq 2\eta^2 M_\mu^2+2\lambda^2\eta^2 R_\mu^2\,\mathbb{E}[\|\widetilde{X}_j^i\|^2]+2\lambda\eta d_{\mathcal{X}}$$

$$\leq 2\eta^2 M_\mu^2+2\lambda^2\eta^2 R_\mu^2\left(\|X_{k-\ell}^i\|^2\vee\mathfrak{s}^\mu\right)+2\lambda\eta d_{\mathcal{X}}$$

by the proof of Lemma C.4. Gronwall's lemma now leads to

$$\mathrm{KL}(\tilde{\mu}_{j+1}^i\|\Pi\,\widehat{\mu}_{k-\ell}^{(N)})-(\mathfrak{K}^\mu\|X_{k-\ell}^i\|^2+\mathfrak{L}^\mu)\leq e^{-\alpha_\mu\lambda\eta}\left(\mathrm{KL}(\tilde{\mu}_j^i\|\Pi\,\widehat{\mu}_{k-\ell}^{(N)})-(\mathfrak{K}^\mu\|X_{k-\ell}^i\|^2+\mathfrak{L}^\mu)\right).$$

Thus, iterating the bound for $k-\ell<j<k$ gives

$$\mathrm{KL}(\tilde{\mu}_k^i\|\Pi\,\widehat{\mu}_{k-\ell}^{(N)})\leq\exp(-(\ell-1)\alpha_\mu\lambda\eta)\,\mathrm{KL}(\tilde{\mu}_{k-\ell+1}^i\|\Pi\,\widehat{\mu}_{k-\ell}^{(N)})+\mathfrak{K}^\mu\|X_{k-\ell}^i\|^2+\mathfrak{L}^\mu,$$

where we have stopped at time $k-\ell+1$ because the initial distribution $\tilde{\mu}_{k-\ell}^i=\delta_{X_{k-\ell}^i}$ is atomic.

Instead, the relative entropy after the first step can be directly bounded; since $X_t^\dagger$ is a rescaled Brownian motion with constant drift, the first iteration of $\delta_{X_{k-\ell}^i}$ is distributed as

$$\tilde{\mu}_{k-\ell+1}^i\overset{d}{=}\mathcal{N}(X_{k-\ell}^i+\eta\,\mathfrak{b}_{k-\ell}^\mu(X_{k-\ell}^i),2\lambda\eta\,\mathrm{I}_{d_{\mathcal{X}}}).$$

The LSI then gives that

$$\mathrm{KL}(\tilde{\mu}_{k-\ell+1}^i\|\Pi\,\widehat{\mu}_{k-\ell}^{(N)})\leq\frac{1}{2\alpha_\mu}\mathbb{E}_{\tilde{\mu}_{k-\ell+1}^i}\left[\left\|\nabla_x\log\frac{\tilde{\mu}_{k-\ell+1}^i}{\Pi\,\widehat{\mu}_{k-\ell}^{(N)}}\right\|^2\right]$$

$$\leq\frac{3}{2\alpha_\mu}\left(\frac{d_{\mathcal{X}}}{2\lambda\eta}+\frac{M_\mu^2}{\lambda^2}+R_\mu^2\,\mathbb{E}_{X_{k-\ell+1}^i|(\mathcal{X},\mathcal{Y})_{1:k-\ell}}[\|X_{k-\ell+1}^i\|^2]\right)$$

$$\leq\frac{3}{2\alpha_\mu}\left(\frac{d_{\mathcal{X}}}{2\lambda\eta}+\frac{M_\mu^2}{\lambda^2}+R_\mu^2\left(\|X_{k-\ell}^i\|^2\vee\mathfrak{s}^\mu\right)\right)$$

$$<\frac{3}{2\eta^2(L_\mu+\lambda R_\mu)^2}(\mathfrak{K}^\mu\|X_{k-\ell}^i\|^2+\mathfrak{L}^\mu).$$

Hence we arrive at the desired statement,

$$\mathrm{KL}(\tilde{\mu}_k^i\|\Pi\,\widehat{\mu}_{k-\ell}^{(N)})\leq\left(1+\frac{3\exp(-(\ell-1)\alpha_\mu\lambda\eta)}{2\eta^2(L_\mu+\lambda R_\mu)^2}\right)(\mathfrak{K}^\mu\|X_{k-\ell}^i\|^2+\mathfrak{L}^\mu).$$

$\square$

The subsequent lemmas provide control over the Wasserstein distance between pushforward distributions. In particular, Lemma C.8 is the discrete analogue of the $O(\beta_t/B_t)$ time derivative bound obtained in the proof of Proposition 3.3.

**Lemma C.7.** *For any two measures $\mu^{(N)},\tilde{\mu}^{(N)}\in\mathcal{P}_2(\mathcal{X}^N)$ it holds that*

$$W_2(\Pi\mu^{(N)},\Pi\tilde{\mu}^{(N)})\leq\frac{1}{\sqrt{N}}W_2(\mu^{(N)},\tilde{\mu}^{(N)}).$$

*Proof.* Recall the dual formulation of $W_2$,

$$W_2^2(\mu,\tilde{\mu})=\sup_{\phi,\psi}\left\{\int\phi\,\mathrm{d}\mu-\int\psi\,\mathrm{d}\tilde{\mu}\;\middle|\;\phi,\psi:\mathcal{X}\to\mathbb{R},\;\phi(x)-\psi(y)\leq\|x-y\|^2\right\}.$$

Then for any pair of functions $\phi, \psi$ such that $\phi(x) - \psi(y) \leq \|x - y\|^2$, the pullback functions $\Pi^*\phi, \Pi^*\psi$ on $\mathcal{X}^N$ satisfy

$$\Pi^*\phi(\mathscr{X}) - \Pi^*\psi(\mathscr{Y}) = \frac{1}{N}\sum_{i=1}^N \phi(X^i) - \psi(Y^i) \leq \frac{1}{N}\sum_{i=1}^N \|X^i - Y^i\|^2 = \frac{1}{N}\|\mathscr{X} - \mathscr{Y}\|_{L^2(\mathcal{X}^N)}^2.$$

Therefore,

$$\int_{\mathcal{X}} \phi(x)\Pi\mu^{(N)}(\mathrm{d}x) - \int_{\mathcal{X}} \psi(x)\Pi\tilde{\mu}^{(N)}(\mathrm{d}x)$$
$$= \int_{\mathcal{X}^N} \Pi^*\phi(\mathscr{X})\mu^{(N)}(\mathrm{d}\mathscr{X}) - \int_{\mathcal{X}^N} \Pi^*\psi(\mathscr{X})\tilde{\mu}^{(N)}(\mathrm{d}\mathscr{X}) \leq \frac{1}{N}W_2^2(\mu^{(N)}, \tilde{\mu}^{(N)}),$$

which yields the assertion by taking the supremum over all permissible $\phi, \psi$. $\qquad\square$

**Lemma C.8.** *The projected 2-Wasserstein distance between $\widehat{\mu}_k^{(N)}$, $\widehat{\mu}_{k-1}^{(N)}$ is bounded as*

$$W_2(\Pi\widehat{\mu}_k^{(N)}, \Pi\widehat{\mu}_{k-1}^{(N)}) \leq \frac{2M_\mu\beta_k}{\alpha_\mu\lambda B_k}.$$

*Proof.* The proof is deferred to Section C.4. $\qquad\square$

### C.3 PROOF OF PROPOSITION 3.6

We take $\ell = \ell^\mu = O(\eta^{-1}\log\eta^{-1})$ as defined in (9) throughout the proof and only consider the case $k \geq 2\ell$ in Steps 1 through 4.

*Step 1.* We first look $\ell - 1$ steps back to the past and control the displacement of the proximal $\Pi\widehat{\mu}_{k-1}^{(N)}$ from the stationary state $\Pi\widehat{\mu}_{k-\ell}^{(N)}$ of the modified process via Lemma C.8, conditioning on the earlier history $(\mathscr{X}, \mathscr{Y})_{1:k-\ell}$.

$$\mathbb{E}_{(\mathscr{X},\mathscr{Y})_{k-\ell+1:k}|(\mathscr{X},\mathscr{Y})_{1:k-\ell}}\left[\int_{\mathcal{X}} F(\mu_{\mathscr{X}_k}, \nu_{\mathscr{Y}_k})(\mu_{\mathscr{X}_k} - \Pi\widehat{\mu}_{k-1}^{(N)})(\mathrm{d}x)\right]$$
$$\leq \mathbb{E}_{(\mathscr{X},\mathscr{Y})_k|(\mathscr{X},\mathscr{Y})_{1:k-\ell}}\left[\int_{\mathcal{X}} F(\mu_{\mathscr{X}_k}, \nu_{\mathscr{Y}_k})(\mu_{\mathscr{X}_k} - \Pi\widehat{\mu}_{k-\ell}^{(N)})(\mathrm{d}x)\right]$$
$$\quad + M_\mu\sum_{j=1}^{\ell-1}\mathbb{E}_{(\mathscr{X},\mathscr{Y})_{k-\ell+1:k-j}|(\mathscr{X},\mathscr{Y})_{1:k-\ell}}\left[W_1(\Pi\widehat{\mu}_{k-j-1}^{(N)}, \Pi\widehat{\mu}_{k-j}^{(N)})\right]$$
$$\leq \mathbb{E}_{(\mathscr{X},\mathscr{Y})_k|(\mathscr{X},\mathscr{Y})_{1:k-\ell}}\left[\int_{\mathcal{X}} F(\mu_{\mathscr{X}_k}, \nu_{\mathscr{Y}_k})(\mu_{\mathscr{X}_k} - \Pi\widehat{\mu}_{k-\ell}^{(N)})(\mathrm{d}x)\right] + \frac{2M_\mu^2}{\alpha_\mu\lambda}\sum_{j=1}^{\ell-1}\frac{\beta_{k-j}}{B_{k-j}}.$$

It is simple to further verify that

$$\frac{2M_\mu^2}{\alpha_\mu\lambda}\sum_{j=1}^{\ell-1}\frac{\beta_{k-j}}{B_{k-j}} \leq \frac{2M_\mu^2}{\alpha_\mu\lambda}\frac{(r+1)(\ell-1)}{k-\ell+1}.$$

*Step 2.* Next, we look back to the future and convert the expectation with respect to $\mu_{\mathscr{X}_k}$ to the corresponding expectation for the modified process. The incurred error can be bounded by utilizing Lemmas C.2, C.3 and C.4 as

$$\mathbb{E}_{(\mathscr{X},\mathscr{Y})_k|(\mathscr{X},\mathscr{Y})_{1:k-\ell}}\left[\int_{\mathcal{X}} F(\mu_{\mathscr{X}_k}, \nu_{\mathscr{Y}_k})(\mu_{\mathscr{X}_k} - \Pi\widehat{\mu}_{k-\ell}^{(N)})(\mathrm{d}x)\right]$$
$$\quad - \mathbb{E}_{(\widetilde{\mathscr{X}},\widetilde{\mathscr{Y}})_k|(\mathscr{X},\mathscr{Y})_{1:k-\ell}}\left[\int_{\mathcal{X}} F(\mu_{\widetilde{\mathscr{X}}_k}, \nu_{\widetilde{\mathscr{Y}}_k})(\mu_{\widetilde{\mathscr{X}}_k} - \Pi\widehat{\mu}_{k-\ell}^{(N)})(\mathrm{d}x)\right]$$
$$= \mathbb{E}_{(\mathscr{X},\widetilde{\mathscr{X}},\mathscr{Y},\widetilde{\mathscr{Y}})_k|(\mathscr{X},\mathscr{Y})_{1:k-\ell}}\left[\int_{\mathcal{X}} F(\mu_{\mathscr{X}_k}, \nu_{\mathscr{Y}_k})(\mu_{\mathscr{X}_k} - \mu_{\widetilde{\mathscr{X}}_k})(\mathrm{d}x)\right.$$

$$+ \int_{\mathcal{X}} \left( F(\mu_{\mathscr{X}_k}, \nu_{\mathscr{Y}_k}) - F(\mu_{\widetilde{\mathscr{X}_k}}, \nu_{\widetilde{\mathscr{Y}_k}}) \right) (\mu_{\widetilde{\mathscr{X}_k}} - \Pi \widehat{\mu}_{k-\ell}^{(N)})(\mathrm{d}x) \bigg]$$

$$\leq \mathbb{E}_{(\mathscr{X}, \widetilde{\mathscr{X}}, \mathscr{Y}, \widetilde{\mathscr{Y}})_k | (\mathscr{X}, \mathscr{Y})_{1:k-\ell}} \bigg[ M_\mu W_1(\mu_{\mathscr{X}_k}, \mu_{\widetilde{\mathscr{X}_k}})$$

$$+ \frac{1}{N} \sum_{i=1}^{N} \left\| F(\mu_{\mathscr{X}_k}, \nu_{\mathscr{Y}_k}) - F(\mu_{\widetilde{\mathscr{X}_k}}, \nu_{\widetilde{\mathscr{Y}_k}}) \right\|_{\mathrm{Lip}} W_1(\delta_{\widetilde{X}_k^i}, \Pi \widehat{\mu}_{k-\ell}^{(N)}) \bigg]$$

$$\leq \frac{(r+1) M_\mu}{k - \ell + 1} \mathfrak{w}_\ell^\mu$$

$$+ \frac{(r+1) L_\mu}{k - \ell + 1} (\mathfrak{w}_\ell^\mu + \mathfrak{w}_\ell^\nu) \, \mathbb{E}_{(\widetilde{X}_k^i | (\mathscr{X}, \mathscr{Y})_{1:k-\ell}} \bigg[ \left( \frac{2}{N} \sum_{i=1}^{N} \int_{\mathcal{X}} \| \widetilde{X}_k^i - x \|^2 \Pi \widehat{\mu}_{k-\ell}^{(N)}(\mathrm{d}x) \right)^{\frac{1}{2}} \bigg]$$

$$\leq \frac{(r+1) M_\mu}{k - \ell + 1} \mathfrak{w}_\ell^\mu + \frac{(r+1) L_\mu}{k - \ell + 1} (\mathfrak{w}_\ell^\mu + \mathfrak{w}_\ell^\nu) \left( \frac{2}{N} \sum_{i=1}^{N} \| X_{k-\ell}^i \|^2 + \mathfrak{q}^\mu + 2 \mathfrak{s}^\mu \right)^{\frac{1}{2}}.$$

*Step 3.* For the modified process, we apply a leave-one-out argument and consider the expectation with respect to each particle $\widetilde{X}_k^i$ which is independent of $\widetilde{\mathscr{X}}_k^{-i}$, $\widetilde{\mathscr{Y}_k}$ when conditioned on the stopped history $(\mathscr{X}, \mathscr{Y})_{1:k-\ell}$. That is,

$$\mathbb{E}_{(\widetilde{\mathscr{X}}, \widetilde{\mathscr{Y}})_k | (\mathscr{X}, \mathscr{Y})_{1:k-\ell}} \left[ \int_{\mathcal{X}} F(\mu_{\widetilde{\mathscr{X}_k}}, \nu_{\widetilde{\mathscr{Y}_k}}) (\mu_{\widetilde{\mathscr{X}_k}} - \Pi \widehat{\mu}_{k-\ell}^{(N)})(\mathrm{d}x) \right]$$

$$= \frac{1}{N} \sum_{i=1}^{N} \mathbb{E}_{\widetilde{\mathscr{X}}_k^{-i}, \widetilde{\mathscr{Y}_k} | (\mathscr{X}, \mathscr{Y})_{1:k-\ell}} \mathbb{E}_{\widetilde{X}_k^i | (\mathscr{X}, \mathscr{Y})_{1:k-\ell}} \left[ \int_{\mathcal{X}} F(\mu_{\widetilde{\mathscr{X}_k}}, \nu_{\widetilde{\mathscr{Y}_k}}) (\delta_{\widetilde{X}_k^i} - \Pi \widehat{\mu}_{k-\ell}^{(N)})(\mathrm{d}x) \right]$$

$$\leq \frac{1}{N} \sum_{i=1}^{N} \mathbb{E}_{\widetilde{\mathscr{X}}_k^{-i}, \widetilde{\mathscr{Y}_k} | (\mathscr{X}, \mathscr{Y})_{1:k-\ell}} \mathbb{E}_{\widetilde{X}_k^i | (\mathscr{X}, \mathscr{Y})_{1:k-\ell}} \left[ \int_{\mathcal{X}} F(\mu_{\widetilde{\mathscr{X}_k^{-i}}}, \nu_{\widetilde{\mathscr{Y}_k}}) (\delta_{\widetilde{X}_k^i} - \Pi \widehat{\mu}_{k-\ell}^{(N)})(\mathrm{d}x) \right]$$

$$+ \frac{1}{N} \sum_{i=1}^{N} \mathbb{E}_{\widetilde{\mathscr{X}_k} | (\mathscr{X}, \mathscr{Y})_{1:k-\ell}} \left[ \left\| F(\mu_{\widetilde{\mathscr{X}_k}}, \nu_{\widetilde{\mathscr{Y}_k}}) - F(\mu_{\widetilde{\mathscr{X}_k^{-i}}}, \nu_{\widetilde{\mathscr{Y}_k}}) \right\|_{\mathrm{Lip}} W_1(\delta_{\widetilde{X}_k^i}, \Pi \widehat{\mu}_{k-\ell}^{(N)}) \right]$$

$$\leq \frac{1}{N} \sum_{i=1}^{N} \mathbb{E}_{\widetilde{\mathscr{X}}_k^{-i}, \widetilde{\mathscr{Y}_k} | (\mathscr{X}, \mathscr{Y})_{1:k-\ell}} \mathbb{E}_{\widetilde{X}_k^i | (\mathscr{X}, \mathscr{Y})_{1:k-\ell}} \left[ \int_{\mathcal{X}} F(\mu_{\widetilde{\mathscr{X}_k^{-i}}}, \nu_{\widetilde{\mathscr{Y}_k}}) (\delta_{\widetilde{X}_k^i} - \Pi \widehat{\mu}_{k-\ell}^{(N)})(\mathrm{d}x) \right]$$

$$+ \frac{L_\mu}{N} \sum_{i=1}^{N} \mathbb{E}_{\widetilde{\mathscr{X}_k} | (\mathscr{X}, \mathscr{Y})_{1:k-\ell}} \left[ W_1(\mu_{\widetilde{\mathscr{X}_k}}, \mu_{\widetilde{\mathscr{X}_k^{-i}}}) W_1(\delta_{\widetilde{X}_k^i}, \Pi \widehat{\mu}_{k-\ell}^{(N)}) \right]$$

$$= \frac{1}{N} \sum_{i=1}^{N} \mathbb{E}_{\widetilde{\mathscr{X}}_k^{-i}, \widetilde{\mathscr{Y}_k} | (\mathscr{X}, \mathscr{Y})_{1:k-\ell}} \left[ \int_{\mathcal{X}} F(\mu_{\widetilde{\mathscr{X}_k^{-i}}}, \nu_{\widetilde{\mathscr{Y}_k}}) (\mu_k^i(\widetilde{X}_k^i) - \Pi \widehat{\mu}_{k-\ell}^{(N)})(\mathrm{d}x) \right]$$

$$+ \frac{L_\mu}{N} \sum_{i=1}^{N} \mathbb{E}_{\widetilde{\mathscr{X}_k} | (\mathscr{X}, \mathscr{Y})_{1:k-\ell}} \left[ W_1(\mu_{\widetilde{\mathscr{X}_k}}, \mu_{\widetilde{\mathscr{X}_k^{-i}}}) W_1(\delta_{\widetilde{X}_k^i}, \Pi \widehat{\mu}_{k-\ell}^{(N)}) \right]$$

$$\leq \frac{M_\mu}{N} \sum_{i=1}^{N} W_1(\mu_k^i, \Pi \widehat{\mu}_{k-\ell}^{(N)})$$

$$+ \frac{L_\mu}{N} \sum_{i=1}^{N} \left( \mathbb{E}_{\widetilde{\mathscr{X}_k} | (\mathscr{X}, \mathscr{Y})_{1:k-\ell}} \left[ W_2^2(\mu_{\widetilde{\mathscr{X}_k}}, \mu_{\widetilde{\mathscr{X}_k^{-i}}}) \right] \mathbb{E}_{\widetilde{\mathscr{X}_k} | (\mathscr{X}, \mathscr{Y})_{1:k-\ell}} \left[ W_2^2(\delta_{\widetilde{X}_k^i}, \Pi \widehat{\mu}_{k-\ell}^{(N)}) \right] \right)^{\frac{1}{2}}$$

$$\leq \frac{2 M_\mu}{N} \sum_{i=1}^{N} \sqrt{\alpha_\mu^{-1} (\mathfrak{K}^\mu \| X_{k-\ell}^i \|^2 + \mathfrak{L}^\mu)}$$

$$+ \frac{2 L_\mu}{N} \sum_{i=1}^{N} \left( \frac{2 \mathfrak{s}^\mu}{N} + \frac{1}{N(N-1)} \sum_{j \neq i} \| X_{k-\ell}^j \|^2 + \frac{1}{N} \| X_{k-\ell}^i \|^2 \right)^{\frac{1}{2}} \left( \| X_{k-\ell}^i \|^2 + \mathfrak{q}^\mu + \mathfrak{s}^\mu \right)^{\frac{1}{2}}$$

by applying Lemma C.2, Lemma C.4 and Proposition C.6.

*Step 4.* Putting things together, we obtain the conditional bound

$$
\mathbb{E}_{(\mathscr{X},\mathscr{Y})_{k-\ell+1:k}|(\mathscr{X},\mathscr{Y})_{1:k-\ell}} \left[ \int_{\mathcal{X}} F(\mu_{\mathscr{X}_k}, \nu_{\mathscr{Y}_k})(\mu_{\mathscr{X}_k} - \Pi \widehat{\mu}_{k-1}^{(N)})(\mathrm{d}x) \right]
$$
$$
\leq \frac{2M_\mu^2}{\alpha_\mu \lambda} \frac{(r+1)(\ell-1)}{k-\ell+1}
$$
$$
+ \frac{(r+1)M_\mu}{k-\ell+1} \mathfrak{w}_\ell^\mu + \frac{(r+1)L_\mu}{k-\ell+1} (\mathfrak{w}_\ell^\mu + \mathfrak{w}_\ell^\nu) \left( \frac{2}{N} \sum_{i=1}^N \|X_{k-\ell}^i\|^2 + \mathfrak{q}^\mu + 2\mathfrak{s}^\mu \right)^{\frac{1}{2}}
$$
$$
+ \frac{2M_\mu}{N} \sum_{i=1}^N \sqrt{\alpha_\mu^{-1}(\mathfrak{K}^\mu \|X_{k-\ell}^i\|^2 + \mathfrak{L}^\mu)}
$$
$$
+ \frac{2L_\mu}{N} \sum_{i=1}^N \left( \frac{2\mathfrak{s}^\mu}{N} + \frac{1}{N(N-1)} \sum_{j \neq i} \|X_{k-\ell}^j\|^2 + \frac{1}{N}\|X_{k-\ell}^i\|^2 \right)^{\frac{1}{2}} \left( \|X_{k-\ell}^i\|^2 + \mathfrak{q}^\mu + \mathfrak{s}^\mu \right)^{\frac{1}{2}}.
$$

Recalling $\mathbb{E}[\|X_{k-\ell}^i\|^2] \leq \mathbb{E}[\|X_1^i\|^2] + \mathfrak{s}^\mu$ from Lemma C.2, taking the expectation with respect to the history $(\mathscr{X},\mathscr{Y})_{1:k-\ell}$ finally gives

$$
\mathbb{E}_{(\mathscr{X},\mathscr{Y})_{1:k}} \left[ \int_{\mathcal{X}} F(\mu_{\mathscr{X}_k}, \nu_{\mathscr{Y}_k})(\mu_{\mathscr{X}_k} - \Pi \widehat{\mu}_{k-1}^{(N)})(\mathrm{d}x) \right]
$$
$$
\leq \frac{r+1}{k-\ell+1} \left( \frac{2M_\mu^2}{\alpha_\mu \lambda}(\ell-1) + M_\mu \mathfrak{w}_\ell^\mu + L_\mu (\mathfrak{w}_\ell^\mu + \mathfrak{w}_\ell^\nu)(2\mathfrak{p}^\mu + \mathfrak{q}^\mu + 4\mathfrak{s}^\mu)^{\frac{1}{2}} \right)
$$
$$
+ 2M_\mu \mathbb{E}_{(\mathscr{X},\mathscr{Y})_{1:k}} \left[ \frac{1}{\alpha_\mu N} \sum_{i=1}^N (\mathfrak{K}^\mu \|X_{k-\ell}^i\|^2 + \mathfrak{L}^\mu) \right]^{\frac{1}{2}}
$$
$$
+ \frac{L_\mu}{N^{\frac{3}{2}}} \sum_{i=1}^N \mathbb{E}_{(\mathscr{X},\mathscr{Y})_{1:k}} \left[ \frac{1}{N-1} \sum_{j=1}^N \|X_{k-\ell}^j\|^2 + \frac{2N-3}{N-1}\|X_{k-\ell}^i\|^2 + \mathfrak{q}^\mu + 3\mathfrak{s}^\mu \right]
$$
$$
\leq \frac{r+1}{k-\ell+1} \left( \frac{2M_\mu^2}{\alpha_\mu \lambda}(\ell-1) + M_\mu \mathfrak{w}_\ell^\mu + L_\mu (\mathfrak{w}_\ell^\mu + \mathfrak{w}_\ell^\nu)(2\mathfrak{p}^\mu + \mathfrak{q}^\mu + 4\mathfrak{s}^\mu)^{\frac{1}{2}} \right)
$$
$$
+ 2M_\mu \left( \frac{\mathfrak{K}^\mu \mathfrak{p}^\mu + \mathfrak{L}^\mu}{\alpha_\mu} \right)^{\frac{1}{2}} + \frac{2L_\mu}{\sqrt{N}} (3\mathfrak{p}^\mu + \mathfrak{q}^\mu + 6\mathfrak{s}^\mu)
$$
$$
\leq \frac{r+1}{k} C_1(\eta) + C_2\sqrt{\eta} + \frac{C_3}{\sqrt{N}},
$$

where the last bound holds if $k \geq 2\ell^\mu$. To be explicit,

$$
C_1(\eta) = 2 \left( \frac{2M_\mu^2}{\alpha_\mu \lambda}(\ell-1) + M_\mu \mathfrak{w}_\ell^\mu + L_\mu (\mathfrak{w}_\ell^\mu + \mathfrak{w}_\ell^\nu)(2\mathfrak{p}^\mu + \mathfrak{q}^\mu + 4\mathfrak{s}^\mu)^{\frac{1}{2}} \right),
$$
$$
C_2 = 2M_\mu \left( \frac{\bar{\eta} R_\mu^2 (L_\mu + \lambda R_\mu)^2 \mathfrak{p}^\mu}{\alpha_\mu^2} + \frac{(L_\mu + \lambda R_\mu)^2}{\alpha_\mu^2 \lambda^2} \left( \bar{\eta} M_\mu^2 + \lambda^2 \bar{\eta} R_\mu^2 \mathfrak{s}^\mu + \lambda d_{\mathcal{X}} \right) \right)^{\frac{1}{2}},
$$
$$
C_3 = 2L_\mu (3\mathfrak{p}^\mu + \mathfrak{q}^\mu + 6\mathfrak{s}^\mu).
$$

The constants $C_2, C_3$ can be taken to be polynomial and independent of $\eta$ by substituting in the upper bound $\bar{\eta} = \frac{r_\mu \lambda}{2(L_\mu + \lambda R_\mu)^2} \wedge \frac{r_\mu}{4\lambda R_\mu^2}$ in the expressions for $\mathfrak{s}^\mu, \mathfrak{K}^\mu/\eta, \mathfrak{L}^\mu/\eta$, while $\ell^\mu = O(\eta^{-1} \log \eta^{-1})$. However, $C_1(\eta)$ contains the dependency

$$
O(\mathfrak{w}_\ell^\mu) = O\left( \eta^{-1} \exp(\ell L_\mu \eta) \right) = O\left( \frac{1}{\eta} \left( \frac{3}{2\eta^2 (L_\mu^2 + \lambda R_\mu^2)^2} \right)^{\frac{L_\mu}{\alpha_\mu \lambda}} \right),
$$

which is a consequence of uniformly bounding the perturbation from the gradient stopped process over a time period of $\ell$.

*Step 5.* For $k < 2\ell$, proceeding similarly without converting to the modified process gives

$$\mathbb{E}_{(\mathscr{X},\mathscr{Y})_{1:k}}\left[\int_{\mathcal{X}} F(\mu_{\mathscr{X}_k},\nu_{\mathscr{Y}_k})(\mu_{\mathscr{X}_k} - \Pi\widehat{\mu}_{k-1}^{(N)})(\mathrm{d}x)\right]$$

$$\leq \frac{M_\mu}{N}\sum_{i=1}^N \mathbb{E}_{(\mathscr{X},\mathscr{Y})_{1:k}}\left[W_1(\delta_{X_k^i}, \Pi\widehat{\mu}_{k-1}^{(N)})\right]$$

$$+ \frac{L_\mu}{N}\sum_{i=1}^N\left(\mathbb{E}_{(\mathscr{X},\mathscr{Y})_{1:k}}\left[W_2^2(\mu_{\widetilde{\mathscr{X}}_k}, \mu_{\widetilde{\mathscr{X}}_k^{-i}})\right]\mathbb{E}_{(\mathscr{X},\mathscr{Y})_{1:k}}\left[W_2^2(\delta_{\widetilde{X}_k^i}, \Pi\widehat{\mu}_{k-1}^{(N)})\right]\right)^{\frac{1}{2}}$$

$$\leq \frac{M_\mu}{N}\sum_{i=1}^N\left(\mathbb{E}[\|X_1^i\|^2] + \mathfrak{q}^\mu + \mathfrak{s}^\mu\right)^{\frac{1}{2}}$$

$$+ \frac{2L_\mu}{N}\sum_{i=1}^N\left(\frac{2\mathfrak{s}^\mu}{N} + \frac{1}{N(N-1)}\sum_{k\neq i}\mathbb{E}[\|X_1^k\|^2] + \frac{1}{N}\mathbb{E}[\|X_1^i\|^2]\right)^{\frac{1}{2}}\left(\mathbb{E}[\|X_1^i\|^2] + \mathfrak{q}^\mu + \mathfrak{s}^\mu\right)^{\frac{1}{2}}$$

$$\leq M_\mu\sqrt{\mathfrak{p}^\mu + \mathfrak{q}^\mu + \mathfrak{s}^\mu} + \frac{2L_\mu(3\mathfrak{p}^\mu + \mathfrak{q}^\mu + 3\mathfrak{s}^\mu)}{\sqrt{N}}$$

$$< \frac{C_1(\eta)}{2\ell} + \frac{C_3}{\sqrt{N}},$$

where the final bound follows by noting $\eta < \frac{r_\mu}{4L_\mu R_\mu} \leq \frac{1}{4L_\mu}$, hence by expanding $(1 + 2\eta L_\mu)^\ell$

$$(\mathfrak{w}_\ell^\mu)^2 > \frac{1}{2L_\mu}\cdot\frac{M_\mu^2}{\eta^2 L_\mu^3}\left(2\eta L_\mu\ell + 2\eta^2 L_\mu^2\ell(\ell-1)\right) > \frac{M_\mu^2\ell^2}{L_\mu^2}$$

and so

$$C_1(\eta) > 2L_\mu\mathfrak{w}_\ell^\mu\left(2\mathfrak{p}^\mu + \mathfrak{q}^\mu + 4\mathfrak{s}^\mu\right)^{\frac{1}{2}} > 2M_\mu\ell\left(\mathfrak{p}^\mu + \mathfrak{q}^\mu + \mathfrak{s}^\mu\right)^{\frac{1}{2}}.$$

Thus the bound holds for all integers $k$. We conclude the proof by taking the maximum with the corresponding quantities for $\nu$. $\qquad\square$

## C.4 PROPERTIES OF CONJUGATE FUNCTIONALS

We proceed to develop the $N$-particle lifted analogues $J_k^{(N)}, \widehat{J}_k^{(N)}$ of the conjugate functionals in the proof of Theorem 3.4. In order to deal with time and particle discretization, we will need a more precise characterization of their perturbative properties. Many of the subsequent results do not follow from standard methods and requires a careful synthesis of the discussion thus far.

**Lemma C.9.** *Given Lipschitz functions $\zeta_\mu : \mathcal{X} \to \mathbb{R}$, $\zeta_\nu : \mathcal{Y} \to \mathbb{R}$ and a pair of $N$-particle probability measures $\mu^{(N)} \in \mathcal{P}_2(\mathcal{X}^N)$, $\nu^{(N)} \in \mathcal{P}_2(\mathcal{Y}^N)$ define the functional*

$$J_k^{(N)}(\mu^{(N)},\nu^{(N)}|\zeta^\mu,\zeta^\nu)$$

$$= -\int_{\mathcal{X}^N}\int_{\mathcal{X}}\zeta^\mu(\mu_{\mathscr{X}} - \rho^\mu)(\mathrm{d}x)\mu^{(N)}(\mathrm{d}\mathscr{X}) + \int_{\mathcal{Y}^N}\int_{\mathcal{Y}}\zeta^\nu(\nu_{\mathscr{Y}} - \rho^\nu)(\mathrm{d}y)\nu^{(N)}(\mathrm{d}\mathscr{Y})$$

$$- \frac{\lambda B_k}{N}\left(\mathrm{KL}(\mu^{(N)}\|\rho^{\mu\otimes N}) + \mathrm{KL}(\nu^{(N)}\|\rho^{\nu\otimes N})\right).$$

*Then the maximum*

$$\widehat{J}_k^{(N)}(\zeta^\mu,\zeta^\nu) = \max_{\mu^{(N)}\in\mathcal{P}_2(\mathcal{X}^N)}\max_{\nu^{(N)}\in\mathcal{P}_2(\mathcal{Y}^N)} J_k^{(N)}(\mu^{(N)},\nu^{(N)}|\zeta^\mu,\zeta^\nu)$$

*exists for all $k \in \mathbb{N}$ and is uniquely attained by the pair of distributions*

$$\widehat{\mu}_k^{(N)}(\zeta^\mu) \propto \rho^{\mu\otimes N}\exp\left(-\frac{N}{\lambda B_k}\int_{\mathcal{X}}\zeta^\mu\mu_{\mathscr{X}}(\mathrm{d}x)\right), \quad \widehat{\nu}_k^{(N)}(\zeta^\nu) \propto \rho^{\nu\otimes N}\exp\left(\frac{N}{\lambda B_k}\int_{\mathcal{Y}}\zeta^\nu\nu_{\mathscr{Y}}(\mathrm{d}y)\right).$$

*Proof.* The proof is similar to Lemma B.3; we only check the first-order condition by setting

$$\frac{\delta J_k^{(N)}}{\delta \mu^{(N)}}(\mu^{(N)})(\mathscr{X}) = -\int_{\mathcal{X}} \zeta^{\mu}(\mu_{\mathscr{X}} - \rho^{\mu})(\mathrm{d}x) - \frac{\lambda B_k}{N} \log \frac{\mu^{(N)}(\mathscr{X})}{\rho^{\mu \otimes N}(\mathscr{X})} = \text{const.}$$

$\square$

The $N$-particle proximal distributions $\widehat{\mu}_k^{(N)}(\zeta^{\mu}), \widehat{\nu}_k^{(N)}(\zeta^{\nu})$, despite being defined over the configuration spaces $\mathcal{X}^N, \mathcal{Y}^N$ also satisfy the log-Sobolev inequality with the same constant as before due to the tensorization property of entropy.

**Lemma C.10** (product log-Sobolev inequality)**.** *Suppose that $\zeta^{\mu}/B_k, \zeta^{\nu}/B_k$ are $M_{\mu}, M_{\nu}$-Lipschitz, respectively. Then $\widehat{\mu}_k^{(N)}(\zeta^{\mu}), \widehat{\nu}_k^{(N)}(\zeta^{\nu})$ satisfy the LSI on $\mathcal{X}^N, \mathcal{Y}^N$, with the same constants $\alpha_{\mu}, \alpha_{\nu}$ as in Proposition 3.2.*

*Proof.* We can write $\mu^{(N)} = \widehat{\mu}_k^{(N)}(\zeta^{\mu})$ as the symmetric product distribution

$$\mu^{(N)}(\mathscr{X}) = \prod_{i=1}^{N} \mu^i(X^i), \quad \mu^i(X^i) = \rho^{\mu}(X^i) \exp\left(-\frac{\zeta^{\mu}(X^i)}{\lambda B_k}\right), \quad 1 \le i \le N,$$

where the marginals $\mu^i(X^i)$ each satisfy the LSI with constant $\alpha_{\mu}$ by Proposition 3.2. Also write $\mu^{-i}(X^{-i}) = \prod_{j \ne i} \mu^i(X^i)$. For an appropriately integrable function $f$ on $\mathcal{X}^N$, denote by $f^i$ for the functions $f^i(X^i) = f(X^1, \cdots, X^i, \cdots, X^N)$. Then by Proposition 2.2 of Ledoux (1999),

$$\text{Ent}_{\mu^{(N)}}(f^2) \le \sum_{i=1}^{N} \mathbb{E}_{\mu^{-i}}[\text{Ent}_{\mu^i}((f^i)^2)] \le \sum_{i=1}^{N} \frac{2}{\alpha_{\mu}} \mathbb{E}_{\mu^{-i}} \mathbb{E}_{\mu^i}[\|\nabla f^i\|^2] = \frac{2}{\alpha_{\mu}} \mathbb{E}_{\mu^{(N)}}[\|\nabla f\|^2].$$

$\square$

**Lemma C.11.** *The functional $\widehat{J}_k^{(N)}$ is convex in both arguments, and admits functional derivatives at any $(\zeta^{\mu}, \zeta^{\nu})$ which are given as*

$$\frac{\delta \widehat{J}_k^{(N)}}{\delta \zeta^{\mu}}(\zeta^{\mu}, \zeta^{\nu}) = -\Pi \widehat{\mu}_k^{(N)}(\zeta^{\mu}) + \rho^{\mu}, \quad \frac{\delta \widehat{J}_k^{(N)}}{\delta \zeta^{\nu}}(\zeta^{\mu}, \zeta^{\nu}) = \Pi \widehat{\nu}_k^{(N)}(\zeta^{\nu}) - \rho^{\nu}.$$

*Proof.* Substituting $\widehat{J}_k^{(N)}(\zeta^{\mu}, \zeta^{\nu}) = J_k^{(N)}(\widehat{\mu}_k^{(N)}(\zeta^{\mu}), \widehat{\mu}_k^{(N)}(\zeta^{\nu}) | \zeta^{\mu}, \zeta^{\nu})$,

$$\frac{\delta \widehat{J}_k^{(N)}}{\delta \zeta^{\mu}}(\zeta^{\mu}, \zeta^{\nu}) = -\frac{\delta}{\delta \zeta^{\mu}} \int_{\mathcal{X}^N} \int_{\mathcal{X}} \zeta^{\mu}(\mu_{\mathscr{X}} - \rho^{\mu})(\mathrm{d}x) \mu^{(N)}(\mathrm{d}\mathscr{X})\bigg|_{\mu^{(N)} = \widehat{\mu}_k^{(N)}(\zeta^{\mu})}$$

$$-\int_{\mathcal{X}^N} \int_{\mathcal{X}} \zeta^{\mu}(\mu_{\mathscr{X}} - \rho^{\mu})(\mathrm{d}x) \frac{\delta \widehat{\mu}_k^{(N)}}{\delta \zeta^{\mu}}(\zeta^{\mu})(\mathrm{d}\mathscr{X}) - \frac{\lambda B_k}{N} \int_{\mathcal{X}^N} \left(\log \frac{\widehat{\mu}_k^{(N)}(\zeta^{\mu})}{\rho^{\mu \otimes N}}\right) \frac{\delta \widehat{\mu}_k^{(N)}}{\delta \zeta^{\mu}}(\zeta^{\mu})(\mathrm{d}\mathscr{X})$$

$$= \frac{\delta}{\delta \zeta^{\mu}} \left(-\int_{\mathcal{X}^N} \frac{1}{N} \sum_{i=1}^{N} \zeta^{\mu}(X^i) \mu^{(N)}(\mathrm{d}\mathscr{X}) + \int_{\mathcal{X}} \zeta^{\mu} \rho^{\mu}(\mathrm{d}x)\right)\bigg|_{\mu^{(N)} = \widehat{\mu}_k^{(N)}(\zeta^{\mu})}$$

$$= -\Pi \widehat{\mu}_k^{(N)}(\zeta^{\mu}) + \rho^{\mu}.$$

The integral over the configuration space measure $\widehat{\mu}_k^{(N)}$ therefore lifts the expectation with respect to the discrete measure $\mu_{\mathscr{X}}$ to a differentiable functional of $\zeta^{\mu}$, which in turn pushes forward $\widehat{\mu}_k^{(N)}$ onto the space $\mathcal{X}$. $\square$

The following proposition is crucial to controlling the evolution of the conjugate functional as well as the proximal distributions over time.

**Proposition C.12.** *Suppose $\zeta^{\mu}/B_k, \tilde{\zeta}^{\mu}/B_k$ are $M_{\mu}$-Lipschitz functions such that the difference $\zeta^{\mu} - \tilde{\zeta}^{\mu}$ is $m_{\mu}$-Lipschitz for some $m_{\mu} > 0$. Then the projected proximal distributions satisfy*

$$W_2(\Pi \widehat{\mu}_k^{(N)}(\zeta^{\mu}), \Pi \widehat{\mu}_k^{(N)}(\tilde{\zeta}^{\mu})) \le \frac{m_{\mu}}{\alpha_{\mu} \lambda B_k}.$$

*Proof.* Taking the first-order conditions

$$-\int_{\mathcal{X}} \zeta^\mu (\mu_{\mathscr{X}} - \rho^\mu)(\mathrm{d}x) - \frac{\lambda B_k}{N} \log \frac{\widehat{\mu}_k^{(N)}(\zeta^\mu)}{\rho^{\mu \otimes N}} = \text{const.},$$

$$-\int_{\mathcal{X}} \tilde{\zeta}^\mu (\mu_{\mathscr{X}} - \rho^\mu)(\mathrm{d}x) - \frac{\lambda B_k}{N} \log \frac{\widehat{\mu}_k^{(N)}(\tilde{\zeta}^\mu)}{\rho^{\mu \otimes N}} = \text{const.}$$

Subtracting both sides and integrating over the difference $\widehat{\mu}_k^{(N)}(\zeta^\mu) - \widehat{\mu}_k^{(N)}(\tilde{\zeta}^\mu)$, we obtain

$$-\int_{\mathcal{X}^N} \int_{\mathcal{X}} (\zeta^\mu - \tilde{\zeta}^\mu)\mu_{\mathscr{X}}(\mathrm{d}x)(\widehat{\mu}_k^{(N)}(\zeta^\mu) - \widehat{\mu}_k^{(N)}(\tilde{\zeta}^\mu))(\mathrm{d}\mathscr{X})$$

$$= \frac{\lambda B_k}{N} \int_{\mathcal{X}^N} \log \frac{\widehat{\mu}_k^{(N)}(\zeta^\mu)}{\widehat{\mu}_k^{(N)}(\tilde{\zeta}^\mu)}(\widehat{\mu}_k^{(N)}(\zeta^\mu) - \widehat{\mu}_k^{(N)}(\tilde{\zeta}^\mu))(\mathrm{d}\mathscr{X}). \tag{10}$$

Now the left-hand side of (10) can be bounded from above by

$$-\int_{\mathcal{X}^N} \int_{\mathcal{X}} (\zeta^\mu - \tilde{\zeta}^\mu)\mu_{\mathscr{X}}(\mathrm{d}x)(\widehat{\mu}_k^{(N)}(\zeta^\mu) - \widehat{\mu}_k^{(N)}(\tilde{\zeta}^\mu))(\mathrm{d}\mathscr{X})$$

$$= -\int_{\mathcal{X}} (\zeta^\mu - \tilde{\zeta}^\mu)(\Pi \widehat{\mu}_k^{(N)}(\zeta^\mu) - \Pi \widehat{\mu}_k^{(N)}(\tilde{\zeta}^\mu))(\mathrm{d}x)$$

$$\leq m_\mu W_1(\Pi \widehat{\mu}_k^{(N)}(\zeta^\mu), \Pi \widehat{\mu}_k^{(N)}(\tilde{\zeta}^\mu)) \leq m_\mu W_2(\Pi \widehat{\mu}_k^{(N)}(\zeta^\mu), \Pi \widehat{\mu}_k^{(N)}(\tilde{\zeta}^\mu)),$$

while the right-hand side of (10) is bounded from below by

$$\frac{\lambda B_k}{N} \left( \text{KL}(\widehat{\mu}_k^{(N)}(\zeta^\mu) \| \widehat{\mu}_k^{(N)}(\tilde{\zeta}^\mu)) + \text{KL}(\widehat{\mu}_k^{(N)}(\tilde{\zeta}^\mu) \| \widehat{\mu}_k^{(N)}(\zeta^\mu)) \right)$$

$$\geq \frac{\alpha_\mu \lambda B_k}{N} W_2^2(\widehat{\mu}_k^{(N)}(\zeta^\mu), \widehat{\mu}_k^{(N)}(\tilde{\zeta}^\mu))$$

$$\geq \alpha_\mu \lambda B_k W_2^2(\Pi \widehat{\mu}_k^{(N)}(\zeta^\mu), \Pi \widehat{\mu}_k^{(N)}(\tilde{\zeta}^\mu)),$$

where we have used Talagrand's inequality from Lemma C.10 and the $W_2$ pushforward bound from Lemma C.7. Combining the two results yields the desired statement. $\qquad\square$

Denote the unnormalized aggregate derivatives as

$$\delta_k^\mu = \sum_{j=1}^k \beta_j \frac{\delta \mathcal{L}}{\delta \mu}(\mu_{\mathscr{X}_j}, \nu_{\mathscr{Y}_j}), \quad \delta_k^\nu = \sum_{j=1}^k \beta_j \frac{\delta \mathcal{L}}{\delta \nu}(\mu_{\mathscr{X}_j}, \nu_{\mathscr{Y}_j})$$

so that $\widehat{\mu}_k^{(N)} = \widehat{\mu}_k^{(N)}(\delta_k^\mu), \widehat{\nu}_k^{(N)} = \widehat{\nu}_k^{(N)}(\delta_k^\nu)$. The functions $\delta_k^\mu / B_k$ and $\delta_k^\nu / B_k$ are $M_\mu$- and $M_\nu$-Lipschitz, respectively, due to Assumption 2. Lemma C.11 and Proposition C.12 then allow us to quantify the change in $\widehat{J}_k^{(N)}(\delta_k^\mu, \delta_k^\nu)$ as time progresses.

**Lemma C.13.** *We have the following one-step relation for $\widehat{J}_k^{(N)}$, $k \geq 2$:*

$$\widehat{J}_k^{(N)}(\delta_k^\mu, \delta_k^\nu) - \widehat{J}_{k-1}^{(N)}(\delta_{k-1}^\mu, \delta_{k-1}^\nu)$$

$$\leq \beta_k \int_{\mathcal{X}} \frac{\delta \mathcal{L}}{\delta \mu}(\mu_{\mathscr{X}_k}, \nu_{\mathscr{Y}_k})(-\Pi \widehat{\mu}_{k-1}^{(N)} + \rho^\mu)(\mathrm{d}x) + \beta_k \int_{\mathcal{Y}} \frac{\delta \mathcal{L}}{\delta \nu}(\mu_{\mathscr{X}_k}, \nu_{\mathscr{Y}_k})(\Pi \widehat{\nu}_{k-1}^{(N)} - \rho^\nu)(\mathrm{d}y)$$

$$- \frac{\lambda \beta_k}{N} \left( \text{KL}(\widehat{\mu}_k^{(N)} \| \rho^{\mu \otimes N}) + \text{KL}(\widehat{\nu}_k^{(N)} \| \rho^{\nu \otimes N}) \right) + \left( \frac{M_\mu^2}{\alpha_\mu} + \frac{M_\nu^2}{\alpha_\nu} \right) \frac{\beta_k^2}{2\lambda B_{k-1}}.$$

*Proof.* By the maximality of $\widehat{J}_k^{(N)}$,

$$\widehat{J}_k^{(N)}(\delta_k^\mu, \delta_k^\nu) = J_k^{(N)}(\widehat{\mu}_k^{(N)}(\delta_k^\mu), \widehat{\mu}_k^{(N)}(\delta_k^\nu) | \delta_k^\mu, \delta_k^\nu)$$

$$= J_{k-1}^{(N)}(\widehat{\mu}_k^{(N)}(\delta_k^\mu), \widehat{\mu}_k^{(N)}(\delta_k^\nu) | \delta_k^\mu, \delta_k^\nu) - \frac{\lambda \beta_k}{N} \left( \text{KL}(\widehat{\mu}_k^{(N)}(\delta_k^\mu) \| \rho^{\mu \otimes N}) + \text{KL}(\widehat{\nu}_k^{(N)}(\delta_k^\nu) \| \rho^{\nu \otimes N}) \right)$$

$$\leq \widehat{J}_{k-1}^{(N)}(\delta_k^\mu, \delta_k^\nu) - \frac{\lambda\beta_k}{N}\left(\mathrm{KL}(\widehat{\mu}_k^{(N)}\|\rho^{\mu\otimes N}) + \mathrm{KL}(\widehat{\nu}_k^{(N)}\|\rho^{\nu\otimes N})\right).$$

Further defining the interpolations

$$\delta_k^\mu(s) = \delta_{k-1}^\mu + s(\delta_k^\mu - \delta_{k-1}^\mu) = \sum_{j=1}^{k-1}\beta_j\frac{\delta\mathcal{L}}{\delta\mu}(\mu_{\mathscr{X}_j}, \nu_{\mathscr{Y}_j}) + s\beta_k\frac{\delta\mathcal{L}}{\delta\mu}(\mu_{\mathscr{X}_k}, \nu_{\mathscr{Y}_k}), \quad 0 \leq s \leq 1$$

and similarly for $\delta_k^\nu(s)$, we have

$$\widehat{J}_{k-1}^{(N)}(\delta_k^\mu, \delta_k^\nu) - \widehat{J}_{k-1}^{(N)}(\delta_{k-1}^\mu, \delta_{k-1}^\nu) = \int_0^1 \frac{\mathrm{d}}{\mathrm{d}s}\widehat{J}_{k-1}^{(N)}(\delta_k^\mu(s), \delta_k^\nu(s))\,\mathrm{d}s$$

$$= \int_0^1 \int_{\mathcal{X}}(\delta_k^\mu - \delta_{k-1}^\mu)\frac{\delta\widehat{J}_{k-1}^{(N)}}{\delta\zeta^\mu}(\delta_k^\mu(s), \delta_k^\nu(s))(\mathrm{d}x) + \int_{\mathcal{Y}}(\delta_k^\nu - \delta_{k-1}^\nu)\frac{\delta\widehat{J}_{k-1}^{(N)}}{\delta\zeta^\nu}(\delta_k^\mu(s), \delta_k^\nu(s))(\mathrm{d}y)\,\mathrm{d}s$$

$$= \int_0^1 \int_{\mathcal{X}}-(\delta_k^\mu - \delta_{k-1}^\mu)\Pi\,\widehat{\mu}_{k-1}^{(N)}(\delta_k^\mu(s))(\mathrm{d}x) + \int_{\mathcal{Y}}(\delta_k^\nu - \delta_{k-1}^\nu)\Pi\,\widehat{\nu}_{k-1}^{(N)}(\delta_k^\nu(s))(\mathrm{d}y)\,\mathrm{d}s$$

$$+ \int_{\mathcal{X}}(\delta_k^\mu - \delta_{k-1}^\mu)\rho^\mu(\mathrm{d}x) - \int_{\mathcal{Y}}(\delta_k^\nu - \delta_{k-1}^\nu)\rho^\nu(\mathrm{d}y)$$

$$\leq \int_0^1 \int_{\mathcal{X}}-(\delta_k^\mu - \delta_{k-1}^\mu)\Pi\,\widehat{\mu}_{k-1}^{(N)}(\delta_{k-1}^\mu)(\mathrm{d}x) + \int_{\mathcal{Y}}(\delta_k^\nu - \delta_{k-1}^\nu)\Pi\,\widehat{\nu}_{k-1}^{(N)}(\delta_{k-1}^\nu)(\mathrm{d}y)\,\mathrm{d}s$$

$$+ \int_{\mathcal{X}}(\delta_k^\mu - \delta_{k-1}^\mu)\rho^\mu(\mathrm{d}x) - \int_{\mathcal{Y}}(\delta_k^\nu - \delta_{k-1}^\nu)\rho^\nu(\mathrm{d}y)$$

$$+ \int_0^1 M_\mu\beta_k W_1(\Pi\,\widehat{\mu}_{k-1}^{(N)}(\delta_k^\mu(s)), \Pi\,\widehat{\mu}_{k-1}^{(N)}(\delta_{k-1}^\mu))\,\mathrm{d}s$$

$$+ \int_0^1 M_\nu\beta_k W_1(\Pi\,\widehat{\nu}_{k-1}^{(N)}(\delta_k^\nu(s)), \Pi\,\widehat{\nu}_{k-1}^{(N)}(\delta_{k-1}^\nu))\,\mathrm{d}s$$

$$\leq \beta_k\int_{\mathcal{X}}-\frac{\delta\mathcal{L}}{\delta\mu}(\mu_{\mathscr{X}_k}, \nu_{\mathscr{Y}_k})\Pi\,\widehat{\mu}_{k-1}^{(N)}(\mathrm{d}x) + \beta_k\int_{\mathcal{Y}}\frac{\delta\mathcal{L}}{\delta\nu}(\mu_{\mathscr{X}_k}, \nu_{\mathscr{Y}_k})\Pi\,\widehat{\nu}_{k-1}^{(N)}(\mathrm{d}y)$$

$$+ \beta_k\int_{\mathcal{X}}\frac{\delta\mathcal{L}}{\delta\mu}(\mu_{\mathscr{X}_k}, \nu_{\mathscr{Y}_k})\rho^\mu(\mathrm{d}x) - \beta_k\int_{\mathcal{Y}}\frac{\delta\mathcal{L}}{\delta\nu}(\mu_{\mathscr{X}_k}, \nu_{\mathscr{Y}_k})\rho^\nu(\mathrm{d}y) + \left(\frac{M_\mu^2}{\alpha_\mu} + \frac{M_\nu^2}{\alpha_\nu}\right)\frac{\beta_k^2}{2\lambda B_{k-1}},$$

where for the first inequality we used the fact that $\delta_k^\mu - \delta_{k-1}^\mu$ is $M_\mu\beta_k$-Lipschitz, and for the second we applied Proposition C.12 with $m_\mu = sM_\mu\beta_k$. $\qquad\square$

We now give the promised proof of the pushforward evolution bound.

**Proof of Lemma C.8.** Note that $\widehat{\mu}_{k-1}^{(N)} = \widehat{\mu}_{k-1}^{(N)}(\delta_{k-1}^\mu)$ may also be written as

$$\widehat{\mu}_{k-1}^{(N)} = \widehat{\mu}_k^{(N)}\left(\frac{B_k}{B_{k-1}}\sum_{j=1}^k\beta_j\frac{\delta\mathcal{L}}{\delta\mu}(\mu_{\mathscr{X}_j}, \nu_{\mathscr{Y}_j})\right) = \widehat{\mu}_k^{(N)}\left(\frac{B_k}{B_{k-1}}\delta_{k-1}^\mu\right).$$

Since $\delta_{k-1}^\mu/B_{k-1}$ is $M_\mu$-Lipschitz and

$$\delta_k^\mu - \frac{B_k}{B_{k-1}}\delta_{k-1}^\mu = -\frac{\beta_k}{B_{k-1}}\sum_{j=1}^{k-1}\beta_j\frac{\delta\mathcal{L}}{\delta\mu}(\mu_{\mathscr{X}_j}, \nu_{\mathscr{Y}_j}) + \beta_k\frac{\delta\mathcal{L}}{\delta\mu}(\mu_{\mathscr{X}_k}, \nu_{\mathscr{Y}_k})$$

is $2M_\mu\beta_k$-Lipschitz, by Proposition C.12 we obtain the bound

$$W_2(\Pi\,\widehat{\mu}_k^{(N)}, \Pi\,\widehat{\mu}_{k-1}^{(N)}) \leq \frac{2M_\mu\beta_k}{\alpha_\mu\lambda B_k}.$$

$\qquad\square$

## C.5 PROOF OF THEOREM 3.7

*Step 1.* We first prove a convergent upper bound of the following surrogate $\mathfrak{N}(\mu_{\overline{\mathscr{X}}_k}, \nu_{\overline{\mathscr{Y}}_k})$ for the NI error of the average distributions. Note that the defining maximum is lifted to the configuration space and the discrete empirical distributions have been replaced with their proximal counterparts for measuring relative entropy. While $\mathfrak{N}$ is not exactly the desired quantity, it arises naturally from the discrete conjugate argument and helps to bound the expected error.

$$\mathfrak{N}(\mu_{\overline{\mathscr{X}}_k}, \nu_{\overline{\mathscr{Y}}_k})$$

$$:= \max_{\mu^{(N)}, \nu^{(N)}} -\frac{1}{B_k} \sum_{j=1}^{k} \beta_j \mathcal{L}(\Pi\mu^{(N)}, \nu_{\mathscr{Y}_j}) - \frac{\lambda}{N} \mathrm{KL}(\mu^{(N)} \| \rho^{\mu \otimes N}) + \frac{\lambda}{NB_k} \sum_{j=1}^{k} \beta_j \mathrm{KL}(\widehat{\nu}_j^{(N)} \| \rho^{\nu \otimes N})$$

$$+ \frac{1}{B_k} \sum_{j=1}^{k} \beta_j \mathcal{L}(\mu_{\mathscr{X}_j}, \Pi\nu^{(N)}) - \frac{\lambda}{N} \mathrm{KL}(\nu^{(N)} \| \rho^{\nu \otimes N}) + \frac{\lambda}{NB_k} \sum_{j=1}^{k} \beta_j \mathrm{KL}(\widehat{\mu}_j^{(N)} \| \rho^{\mu \otimes N})$$

$$\leq \max_{\mu^{(N)}, \nu^{(N)}} -\int_{\mathcal{X}^N} \int_{\mathcal{X}} \frac{1}{B_k} \sum_{j=1}^{k} \beta_j \frac{\delta\mathcal{L}}{\delta\mu}(\mu_{\mathscr{X}_j}, \nu_{\mathscr{Y}_j})(\mu_{\mathscr{X}} - \mu_{\mathscr{X}_j})(\mathrm{d}x)\mu^{(N)}(\mathrm{d}\mathscr{X})$$

$$+ \int_{\mathcal{Y}^N} \int_{\mathcal{Y}} \frac{1}{B_k} \sum_{j=1}^{k} \beta_j \frac{\delta\mathcal{L}}{\delta\nu}(\mu_{\mathscr{X}_j}, \nu_{\mathscr{Y}_j})(\nu_{\mathscr{Y}} - \nu_{\mathscr{Y}_j})(\mathrm{d}y)\nu^{(N)}(\mathrm{d}\mathscr{Y})$$

$$- \frac{\lambda}{N} \left( \mathrm{KL}(\mu^{(N)} \| \rho^{\mu \otimes N}) + \mathrm{KL}(\nu^{(N)} \| \rho^{\nu \otimes N}) \right)$$

$$+ \frac{\lambda}{NB_k} \sum_{j=1}^{k} \beta_j \left( \mathrm{KL}(\widehat{\mu}_j^{(N)} \| \rho^{\mu \otimes N}) + \mathrm{KL}(\widehat{\nu}_j^{(N)} \| \rho^{\nu \otimes N}) \right)$$

$$= \frac{1}{B_k} \left[ \widehat{\mathcal{J}}_k^{(N)}(\delta_k^\nu, \delta_k^\nu) + \frac{\lambda}{N} \sum_{j=1}^{k} \beta_j \left( \mathrm{KL}(\widehat{\mu}_j^{(N)} \| \rho^{\mu \otimes N}) + \mathrm{KL}(\widehat{\nu}_j^{(N)} \| \rho^{\nu \otimes N}) \right) \right.$$

$$\left. + \int_{\mathcal{X}} \sum_{j=1}^{k} \beta_j \frac{\delta\mathcal{L}}{\delta\mu}(\mu_{\mathscr{X}_j}, \nu_{\mathscr{Y}_j})(\mu_{\mathscr{X}_j} - \rho^\mu)(\mathrm{d}x) - \int_{\mathcal{Y}} \sum_{j=1}^{k} \beta_j \frac{\delta\mathcal{L}}{\delta\nu}(\mu_{\mathscr{X}_j}, \nu_{\mathscr{Y}_j})(\nu_{\mathscr{Y}_j} - \rho^\nu)(\mathrm{d}y) \right],$$

due to the convex-concavity of $\mathcal{L}$. Recursively applying Lemma C.13 then yields

$$\mathfrak{N}(\mu_{\overline{\mathscr{X}}_k}, \nu_{\overline{\mathscr{Y}}_k})$$

$$\leq \frac{1}{B_k} \left[ \sum_{j=2}^{k} \left( \widehat{\mathcal{J}}_j^{(N)}(\delta_j^\nu, \delta_j^\nu) - \widehat{\mathcal{J}}_{j-1}^{(N)}(\delta_{j-1}^\nu, \delta_{j-1}^\nu) \right) + \frac{1}{B_k} \widehat{\mathcal{J}}_1^{(N)}(\delta_1^\nu, \delta_1^\nu) \right.$$

$$+ \int_{\mathcal{X}} \sum_{j=1}^{k} \beta_j \frac{\delta\mathcal{L}}{\delta\mu}(\mu_{\mathscr{X}_j}, \nu_{\mathscr{Y}_j})(\mu_{\mathscr{X}_j} - \rho^\mu)(\mathrm{d}x) - \int_{\mathcal{Y}} \sum_{j=1}^{k} \beta_j \frac{\delta\mathcal{L}}{\delta\nu}(\mu_{\mathscr{X}_j}, \nu_{\mathscr{Y}_j})(\nu_{\mathscr{Y}_j} - \rho^\nu)(\mathrm{d}y)$$

$$\left. + \frac{\lambda}{N} \sum_{j=1}^{k} \beta_j \left( \mathrm{KL}(\widehat{\mu}_j^{(N)} \| \rho^{\mu \otimes N}) + \mathrm{KL}(\widehat{\nu}_j^{(N)} \| \rho^{\nu \otimes N}) \right) \right]$$

$$\leq \frac{1}{B_k} \left[ \sum_{j=1}^{k} \beta_j \int_{\mathcal{X}} \frac{\delta\mathcal{L}}{\delta\mu}(\mu_{\mathscr{X}_j}, \nu_{\mathscr{Y}_j})(\mu_{\mathscr{X}_j} - \Pi\widehat{\mu}_{j-1}^{(N)})(\mathrm{d}x) \right.$$

$$\left. - \sum_{j=1}^{k} \beta_j \int_{\mathcal{Y}} \frac{\delta\mathcal{L}}{\delta\nu}(\mu_{\mathscr{X}_j}, \nu_{\mathscr{Y}_j})(\nu_{\mathscr{Y}_j} - \Pi\widehat{\nu}_{j-1}^{(N)})(\mathrm{d}y) + \frac{1}{2\lambda} \left( \frac{M_\mu^2}{\alpha_\mu} + \frac{M_\nu^2}{\alpha_\nu} \right) \sum_{j=2}^{k} \frac{\beta_j^2}{B_{j-1}} \right],$$

where the initial term is substituted as $\widehat{\mathcal{J}}_1^{(N)}(\delta_1^\nu, \delta_1^\nu) = \mathcal{J}_1^{(N)}(\widehat{\mu}_1^{(N)}, \widehat{\nu}_1^{(N)} | \delta_1^\nu, \delta_1^\nu)$ with the convention that $\widehat{\mu}_0^{(N)} = \widehat{\mu}_1^{(N)}, \widehat{\nu}_0^{(N)} = \widehat{\nu}_1^{(N)}$. Now taking the expectation over the full history and applying Proposition 3.6, we arrive at

$$\mathbb{E}_{(\mathscr{X}, \mathscr{Y})_{1:k}} \left[ \mathfrak{N}(\mu_{\overline{\mathscr{X}}_k}, \nu_{\overline{\mathscr{Y}}_k}) \right]$$

$$\leq \frac{1}{B_k}\mathbb{E}_{(\mathscr{X},\mathscr{Y})_{1:k}}\left[\sum_{j=1}^k \beta_j \int_{\mathcal{X}} \frac{\delta\mathcal{L}}{\delta\mu}(\mu_{\mathscr{X}_j},\nu_{\mathscr{Y}_j})(\mu_{\mathscr{X}_j} - \Pi\,\widehat{\mu}_{j-1}^{(N)})(\mathrm{d}x)\right.$$

$$\left. - \sum_{j=1}^k \beta_j \int_{\mathcal{Y}} \frac{\delta\mathcal{L}}{\delta\nu}(\mu_{\mathscr{X}_j},\nu_{\mathscr{Y}_j})(\nu_{\mathscr{Y}_j} - \Pi\,\widehat{\nu}_{j-1}^{(N)})(\mathrm{d}y) + \frac{1}{2\lambda}\left(\frac{M_\mu^2}{\alpha_\mu} + \frac{M_\nu^2}{\alpha_\nu}\right)\sum_{j=2}^k \frac{\beta_j^2}{B_{j-1}}\right]$$

$$\leq \frac{1}{B_k}\left[2\sum_{j=1}^k \beta_j\left(\frac{r+1}{j}C_1(\eta) + C_2\sqrt{\eta} + \frac{C_3}{\sqrt{N}}\right) + \frac{1}{2\lambda}\left(\frac{M_\mu^2}{\alpha_\mu} + \frac{M_\nu^2}{\alpha_\nu}\right)\sum_{j=2}^k \frac{\beta_j^2}{B_{j-1}}\right]$$

$$\leq \left(\frac{(r+1)^2}{rk} + O(k^{-2})\right)\left(2C_1(\eta) + \frac{1}{2\lambda}\left(\frac{M_\mu^2}{\alpha_\mu} + \frac{M_\nu^2}{\alpha_\nu}\right)\right) + 2C_2\sqrt{\eta} + \frac{2C_3}{\sqrt{N}}$$

$$\leq \left(\frac{(r+1)^2}{rk} + O(k^{-2})\right)\cdot\frac{9}{4}C_1(\eta) + 2C_2\sqrt{\eta} + \frac{2C_3}{\sqrt{N}}$$

by simply using $\ell > 1$. For $r = 0$, the last expression is replaced by the exact bound

$$\frac{1+\log k}{k}\cdot\frac{9}{4}C_1(\eta) + 2C_2\sqrt{\eta} + \frac{2C_3}{\sqrt{N}}.$$

*Step 2.* We now control the NI error of the averaged pushforward proximal distributions using $\mathfrak{N}$. In the defining maximum over $\mu^{(N)} \in \mathcal{P}_2(\mathcal{X}^N), \nu^{(N)} \in \mathcal{P}_2(\mathcal{Y}^N)$, we may restrict to product distributions $\mu^{(N)} = \mu^{\otimes N}, \nu^{(N)} = \nu^{\otimes N}$ so that

$$\mathbb{E}_{(\mathscr{X},\mathscr{Y})_{1:k}}\left[\mathfrak{N}(\mu_{\overline{\mathscr{X}}_k}, \nu_{\overline{\mathscr{Y}}_k})\right]$$

$$\geq \mathbb{E}_{(\mathscr{X},\mathscr{Y})_{1:k}}\left[\max_{\mu,\nu} -\frac{1}{B_k}\sum_{j=1}^k \beta_j\mathcal{L}(\mu,\nu_{\mathscr{Y}_j}) - \lambda\,\mathrm{KL}(\mu\|\rho^\mu) + \frac{\lambda}{B_k}\sum_{j=1}^k \beta_j\,\mathrm{KL}(\Pi\,\widehat{\nu}_j^{(N)}\|\rho^\nu)\right.$$

$$\left. + \frac{1}{B_k}\sum_{j=1}^k \beta_j\mathcal{L}(\mu_{\mathscr{X}_j},\nu) - \lambda\,\mathrm{KL}(\nu\|\rho^\nu) + \frac{\lambda}{B_k}\sum_{j=1}^k \beta_j\,\mathrm{KL}(\Pi\,\widehat{\mu}_j^{(N)}\|\rho^\mu)\right]$$

$$\geq \max_{\mu,\nu}\mathbb{E}_{(\mathscr{X},\mathscr{Y})_{1:k}}\left[-\frac{1}{B_k}\sum_{j=1}^k \beta_j\mathcal{L}(\mu,\nu_{\mathscr{Y}_j}) - \lambda\,\mathrm{KL}(\mu\|\rho^\mu) + \frac{\lambda}{B_k}\sum_{j=1}^k \beta_j\,\mathrm{KL}(\Pi\,\widehat{\nu}_j^{(N)}\|\rho^\nu)\right.$$

$$\left. + \frac{1}{B_k}\sum_{j=1}^k \beta_j\mathcal{L}(\mu_{\mathscr{X}_j},\nu) - \lambda\,\mathrm{KL}(\nu\|\rho^\nu) + \frac{\lambda}{B_k}\sum_{j=1}^k \beta_j\,\mathrm{KL}(\Pi\,\widehat{\mu}_j^{(N)}\|\rho^\mu)\right]$$

$$\geq \max_{\mu,\nu} -\mathcal{L}(\mu, \mathbb{E}[\nu_{\overline{\mathscr{Y}}_k}]) - \lambda\,\mathrm{KL}(\mu\|\rho^\mu) + \lambda\,\mathrm{KL}(\mathbb{E}[\overline{\Pi}\,\widehat{\nu}_k]\|\rho^\nu)$$

$$+ \mathcal{L}(\mathbb{E}[\mu_{\overline{\mathscr{X}}_k}], \nu) - \lambda\,\mathrm{KL}(\nu\|\rho^\nu) + \lambda\,\mathrm{KL}(\mathbb{E}[\overline{\Pi}\,\widehat{\mu}_k]\|\rho^\mu)$$

by convex-concavity of $\mathcal{L}$ as well as convexity of KL divergence, where we have written

$$\overline{\Pi}\,\widehat{\mu}_k := \frac{1}{B_k}\sum_{j=1}^k \beta_j\Pi\,\widehat{\mu}_j^{(N)}, \quad \overline{\Pi}\,\widehat{\nu}_k := \frac{1}{B_k}\sum_{j=1}^k \beta_j\Pi\,\widehat{\nu}_j^{(N)}.$$

Again by Proposition 3.6, this is further bounded as

$$\mathbb{E}_{(\mathscr{X},\mathscr{Y})_{1:k}}\left[\mathfrak{N}(\mu_{\overline{\mathscr{X}}_k}, \nu_{\overline{\mathscr{Y}}_k})\right]$$

$$\geq \max_{\mu,\nu} -\mathcal{L}_\lambda(\mu, \mathbb{E}[\overline{\Pi}\,\widehat{\nu}_k]) - \mathbb{E}_{(\mathscr{X},\mathscr{Y})_{1:k}}\left[\frac{1}{B_k}\int_{\mathcal{Y}}\sum_{j=1}^k \beta_j\frac{\delta\mathcal{L}}{\delta\nu}(\mu, \mathbb{E}[\overline{\Pi}\,\widehat{\nu}_k])(\nu_{\mathscr{Y}_j} - \Pi\,\widehat{\nu}_j^{(N)})(\mathrm{d}y)\right]$$

$$+ \mathcal{L}_\lambda(\mathbb{E}[\overline{\Pi}\,\widehat{\mu}_k], \nu) + \mathbb{E}_{(\mathscr{X},\mathscr{Y})_{1:k}}\left[\frac{1}{B_k}\int_{\mathcal{X}}\sum_{j=1}^k \beta_j\frac{\delta\mathcal{L}}{\delta\mu}(\mathbb{E}[\overline{\Pi}\,\widehat{\mu}_k], \nu)(\mu_{\mathscr{X}_j} - \Pi\,\widehat{\mu}_j^{(N)})(\mathrm{d}x)\right]$$

$$\geq \mathrm{NI}(\mathbb{E}[\overline{\Pi}\,\widehat{\mu}_k], \mathbb{E}[\overline{\Pi}\,\widehat{\nu}_k]) - \left(\frac{(r+1)^2}{rk} + O(k^{-2})\right) \cdot 2C_1(\eta) - 2C_2\sqrt{\eta} - \frac{2C_3}{\sqrt{N}},$$

with the appropriate modification for $r = 0$.

*Step 3.* Finally, we convert the above pushforward proximal bounds back to a Wasserstein distance bound for the expected empirical measures. By Lemma 3.5 and Talagrand's inequality for the MNE $(\mu^*, \nu^*)$,

$$W_2^2(\mathbb{E}[\overline{\Pi}\,\widehat{\mu}_k], \mu^*) + W_2^2(\mathbb{E}[\overline{\Pi}\,\widehat{\nu}_k], \nu^*)$$

$$\leq \frac{2}{\alpha_\mu} \vee \frac{2}{\alpha_\nu} \left(\mathrm{KL}(\mathbb{E}[\overline{\Pi}\,\widehat{\mu}_k]\|\mu^*) + \mathrm{KL}(\mathbb{E}[\overline{\Pi}\,\widehat{\nu}_k]\|\nu^*)\right)$$

$$\leq \frac{2}{\alpha_\mu\lambda} \vee \frac{2}{\alpha_\nu\lambda} \, \mathrm{NI}(\mathbb{E}[\overline{\Pi}\,\widehat{\mu}_k], \mathbb{E}[\overline{\Pi}\,\widehat{\nu}_k])$$

$$\leq \frac{2}{\alpha_\mu\lambda} \vee \frac{2}{\alpha_\nu\lambda} \left[ \left(\frac{(r+1)^2}{rk} + O(k^{-2})\right) \cdot \frac{17}{4}C_1(\eta) + 4C_2\sqrt{\eta} + \frac{4C_3}{\sqrt{N}} \right].$$

Note also by Proposition 3.6 and Lemma C.8 that

$$M_\mu W_1(\mathbb{E}[\mu_{\overline{\mathscr{X}}_k}], \mathbb{E}[\overline{\Pi}\,\widehat{\mu}_k])$$

$$= \sup_{\|F\|_{\mathrm{Lip}} \leq M_\mu} \mathbb{E}_{(\mathscr{X},\mathscr{Y})_{1:k}} \left[ \frac{1}{B_k} \sum_{j=1}^k \beta_j \int_{\mathcal{X}} F(\mu_{\mathscr{X}_j} - \Pi\widehat{\mu}_j^{(N)})(\mathrm{d}x) \right]$$

$$\leq \frac{1}{B_k} \sum_{j=1}^k \beta_j \left(\frac{r+1}{j}C_1(\eta) + C_2\sqrt{\eta} + \frac{C_3}{\sqrt{N}}\right) + \frac{1}{B_k} \sum_{j=2}^k \beta_j W_1(\Pi\widehat{\mu}_j^{(N)}, \Pi\widehat{\mu}_{j-1}^{(N)})$$

$$\leq \frac{1}{B_k} \sum_{j=1}^k \beta_j \left(\frac{r+1}{j}C_1(\eta) + C_2\sqrt{\eta} + \frac{C_3}{\sqrt{N}}\right) + \frac{2M_\mu}{\alpha_\mu\lambda B_k} \sum_{j=2}^k \frac{\beta_j^2}{B_j}$$

$$\leq \left(\frac{(r+1)^2}{rk} + O(k^{-2})\right) \cdot \frac{3}{2}C_1(\eta) + C_2\sqrt{\eta} + \frac{C_3}{\sqrt{N}},$$

so the square of this term can be ignored. Hence we can conclude that

$$W_1^2(\mathbb{E}[\mu_{\overline{\mathscr{X}}_k}], \mu^*) + W_1^2(\mathbb{E}[\nu_{\overline{\mathscr{Y}}_k}], \nu^*) \leq \frac{(r+1)^2}{rk}\widetilde{C}_1(\eta) + \widetilde{C}_2\sqrt{\eta} + \frac{\widetilde{C}_3}{\sqrt{N}},$$

again with the $1 + \log k$ modification when $r = 0$. $\square$

## C.6   Expected Wasserstein Distance

Theorem 3.7 gives error bounds for the expected distributions $\mathbb{E}[\mu_{\overline{\mathscr{X}}_k}]$ and $\mathbb{E}[\nu_{\overline{\mathscr{Y}}_k}]$. This quantifies a sort of bias of the MFL-AG outputs, but does not tell us anything about the variance. Can we similarly bound the expected distance $\mathbb{E}[W_1(\mu_{\overline{\mathscr{X}}_k}, \mu^*) + W_1(\nu_{\overline{\mathscr{Y}}_k}, \nu^*)]$ of the empirical distributions to the MNE? The following fundamental fact about Wasserstein distance tells us that this is impossible:

**Theorem C.14** (Rate of convergence of the empirical measure, adapted from Fournier & Guillin (2015), Theorem 1). *Let $X^i$ be independent samples drawn from $\mu^i \in \mathcal{P}_2(\mathbb{R}^d)$ for each $i \in [N]$. If $d \geq 3$, the 1-Wasserstein distance between the empirical measure $\mu_{\mathscr{X}} = \frac{1}{N}\sum_{i=1}^N \delta_{X^i}$ and the underlying averaged measure $\mu = \frac{1}{N}\sum_{i=1}^N \mu^i$ is bounded in expectation as*

$$\mathbb{E}[W_1(\mu_{\mathscr{X}}, \mu)] \leq C_W \sqrt{\mathfrak{m}_2(\mu)} \cdot N^{-1/d},$$

*where $\mathfrak{m}_2(\mu)$ is the raw second moment of $\mu$ and $C_W$ is a universal constant. If $d = 2$, the rate is $O(N^{-1/2}(\log N)^2)$; if $d = 1$, the rate is $O(N^{-1/2})$. Furthermore, this rate is tight up to constants.*

*Proof.* The original theorem only considers i.i.d. samples $\mu^1 = \cdots = \mu^N = \mu$ and omits the $W_1$ case for simplicity, so we present the necessary modifications.

For a Borel subset $A \subset \mathbb{R}^d$, the quantity $N\mu_{\mathscr{X}}(A)$ is not distributed as Binomial$(N, \mu(A))$ but as a sum of independent Bernoulli$(\mu^i(A))$ random variables. Nonetheless, we obtain the same bound

$$\mathbb{E}[|\mu_{\mathscr{X}}(A) - \mu(A)|] \leq (\mathbb{E}[\mu_{\mathscr{X}}(A)] + \mu(A)) \wedge \sqrt{\operatorname{Var}\mu_{\mathscr{X}}(A)}$$
$$\leq 2\mu(A) \wedge \sqrt{\mu(A)/N}.$$

We now repeat the same arguments and substitute $p = 1, q = 2$ to arrive at the following inequality,

$$\mathbb{E}[W_1(\mu_{\mathscr{X}}, \mu)] \leq C\sqrt{\mathfrak{m}_2(\mu)} \cdot \sum_{n=0}^{\infty} \sum_{m=0}^{\infty} 2^{-m}(2^{-n} \wedge (2^{dm}/N)^{1/2})$$

from which point we give explicit computations. Defining

$$m_N = \left\lceil \frac{\log_2 N}{d} \right\rceil, \quad n_m = \left\lceil \frac{\log_2 N - dm}{2} \right\rceil,$$

we have for $d \geq 3$ that

$$\sum_{n=0}^{\infty} \sum_{m=0}^{\infty} 2^{-m}(2^{-n} \wedge (2^{dm}/N)^{1/2})$$
$$= \sum_{m=0}^{m_N-1} 2^{-m} n_m (2^{dm}/N)^{1/2} + \sum_{m=0}^{m_N-1} \sum_{n=n_m}^{\infty} 2^{-m-n} + \sum_{m=m_N}^{\infty} \sum_{n=0}^{\infty} 2^{-m-n}$$
$$\leq \frac{1}{2\sqrt{N}} \sum_{m=0}^{m_N-1} (dm_N - dm + 2)2^{(d/2-1)m} + \sum_{m=0}^{m_N-1} 2^{1-m-n_m} + 2^{2-m_N}$$
$$\leq \frac{(2+d)2^{(d/2-1)(m_N+1)}}{(2^{d/2}-2)\sqrt{N}} + \frac{2^{2+(d/2-1)m_N}}{(2^{d/2}-2)\sqrt{N}} + 2^{2-m_N}$$
$$= O(N^{-1/d}).$$

When $d = 2$, the rate is easily checked to be $N^{-1/2}(\log N)^2$. The tight rate in one dimension is derived using different techniques in Bobkov & Ledoux (2016), Section 3. □

That is, even in the ideal case where chaos does not propagate and the particles are somehow i.i.d. sampled directly from the true distribution, the expected Wasserstein distance will always be of order $N^{-1/d_{\mathscr{X}} \vee d_{\mathscr{Y}}}$, automatically incurring the curse of dimensionality. We emphasize that the uniform law of large numbers and short-term perturbation methods developed throughout Section C as well as the presentation of Theorem 3.7 have been carefully designed to bypass this technicality.

Nevertheless, it is still possible to bound the expected Wasserstein distance in a similar manner save for the unavoidable $N^{-1/d_{\mathscr{X}} \vee d_{\mathscr{Y}}}$ dependency.[2] We first present a more direct bound for the proximal gap.

**Proposition C.15.** *The following inequality holds for all $k$,*

$$\mathbb{E}\left[W_1(\mu_{\mathscr{X}_k}, \Pi\widehat{\mu}_k^{(N)})\right] \leq \frac{r+1}{k}C_1'(\eta) + C_2'\sqrt{\eta} + C_3'N^{-1/d_{\mathscr{X}}}.$$

*Proof.* The derivations are similar and more straightforward compared to the proof of Proposition 3.6. We only look at $k \geq 2\ell$ and directly compare $\mu_{\mathscr{X}_k}$ to $\mu_{\widetilde{\mathscr{X}}_k}$ using Lemma C.3, $\mu_{\widetilde{\mathscr{X}}_k}$ to the expected modified distribution using Theorem C.14 (recall that the modified particle trajectories $\widetilde{X}_k^i$ are independent when conditioned on $(\mathscr{X}, \mathscr{Y})_{1:k-\ell}$), the expected modified distribution to the stationary distribution $\Pi\widehat{\mu}_{k-\ell}^{(N)}$ using Proposition C.6, and $\Pi\widehat{\mu}_{k-\ell}^{(N)}$ back to $\Pi\widehat{\mu}_k^{(N)}$ using Lemma C.8.

$$\mathbb{E}[W_1(\mu_{\mathscr{X}_k}, \Pi\widehat{\mu}_k^{(N)})]$$
$$\leq \mathbb{E}[W_2(\mu_{\mathscr{X}_k}, \mu_{\widetilde{\mathscr{X}}_k})] + \mathbb{E}[W_1(\mu_{\widetilde{\mathscr{X}}_k}, \mathbb{E}[\mu_{\widetilde{\mathscr{X}}_k}])]$$

---

[2]Of course, we may also simply run the algorithm multiple ($M$) times and take the average of the outputs, which would also bypass the issue and yield the standard $1/\sqrt{M}$ convergence.

$$+ \frac{1}{N}\sum_{i=1}^{N}\mathbb{E}[W_2(\mu_k^i, \Pi\,\widehat{\mu}_{k-\ell}^{(N)})] + \sum_{j=0}^{\ell-1}\mathbb{E}[W_2(\Pi\,\widehat{\mu}_{k-j-1}^{(N)}, \Pi\,\widehat{\mu}_{k-j}^{(N)})]$$

$$\leq \frac{r+1}{k-\ell+1}\mathfrak{w}_\ell^\mu + C_W\sqrt{\mathbb{E}[\mathfrak{m}_2(\mu_{\widetilde{\mathscr{X}}_{k-\ell}})]}\cdot N^{-1/d_{\mathcal{X}}}$$

$$+ \frac{1}{N}\sum_{i=1}^{N}\sqrt{\frac{4}{\alpha_\mu}(\mathfrak{K}^\mu\|X_{k-\ell}^i\|^2 + \mathfrak{L}^\mu)} + \frac{2M_\mu^2}{\alpha_\mu\lambda}\sum_{j=0}^{\ell-1}\frac{\beta_{k-j}}{B_{k-j}}$$

$$\leq \frac{r+1}{k-\ell+1}\mathfrak{w}_\ell^\mu + C_W\sqrt{\mathfrak{p}^\mu + \mathfrak{s}^\mu}\cdot N^{-1/d_{\mathcal{X}}} + \sqrt{\frac{4}{\alpha_\mu}(\mathfrak{K}^\mu(\mathfrak{p}^\mu + \mathfrak{s}^\mu) + \mathfrak{L}^\mu)} + \frac{2M_\mu}{\alpha_\mu\lambda}\frac{(r+1)\ell}{k-\ell+1}$$

$$= \frac{r+1}{k}C_1'(\eta) + C_2'\sqrt{\eta} + C_3'N^{-1/d_{\mathcal{X}}}.$$

$$\square$$

We now give the desired bound for the expected Wasserstein distance to the MNE. Note the effect of dimensionality compared to Theorem 3.7.

**Theorem C.16** (Variance of discretized MFL-AG). *If $\eta \leq \bar{\eta}$ and $\beta_k = k^r$ with $r > 0$, the MFL-AG discrete update satisfies for all $K, N$,*

$$\mathbb{E}[W_1(\mu_{\overline{\mathscr{X}}_K}, \mu^*)]^2 + \mathbb{E}[W_1(\nu_{\overline{\mathscr{Y}}_K}, \nu^*)]^2 \leq \frac{(r+1)^2}{rK}\widetilde{C}_1(\eta) + \widetilde{C}_2\sqrt{\eta} + \widetilde{C}_3 N^{-1/d_{\mathcal{X}}\vee d_{\mathcal{Y}}}$$

*with similar constants as in Proposition 3.6. When $r = 0$, the first term is replaced by $O(\log K/K)$. If $d_{\mathcal{X}} \vee d_{\mathcal{Y}} = 2$, the third term is replaced by $O(N^{-1/2}(\log N)^2)$; if $d_{\mathcal{X}} = d_{\mathcal{Y}} = 1$, by $O(N^{-1/2})$.*

*Proof.* Note that by convexity,

$$\mathcal{L}(\mu_{\mathscr{X}_k}, \nu) - \mathcal{L}(\Pi\,\widehat{\mu}_k^{(N)}, \nu) \geq \int_{\mathcal{X}}\frac{\delta\mathcal{L}}{\delta\mu}(\Pi\,\widehat{\mu}_k^{(N)}, \nu)(\mu_{\mathscr{X}_k} - \Pi\,\widehat{\mu}_k^{(N)})(\mathrm{d}x) \geq -M_\mu W_1(\mu_{\mathscr{X}_k}, \Pi\,\widehat{\mu}_k^{(N)}).$$

We can modify Step 2 of Section C.5 using Proposition C.15 as follows.

$$\mathbb{E}_{(\mathscr{X},\mathscr{Y})_{1:k}}\left[\mathfrak{N}(\mu_{\overline{\mathscr{X}}_k}, \nu_{\overline{\mathscr{Y}}_k})\right]$$

$$\geq \mathbb{E}_{(\mathscr{X},\mathscr{Y})_{1:k}}\left[\max_{\mu,\nu} -\frac{1}{B_k}\sum_{j=1}^{k}\beta_j\mathcal{L}(\mu, \nu_{\mathscr{Y}_j}) - \lambda\,\mathrm{KL}(\mu\|\rho^\mu) + \frac{\lambda}{B_k}\sum_{j=1}^{k}\beta_j\,\mathrm{KL}(\Pi\,\widehat{\nu}_j^{(N)}\|\rho^\nu)\right.$$

$$\left. + \frac{1}{B_k}\sum_{j=1}^{k}\beta_j\mathcal{L}(\mu_{\mathscr{X}_j}, \nu) - \lambda\,\mathrm{KL}(\nu\|\rho^\nu) + \frac{\lambda}{B_k}\sum_{j=1}^{k}\beta_j\,\mathrm{KL}(\Pi\,\widehat{\mu}_j^{(N)}\|\rho^\mu)\right]$$

$$\geq \mathbb{E}_{(\mathscr{X},\mathscr{Y})_{1:k}}\left[\max_{\mu,\nu} -\frac{1}{B_k}\sum_{j=1}^{k}\beta_j\mathcal{L}(\mu, \Pi\,\widehat{\nu}_j^{(N)}) - \lambda\,\mathrm{KL}(\mu\|\rho^\mu) + \frac{\lambda}{B_k}\sum_{j=1}^{k}\beta_j\,\mathrm{KL}(\Pi\,\widehat{\nu}_j^{(N)}\|\rho^\nu)\right.$$

$$+ \frac{1}{B_k}\sum_{j=1}^{k}\beta_j\mathcal{L}(\Pi\,\widehat{\mu}_j^{(N)}, \nu) - \lambda\,\mathrm{KL}(\nu\|\rho^\nu) + \frac{\lambda}{B_k}\sum_{j=1}^{k}\beta_j\,\mathrm{KL}(\Pi\,\widehat{\mu}_j^{(N)}\|\rho^\mu)$$

$$\left. - \frac{M_\mu}{B_k}\sum_{j=1}^{k}\beta_j W_1(\mu_{\mathscr{X}_k}, \Pi\,\widehat{\mu}_k^{(N)}) - \frac{M_\nu}{B_k}\sum_{j=1}^{k}\beta_j W_1(\nu_{\mathscr{Y}_k}, \Pi\,\widehat{\nu}_k^{(N)})\right]$$

$$\geq \mathbb{E}_{(\mathscr{X},\mathscr{Y})_{1:k}}\left[\max_{\mu,\nu} -\mathcal{L}(\mu, \overline{\Pi}\,\widehat{\mu}_k) - \lambda\,\mathrm{KL}(\mu\|\rho^\mu) + \lambda\,\mathrm{KL}(\overline{\Pi}\,\widehat{\nu}_k\|\rho^\nu)\right.$$

$$\left. + \mathcal{L}(\overline{\Pi}\,\widehat{\mu}_k, \nu) - \lambda\,\mathrm{KL}(\nu\|\rho^\nu) + \lambda\,\mathrm{KL}(\overline{\Pi}\,\widehat{\mu}_k\|\rho^\mu)\right]$$

$$- \frac{M_\mu}{B_k}\sum_{j=1}^{k}\beta_j\mathbb{E}_{(\mathscr{X},\mathscr{Y})_{1:k}}[W_1(\mu_{\mathscr{X}_k}, \Pi\,\widehat{\mu}_k^{(N)})] - \frac{M_\nu}{B_k}\sum_{j=1}^{k}\beta_j\mathbb{E}_{(\mathscr{X},\mathscr{Y})_{1:k}}[W_1(\nu_{\mathscr{Y}_k}, \Pi\,\widehat{\nu}_k^{(N)})]$$

$$\geq \mathbb{E}_{(\mathscr{X},\mathscr{Y})_{1:k}} \left[ \mathrm{NI}(\overline{\Pi}\,\widehat{\mu}_k, \overline{\Pi}\,\widehat{\nu}_k) \right] - \frac{M_\mu}{B_k} \sum_{j=1}^{k} \beta_j \left( \frac{r+1}{j} C_1'(\eta) + C_2'\sqrt{\eta} + C_3' N^{-1/d_{\mathscr{X}} \vee d_{\mathscr{Y}}} \right).$$

Combining with Step 1 and Lemma 3.5 gives that

$$\mathbb{E}\left[ \mathrm{KL}(\overline{\Pi}\,\widehat{\mu}_k \| \mu^*) + \mathrm{KL}(\overline{\Pi}\,\widehat{\nu}_k \| \nu^*) \right] \leq \frac{(r+1)^2}{rk} C_1''(\eta) + C_2'' \sqrt{\eta} + C_3'' N^{-1/d_{\mathscr{X}} \vee d_{\mathscr{Y}}}.$$

Finally, we convert back to a Wasserstein distance bound by invoking Talagrand's inequality and Proposition C.15 again:

$$\mathbb{E}[W_1(\mu_{\overline{\mathscr{X}}_k}, \mu^*)]^2 \leq 2 \left( \frac{1}{B_k} \sum_{j=1}^{k} \beta_j \mathbb{E}[W_1(\mu_{\mathscr{X}_j}, \Pi \widehat{\mu}_j^{(N)})] \right)^2 + \frac{4}{\alpha_\mu} \mathbb{E}[\mathrm{KL}(\overline{\Pi}\,\widehat{\mu}_k \| \mu^*)].$$

This concludes the proof. $\qquad \square$

*Remark.* If we assume a higher degree of regularity so that all relevant distributions have finite fourth moments, say, then Theorem C.14 actually holds for the 2-Wasserstein metric. Theorem C.16 can then be stated in terms of the 2-Wasserstein distance to the MNE, guaranteeing us slightly better control over the error compared to Proposition 3.6 which only allows a $W_1$ formulation.

## D  CONVERGENCE ANALYSIS OF MFL-ABR

### D.1  INNER LOOP CONVERGENCE

The convergence of the decoupled inner loop is a simple consequence of the convex analysis for single optimization (Nitanda et al., 2022a); we reproduce the proof here for completeness.

**Proposition D.1** (Convergence of MFL-ABR inner loop). *Under Assumptions 1 and 3,*

$$\mathrm{KL}(\mu_{k,\tau}^\dagger \| \widehat{\mu}_k) \leq \frac{2C_\mu}{\lambda} \exp(-2\alpha\lambda\tau), \quad \mathrm{KL}(\nu_{k,\tau}^\dagger \| \widehat{\nu}_k) \leq \frac{2C_\nu}{\lambda} \exp(-2\alpha\lambda\tau).$$

*Proof.* For any $0 \leq t \leq \tau$, the KL gap converges as

$$\frac{\mathrm{d}}{\mathrm{d}t} \mathrm{KL}(\mu_{k,t}^\dagger \| \widehat{\mu}_k) = \int_{\mathscr{X}} \log \frac{\mu_{k,t}^\dagger}{\widehat{\mu}_k} \partial_t \mu_{k,t}^\dagger(\mathrm{d}x) = \lambda \int_{\mathscr{X}} \log \frac{\mu_{k,t}^\dagger}{\widehat{\mu}_k} \nabla_x \cdot \left( \mu_{k,t}^\dagger \nabla_x \log \frac{\mu_{k,t}^\dagger}{\widehat{\mu}_k} \right)(\mathrm{d}x)$$

$$= -\lambda \int_{\mathscr{X}} \left\| \nabla_x \log \frac{\mu_{k,t}^\dagger}{\widehat{\mu}_k} \right\|^2 \mu_{k,t}^\dagger(\mathrm{d}x) \leq -2\alpha\lambda \cdot \mathrm{KL}(\mu_{k,t}^\dagger \| \widehat{\mu}_k)$$

by substituting the Fokker-Planck equation for $\mu_{k,t}^\dagger$ and applying the LSI for $\widehat{\mu}_k$ via Theorem A.6. Invoking Gronwall's lemma and Lemma D.2 below for $\widehat{\mu}_k$ concludes the proof. $\qquad \square$

The following result gives uniform bounds to control the magnitude of perturbations.

**Lemma D.2.** *For any $w > 0$, define the class*

$$\mathcal{F}_w^\mu := \left\{ \mu \in \mathcal{P}_2(\mathscr{X}) : \left\| \log \frac{\mu}{\rho^\mu} \right\|_\infty \leq \frac{wC_\mu}{\lambda} \right\}.$$

*Then under Assumption 3, the distribution $\widehat{\mu}_k \in \mathcal{F}_2^\mu$ and $\mu_k, \mu_{k,\tau}^\dagger \in \mathcal{F}_4^\mu$.*

*Proof.* For $\widehat{\mu}_k$, the exponential term and the normalizing integral

$$\exp\left( -\frac{1}{\lambda} \frac{\delta\mathcal{L}}{\delta\mu}(\mu_k, \nu_k) \right), \quad Z_k^\mu = \int_{\mathscr{X}} \rho^\mu \exp\left( -\frac{1}{\lambda} \frac{\delta\mathcal{L}}{\delta\mu}(\mu_k, \nu_k) \right) \mathrm{d}x$$

are both bounded by $T^\mu/\lambda$, proving the assertion. For $\mu_{k,\tau}^\dagger$, define the density ratio $h_t = \mu_{k,t}^\dagger / \widehat{\mu}_k$. The Fokker-Planck equation for $\mu_{k,t}^\dagger$ reads

$$\partial_t \mu_{k,t}^\dagger = \nabla_x \cdot \left( \mu_{k,t}^\dagger \nabla_x \left( \frac{\delta\mathcal{L}}{\delta\mu}(\mu_k, \nu_k) + \lambda \nabla_x U^\mu \right) \right) + \lambda \Delta_x \mu_{k,t}^\dagger = \lambda \nabla_x \cdot \left( \mu_{k,t}^\dagger \nabla_x \log \frac{\mu_{k,t}^\dagger}{\widehat{\mu}_k} \right),$$

so that the parabolic partial differential equation satisfied by $h_t$ is derived as

$$\partial_t h_t = \widehat{\mu}_k^{-1} \, \partial_t \mu_{k,t}^\dagger$$
$$= \lambda \, \widehat{\mu}_k^{-1} \, \nabla_x \cdot (\widehat{\mu}_k \, h_t \nabla_x \log h_t)$$
$$= \lambda \nabla_x \log \widehat{\mu}_k \cdot \nabla_x h_t + \lambda \Delta h_t$$
$$= -\nabla_x \left( \frac{\delta \mathcal{L}}{\delta \mu}(\mu_k, \nu_k) + \lambda \nabla_x U^\mu \right) \cdot \nabla_x h_t + \lambda \Delta h_t$$
$$= \mathbf{L}^\dagger h_t,$$

where $\mathbf{L}^\dagger$ is the infinitesimal generator for the stochastic process $X_t^\dagger$. Hence by the Feynman-Kac formula, we may write for any $t \in [0, \tau]$

$$h_t(x) = \mathbb{E}^x[h_0(X_t^\dagger)] = \mathbb{E}^x \left[ \frac{\rho^\mu}{\widehat{\mu}_k}(X_t) \right].$$

Since $\|\log(\widehat{\mu}_k / \rho^\mu)\|_\infty \le 2C_\mu/\lambda$ as discussed above, we infer that $\|h_\tau\| \le 2C_\mu/\lambda$ and therefore

$$\left\| \log \frac{\mu_{k,\tau}^\dagger}{\rho^\mu} \right\|_\infty \le \left\| \log \frac{\mu_{k,\tau}^\dagger}{\widehat{\mu}_k} \right\|_\infty + \left\| \log \frac{\widehat{\mu}_k}{\rho^\mu} \right\|_\infty \le \frac{4C_\mu}{\lambda},$$

i.e. $\mu_{k,\tau}^\dagger \in \mathcal{F}_\infty^\mu$ for all $k$. Finally, since $\mathcal{F}_\infty^\mu$ is closed under linear combinations in $\mathcal{P}_2(\mathcal{X})$ we conclude that

$$\mu_k = \beta \mu_{k,\tau}^\dagger + \beta(1-\beta)\mu_{k-1,\tau}^\dagger + \cdots \beta(1-\beta)^k \mu_{0,\tau}^\dagger \in \mathcal{F}_\infty^\mu.$$

$\square$

## D.2  PROOF OF THEOREM 4.1

We perform one-step analysis of the outer loop by setting for $0 \le s \le 1$

$$\mu(s) = (1-\beta s)\mu_k + \beta s \mu_{k,\tau}^\dagger, \quad \nu(s) = (1-\beta s)\nu_k + \beta s \nu_{k,\tau}^\dagger,$$

so that $\mu(0) = \mu_k$, $\mu(1) = \mu_{k+1}$ and $\nu(0) = \nu_k$, $\nu(1) = \nu_{k+1}$. We track the KL divergence to the interpolated proximal distributions defined as

$$\widehat{\mu}(s) = \frac{\rho^\mu}{Z^\mu(s)} \exp\left( -\frac{1}{\lambda} \frac{\delta \mathcal{L}}{\delta \mu}(\mu(s), \nu(s)) \right), \quad \widehat{\nu}(s) = \frac{\rho^\nu}{Z^\nu(s)} \exp\left( \frac{1}{\lambda} \frac{\delta \mathcal{L}}{\delta \nu}(\mu(s), \nu(s)) \right).$$

Note that the second order bounds in Assumption 3 immediately imply the following Lipschitz property in TV distance,

$$\left\| \frac{\delta \mathcal{L}}{\delta \mu}(\mu, \nu) - \frac{\delta \mathcal{L}}{\delta \mu}(\mu', \nu') \right\|_\infty \le 2C_{\mu\mu} \, \mathrm{TV}(\mu, \mu') + 2C_{\mu\nu} \, \mathrm{TV}(\nu, \nu').$$

Similarly to Lascu et al. (2023), Lemma A.2 we can then prove that

$$\mathrm{TV}(\widehat{\mu}_k, \widehat{\mu}(s))$$
$$\le \frac{1}{2\lambda} \left( \exp\left( \frac{C_\mu}{\lambda} \right) + \exp\left( \frac{2C_\mu}{\lambda} \right) \right) (2C_{\mu\mu} \, \mathrm{TV}(\mu_k, \mu(s)) + 2C_{\mu\nu} \, \mathrm{TV}(\nu_k, \nu(s)))$$
$$\le \beta s \mathfrak{t}^\mu,$$

where we have written

$$\mathfrak{t}^\mu := \frac{C_{\mu\mu} + C_{\mu\nu}}{\lambda} \left( \exp\left( \frac{C_\mu}{\lambda} \right) + \exp\left( \frac{2C_\mu}{\lambda} \right) \right).$$

Also, $\widehat{\mu}(s) \in \mathcal{F}_2^\mu$ and $\mu(s) \in \mathcal{F}_4^\mu$ by Lemma D.2 which implies $\|\log(\mu(s)/\widehat{\mu}(s))\|_\infty \le 6C_\mu/\lambda$.

Now the derivative of the KL gap of the max policy for any $0 \le s \le 1$ is

$$\frac{\mathrm{d}}{\mathrm{d}s} \, \mathrm{KL}(\mu(s) \| \widehat{\mu}(s)) = \int_{\mathcal{X}} \log \frac{\mu(s)}{\widehat{\mu}(s)} \partial_s \mu(s)(\mathrm{d}x) - \int_{\mathcal{X}} \partial_s \log \widehat{\mu}(s) \mu(s)(\mathrm{d}x).$$

The first term can be decomposed as

$$\int_{\mathcal{X}} \log \frac{\mu(s)}{\widehat{\mu}(s)} \partial_s \mu(s)(\mathrm{d}x)$$

$$= \beta \int_{\mathcal{X}} \log \frac{\mu(s)}{\widehat{\mu}(s)} (\mu_{k,\tau}^{\dagger} - \mu_k)(\mathrm{d}x)$$

$$= \beta \int_{\mathcal{X}} \log \frac{\mu(s)}{\widehat{\mu}(s)} (\widehat{\mu}(s) - \mu(s) + \mu(s) - \mu_k + \mu_{k,\tau}^{\dagger} - \widehat{\mu}_k + \widehat{\mu}_k - \widehat{\mu}(s))(\mathrm{d}x)$$

$$\leq -\beta \left( \mathrm{KL}(\mu(s)\| \widehat{\mu}(s)) + \mathrm{KL}(\mu(s)\| \widehat{\mu}(s)) \right) + 2\beta^2 s \left\| \log \frac{\mu(s)}{\widehat{\mu}(s)} \right\|_{\infty} \mathrm{TV}(\mu_{k,\tau}^{\dagger}, \mu_k)$$

$$\quad + \beta \left\| \log \frac{\mu(s)}{\widehat{\mu}(s)} \right\|_{\infty} \sqrt{2\,\mathrm{KL}(\mu_{k,\tau}^{\dagger}\| \widehat{\mu}_k)} + 2\beta \left\| \log \frac{\mu(s)}{\widehat{\mu}(s)} \right\|_{\infty} \mathrm{TV}(\widehat{\mu}_k, \widehat{\mu}(s))$$

$$\leq -\beta\,\mathrm{KL}(\mu(s)\| \widehat{\mu}(s)) + \frac{12\beta C_\mu}{\lambda} \left( 2\beta s(\mathsf{t}^\mu + 1) + \sqrt{\frac{C_\mu}{\lambda}} \exp(-\alpha\lambda\tau) \right).$$

by Proposition D.1. For the second term, we may follow the derivations presented in Section 3 of Lascu et al. (2023) with minimal modifications to obtain

$$- \int_{\mathcal{X}} \partial_s \log \widehat{\mu}(s) \mu(s)(\mathrm{d}x)$$

$$= -\frac{\beta}{\lambda} \iint_{\mathcal{X}\times\mathcal{X}} \frac{\delta^2 \mathcal{L}}{\delta\mu^2}(\mu(s), \nu(s), x, z)(\widehat{\mu}(s) - \mu(s))(\mathrm{d}x)(\widehat{\mu}_k - \mu_k)(\mathrm{d}z)$$

$$\quad + \frac{\beta}{\lambda} \iint_{\mathcal{X}\times\mathcal{Y}} \frac{\delta^2 \mathcal{L}}{\delta\mu\delta\nu}(\mu(s), \nu(s), x, w)(\widehat{\mu}(s) - \mu(s))(\mathrm{d}x)(\widehat{\nu}_k - \nu_k)(\mathrm{d}w).$$

When $s = 0$, the first integral is nonpositive due to convexity while the second integral cancels out when adding with the corresponding term for the KL gap of the max policy, which completes the argument in Lascu et al. (2023). Hence the remaining error we must control is

$$-\frac{\beta}{\lambda} \iint_{\mathcal{X}\times\mathcal{X}} \frac{\delta^2 \mathcal{L}}{\delta\mu^2}(\mu(s), \nu(s), x, z)(\widehat{\mu}(s) - \widehat{\mu}_k + \mu_k - \mu(s))(\mathrm{d}x)(\widehat{\mu}_k - \mu_k)(\mathrm{d}z)$$

$$\quad + \frac{\beta}{\lambda} \iint_{\mathcal{X}\times\mathcal{Y}} \frac{\delta^2 \mathcal{L}}{\delta\mu\delta\nu}(\mu(s), \nu(s), x, w)(\widehat{\mu}(s) - \widehat{\mu}_k + \mu_k - \mu(s))(\mathrm{d}x)(\widehat{\nu}_k - \nu_k)(\mathrm{d}w)$$

$$\leq \frac{4\beta}{\lambda}(C_{\mu\mu} + C_{\mu\nu})(\mathrm{TV}(\widehat{\mu}(s), \widehat{\mu}_k) + \mathrm{TV}(\mu_k, \mu(s)))$$

$$\leq \frac{4\beta^2 s}{\lambda}(C_{\mu\mu} + C_{\mu\nu})(\mathsf{t}^\mu + 1).$$

Adding everything up, we obtain

$$\frac{\mathrm{d}}{\mathrm{d}s} \left( \mathrm{KL}(\mu(s)\| \widehat{\mu}(s)) + \mathrm{KL}(\nu(s)\| \widehat{\nu}(s)) \right)$$

$$\leq -\beta \left( \mathrm{KL}(\mu(s)\| \widehat{\mu}(s)) + \mathrm{KL}(\nu(s)\| \widehat{\nu}(s)) \right)$$

$$\quad + \frac{12\beta C_\mu}{\lambda} \left( 2\beta(\mathsf{t}^\mu + 1) + \sqrt{\frac{C_\mu}{\lambda}} \exp(-\alpha\lambda\tau) \right) + \frac{4\beta^2}{\lambda}(C_{\mu\mu} + C_{\mu\nu})(\mathsf{t}^\mu + 1)$$

$$\quad + \frac{12\beta C_\nu}{\lambda} \left( 2\beta(\mathsf{t}^\nu + 1) + \sqrt{\frac{C_\nu}{\lambda}} \exp(-\alpha\lambda\tau) \right) + \frac{4\beta^2}{\lambda}(C_{\mu\nu} + C_{\nu\nu})(\mathsf{t}^\nu + 1).$$

By applying Gronwall's lemma over $s \in [0, 1]$ and iterating over $k$, we conclude that

$$\mathrm{KL}(\mu_k\| \widehat{\mu}_k) + \mathrm{KL}(\nu_k\| \widehat{\nu}_k)$$

$$\leq \frac{2(C_\mu + C_\nu)}{\lambda} \exp(-\beta k) + \frac{12}{\lambda^{\frac{3}{2}}} \left( C_\mu^{\frac{3}{2}} + C_\nu^{\frac{3}{2}} \right) \exp(-\alpha\lambda\tau)$$

$$+ \frac{4\beta}{\lambda} \left((6C_\mu + C_{\mu\mu} + C_{\mu\nu})(\mathfrak{t}^\mu + 1) + (6C_\nu + C_{\mu\nu} + C_{\nu\nu})(\mathfrak{t}^\nu + 1)\right).$$

Finally, applying Lemma 3.4 of Lascu et al. (2023) yields the suboptimality bound

$$\mathrm{NI}(\mu_k, \nu_k) \le 2(C_\mu + C_\nu)\exp(-\beta k) + \frac{12}{\sqrt{\lambda}}\left(C_\mu^{\frac{3}{2}} + C_\nu^{\frac{3}{2}}\right)\exp(-\alpha\lambda\tau) + C\beta.$$

Hence an $\epsilon$-MNE may be obtained in $k = O(\frac{1}{\epsilon}\log\frac{1}{\epsilon})$ outer loop iterations by taking $\beta = O(\epsilon)$ and $\tau = O(\log\frac{1}{\epsilon})$. $\qquad\square$

