# OpenReview forum: "Symmetric Mean-field Langevin Dynamics for Distributional Minimax Problems"
_ICLR.cc/2024/Conference — ICLR 2024 spotlight_

### Official Review · Reviewer_tSuv · 2023-10-18

**Soundness:** 4 excellent
**Presentation:** 3 good
**Contribution:** 4 excellent
**Rating:** 8
**Confidence:** 2

**Summary:**

The authors considered the problem of developing a symmetric mean-field Langevin dynamics (MFLD) algorithm for distributional minimax problems with global convergence guarantees. Based on the mean-field Langevin descent ascent (MFL-DA) dynamics, the authors proposed mean-field Langevin averaged gradient (MFL-AG) flow that replaces the MFL-DA drift with the historical weighted average; based on the mean-field best response (MF-BR) flow, the authors proposed the mean-field Langevin anchored best response (MFL-ABR) process. Extensive convergence analyses were provided for both methods. The authors also studied the time and particle discretization error and applied their methods to zero-sum Markov games. Numerical experiments were conducted to compare their methods with MFL-DA.

**Strengths:**

Compactly written, this paper is the first to establish convergence results in using MFLD for minimax optimization over probability measures. By incorporating a weighting scheme into MFL-AG, the authors offered a creative direction for future works on MFLD. The comprehensive discretization analysis complements the continuous-time convergence result well.  Their second method, MFL-ABR, is also an interesting improvement upon MF-BR that is realizable by a particle algorithm and enjoys convergence guarantees.

**Weaknesses:**

The paper could be improved by providing more intuitions for the proposed methods. For example, what is the intuition for using a weighting scheme in MFL-AG, and what might cause MFL-AG and MFL-ABR to converge more slowly than MFL-DA in the numerical experiments? Is it correct to think that the slowdown of the convergence ensures better convergence?

**Questions:**

See weakness. Also, what are some limitations of the proposed methods?

---

> ### Author Response · Authors · 2023-11-17
>
> Thank you for your positive review and helpful suggestions! We have updated the paper and clarified some technical details as well as improving presentation.
>
> **Addressing Weaknesses**
>
> **Intuition.** The intuition for the MFL-AG averaging scheme is twofold. First, it is based on the general scheme of dual averaging notably explored in [1,2], where specific weightings for regularized problems were also considered. A more recent work [3] proposed the Particle Dual Averaging algorithm for particle-based convex optimization with weighting $\propto t$. This agrees with our results (Theorem 3.4, 3.7) which show that weighting $t^r$ with exponent $r=1$ is optimal. Second, it is related to the game theory concept of fictitious play, where each player assumes their opponent is playing a stationary strategy and takes the best response to their average behavior, that is the unweighted averaged-gradient proximal distribution. Specifically, MFL-AG is the mean-field Langevin flow that realizes ideal stepwise fictitious play. Adding a weighting then has the meaning of putting higher trust in more recent information, which is appropriate if the opponent strategy or environment is not quite stationary.
>
> **Convergence.** MFL-AG always exhibits a "slow but steady" convergence rate of $\Theta(1/t)$ induced by drift averaging. In contrast, MFL-ABR is slower due to the double loop structure by a factor of the inner loop max iteration. These features each contribute to the improved optimality of the two algorithms. The performance of MFL-ABR also depended on the choice of hyperparameters. For example, speeding up the inner loop of by decreasing iterations and increasing $\eta$, or speeding up the outer loop by increasing $\beta$ can accelerate convergence of MFL-ABR, but also leads to instability in some runs.
>
> In addition, MFL-ABR and MFL-DA performed better for problems where $Q(x,y)$ factored into a sum (or product) of separate potentials for $x,y$. In this case the dynamics decouples into two single-optimization problems which are known to converge exponentially. In contrast, MFL-AG was preferable for potentials with complex interactions between $x,y$. For example for the sigmoid potential in Section 6, $\mu$ pulls $\nu$ toward itself while $\nu$ pushes away, leading to a more complicated interacting dynamics that is better solved by the more stable MFL-AG.
>
> **Addressing Questions**
>
> Here are some limitations of the proposed algorithms.
>
> * If the objective is not bilinear, the discussion in Section 5.1 does not apply and MFL-AG requires the entire history to be stored in memory which is costly considering the longer runtime. Nonetheless, we can avoid iterating over the full history to compute the average by random sampling from past states with weight $\propto\beta_k$ so that computational complexity does not increase.
> * Returning the average iterate after $K$ steps is also not ideal from an algorithmic perspective. In practice we can simply compare optimality to the last output and choose the better solution, record the best performing last/average iterate, implement early stopping based on relative NI error, etc.
> * The speed and stability of MFL-ABR is highly dependent on the inner loop parameters. The learning rate and max iterations must both be tuned for best performance.
> * Some reviewers mentioned the adverse dependency of the convergence rate on $\lambda$ which is common to MFLD methods. This can be overcome by incorporating a $\lambda_t\propto 1/\log t$ cooling to converge to the saddle point of the unregularized objective $\mathcal{L}$; please see our response to Q3 of Review q6kk for details. For certain problems, exponential cooling can also be achieved [3,4].
>
> [1] Nesterov. Primal-dual subgradient methods for convex problems. Mathematical Programming, 120, 2007.
>
> [2]. Xiao. Dual averaging method for regularized stochastic
> learning and online optimization. NIPS 2009.
>
> [3] Suzuki et al. Feature learning via mean-field Langevin dynamics:
> classifying sparse parities and beyond. NeurIPS 2023.
>
> [4] Chizat. Mean-field Langevin dynamics: exponential convergence and annealing. TMLR 2022.

---

> > ### Comment · Reviewer_tSuv · 2023-11-22
> >
> > Thank you for your detailed response. I will keep my current score.

---

### Official Review · Reviewer_q6kk · 2023-10-25

**Soundness:** 3 good
**Presentation:** 2 fair
**Contribution:** 2 fair
**Rating:** 6
**Confidence:** 4

**Summary:**

This paper studied the mean-field minimax theorem with entropic regularization. The contributions can be summarized as follows:

1. The authors proposed the MFL-AG algorithm, which convergence polynomially in continuous time. The author also proposed a discretized algorithm with a uniform-in-time propagation of chaos.

2. The authors proposed the MFL-ABR algorithm and provided a convergence rate for the discrete outer loop.

**Strengths:**

1. The proposed MFL-AG dynamic is new in the literature, and the weighted average of the dynamic provably converges to the MNE polynomially in time.

2. The authors provide a uniform-in-time propagation of chaos for a finite-particle implementation of the MFL-AG flow. Despite I have a few questions on the utility of the results, I believe the techniques based on Chen et.al 2022 are novel in mean-field minimax problem literature, and could potentially benefit future works.

3. The proposed MFL-ABR flow provides a potential way to find a discrete finite-particle algorithm for the best-response dynamics in mean-field min-max problem settings.

**Weaknesses:**

**Regarding the MFL-AG**:

1. The utility of the finite particle analysis in section 3.3 is not very clear to me. I have the following questions:

- 1.1 In Theorem 3.7, you bound the Wasserstein distance  $ W_1( {E}[ \mu_{ \overline{\mathscr{X}}\_k } ], \mu\_* )$. However, I didn't see why it is useful to consider ${E}[ \mu_{ \overline{\mathscr{X}}\_k } ]$ since ${E}[ \mu_{ \overline{\mathscr{X}}\_k } ]$ is basically the average of the pushforward of joint measures to each particle, what does the convergence of this measure to MNE implies? To my understanding, the propagation of chaos refers to that the particles behave almost i.i.d, how does this convergence results imply the particles are almost i.i.d ? Thus, it makes more sense to me to consider  $ W_1 ( \mu_{ \overline{\mathscr{X}}\_k }, \mu_*)  $ for example.

- 1.2 Cont'd: you mentioned in the footnote on page 6 that the typical rate Wasserstein distance between a measure and **i.i.d** sampled empirical measure is $N^{-1/d}$. In this case, it seems that the particles in $ \mu_{ \overline{\mathscr{X}}\_k}$ are highly correlated since the dynamics are coupled. The correlation is not only between the particles in $ \overline{\mathscr{X}}\_k$ for a fixed $k$, but also between particles in different times (i.e.$ \overline{\mathscr{X}}\_k$ and $ \overline{\mathscr{X}}\_j$ for $j \neq k$ ). Thus, it is unclear to me what is the convergence rate of $W_1(\mu_{ \overline{\mathscr{X}}\_k},   \mathbb{E} [ \mu_{ \overline{\mathscr{X}}\_k} ] )$?

- 1.3 The discussion on top of page 7 is not very clear to me. First of all, you say you expand around $\mu, \hat{\nu} $, but the first order condition in (3) is only for MNE, i.e. it is not clear to me why $ \frac{\delta }{\delta \mu} \mathcal{L}\_\lambda(\mu, \hat{\nu}) =  \frac{\delta }{\delta \hat{\nu}} \mathcal{L}\_\lambda(\mu, \hat{\nu}) = 0$ for an arbitrary $\mu$. Do you mean you expand around $  \mu_*, \nu_*$? Also, it is unclear to me why one can ignore the higher-order terms. Thus, currently, I don't understand how the upper bound on $W_1$ implies an upper bound on $NI$ error. In fact, since the $NI$ error upper bound the KL-divergence, and the KL-divergence upper bound $W_1$, I doubt the correctness of that the $NI$ error can be upper bounded by the sum of $W_1$ in general.

- 1.4 Cont'd: in the case where $\mathcal{L}$ is bilinear and assumes good regularity (e.g. the kernel is uniformly bounded), is that possible to upper bound  $NI( \mathbb{E}[ \mu\_{ \overline{\mathscr{X}}\_k } ], \mathbb{E}[\nu_{ \overline{\mathscr{Y}}_k } ])$ by the sum of $W_1$ in the propagation of chaos results?

- 1.5 Cont'd: I suggest making the discussions on top of page 7 more formal and rigorous. The current ambiguous discussion makes it hard to understand the utility of the propagation of chaos results.


**Regarding the MFL-ABR**

2. The utility of the results in section 4 is also not very clear to me, I have the following questions:

- 2.1 It's not very clear to me why algorithm 2 works. Since in each round, one should input $\mu_k, \nu_k$ which comes from the outer iteration, while in algorithm 2, input $\mu\_{\mathscr{X}\_k}, \nu\_{\mathscr{Y}\_k} $ which are empirical measures. Due to the lack of propagation of chaos results, it is not clear to me that $\mu\_{\mathscr{X}\_k}, \nu\_{\mathscr{Y}\_k} $ will converge to $\mu\_k, \nu\_k $.

- 2.2 Theorem 4.1 analyzes the convergence of the discretized version of MFL-ABR. However, compared to the results in Lascu et al. (2023), it seems to me that there are two drawbacks due to the inner Langevin dynamics: (1) There's a non-vanishing error $C \beta$, so one has to make $\beta$ small which in turn will make the convergence rate small. (2) The convergence rate of inner Langevin dynamics depends on the LSI, in the case of small $\lambda$, the convergence rate can be small.  These two drawbacks might limit the utility of the MFL-ABR algorithm.

**Questions:**

Please also refer to the strengths and weaknesses part.

1.  In Chen et al. (2022), they provide a propagation of chaos in KL-divergence, under extra regularity assumptions (see Theorem 2.4 in Chen et al. (2022) ). In this paper's settings, do you expect a similar propagation of chaos in KL divergence? Also, the propagation of chaos in Chen et al. 2022 provides a $O(1/N)$ error, which is optimal in order of $N$. Is it possible to improve the bounds to  $O(1/N)$ or is there any lower bound on the convergence rate (of propagation of chaos)?

2. In the numerical experiments, you use a 3-point NI error to approximate NI error, what's the justification for this? Since in principle for given $\mu_{\mathscr{X}^i}$, one should compute $\arg\max_{\nu} L_{\lambda}(\mu_{\mathscr{X}^i}, \nu )  $ , it could be that $ \nu_{\mathscr{Y}^j}$ are far from $\arg\max_{\nu} L_{\lambda}(\mu_{\mathscr{X}^i}, \nu ) $. Besides, for fixed $\mu_{\mathscr{X}^i} $, to compute $\arg\max_{\nu} L_{\lambda}(\mu_{\mathscr{X}^i}, \nu )  $, can't one directly run a mean-field Langevin dynamics for example?

3 . Is it possible to apply the annealing techniques in Lu (2022) to achieve the MNE of the unregularized object (assume there is)?

---

> ### Author Response · Authors · 2023-11-17
>
> We are extremely grateful for your detailed review of the various analyses and in-depth suggestions! We have updated and clarified various important points in the manuscript. We hope our discussion can convince the reviewer of the technical novelty and complexity of our overall theory as well as the utility of the results.
>
> **Addressing Weaknesses**
>
> **1.1-2.** $\mathbb{E}[\mu\_{\overline{\mathscr{X}}\_k}]$ is the expected output of MFL-AG, hence Theorem 3.7 quantifies a sort of bias. While this does not provide the full picture, we put the expectation inside $W_1$ and carefully designed our approach (using uniform LLN, short-term perturbation, etc) to bypass the curse of dimensionality. We mentioned the i.i.d. sample bound simply to illustrate that the CoD is unavoidable for $W_1(\mu\_{\mathscr{X}\_k}, \mathbb{E}[\mu\_{\mathscr{X}\_k}])$ or $W_1(\mu\_{\mathscr{X}\_k}, \mu^*)$ even in the ideal situation (no chaos) where the particles are perfectly independently sampled from the true distribution.
>
> Nonetheless, we see the utility in obtaining a bound for $W_1(\mu\_{\mathscr{X}\_k}, \mu^*)$. Thus, we have added a completely new section which fleshes out the above discussion and develops similar bounds for $W_1(\mu\_{\mathscr{X}\_k}, \mu^*)$ to Theorem 3.7 save for the CoD, which is unavoidable in the weakly interacting setting of propagation of chaos. Please see Appendix C.6.
>
> Of course, in practice we may also simply run the algorithm multiple ($M$) times and take the average of the outputs, which would also bypass the issue and yield the standard $1/\sqrt{M}$ convergence.
>
> We also remark that while the statement of Theorem 3.7 does not directly imply i.i.d. behavior of particles, the philosophy of propagation of chaos plays a key role in the proof. For example when developing the uniform law of large numbers we consider the system as a perturbed version of the gradient-stopped process, whose particles are independent when conditioned on the history up to a certain point in time. Hence, we emphasize the utility of our results lies not only in Theorem 3.7 but also the novel perturbation analysis developed throughout Appendix C.
>
> **1.3-5.** Thank you for your astute observations. The upper bound does not hold in general; the sketched idea only works for the *unregularized* NI error (without KL terms), which we have not considered in the paper. At present, there seems to be no clean way to bound entropic terms (KL, NI) regarding $\mathbb{E[\mu\_{\mathscr{X}\_k}]}, \mathbb{E[\nu\_{\mathscr{Y}\_k}]}$. We apologize for the mistake, and the claims have been deleted as discussing unregularized error will be too confusing. (Propagation of chaos is often presented in terms of Wasserstein distance throughout the literature. We merely wanted to strengthen the remark that the squared distance $W_1^2$ (and not $W_1$) is a natural unit of measurement on the left-hand side of Theorem 3.7 in view of Talagrand's and sandwich inequalities.)
>
> **2.1.** In general, one does not know $\mu\_{\mathscr{X}\_k}$ will converge to $\mu_k$ a priori since error can accumulate as training progresses, and instead directly proves convergence to $\mu^*$ (which then also implies $\mu\_{\mathscr{X}\_k}$ and $\mu_k$ are close). For each outer loop anchor $\mu\_{\mathscr{X}\_k}, \nu\_{\mathscr{Y}\_k}$ the propagation of chaos for the inner loop is guaranteed by [1] since the dynamics decouples into single optimization; the laws of each of the inner loop particles converge to the Gibbs proximal distributions of the anchors. Then intuitively we expect the Lyapunov function proof in Section D.2 to apply to $\mu\_{\mathscr{X}\_k}, \nu\_{\mathscr{Y}\_k}$ plus a $O(1/N)$ error (without the need to appeal to $\mu_k,\nu_k$). In particular, the outer loop random sampling update satisfies on average $\mathbb{E}[\mu\_{\mathscr{X}\_{k+1}}] = \beta\cdot\mathbb{E}[\mu\_{\mathscr{X}\_k}]+(1-\beta)\mathbb{E}[\widehat{\mu}\_{\mathscr{X}\_k}]+O(e^{-\Omega(\tau)}+\eta)$, which is identical to the mean-field best response update plus the inner loop error.
>
> The reason we did not complete the proof is that in order to make the above argument rigorous, the $N$-particle joint distribution must also behave roughly like an independent sample from the average $\mathbb{E}[\mu\_{\mathscr{X}\_k}]$ (like the uniform LLN for MFL-AG). However, we found quantifying this difficult due to the combinatorically complicated effect of random sampling & replacing on the joint distribution. Since this issue is unrelated to Langevin dynamics, and also since there are other methods to perform the update (such as that suggested by Reviewer DQbp), we decided to not spend too much time on the issue.
>
> (continued)

---

> ### Author Response · Authors · 2023-11-17
>
> (continued)
>
> **2.2.** (1) We view the $C\beta$ time discretization error as a natural consequence of turning a continuous flow into a realizable algorithm rather than a drawback; the same can be said for any gradient descent algorithm with fixed learning rate. However it should be possible to implement a learning rate annealing for $\beta$ as $1/\log t$ to avoid this error.
>
> (2) Indeed, the potential adverse dependency on $\lambda$ is a problem that plagues all MFLD-based algorithms. In our response to Q3, we discuss how $\lambda_k=O(1/\log k)$ temperature annealing can overcome this issue. In addition, for certain classification problems where $\lVert\delta\mathcal{L}/\delta\mu\rVert_\infty$ can be bounded by $\mathcal{L}$, a faster exponential annealing of $\lambda_k$ can be implemented so that the convergence rate is independent of $\lambda_k$. [2]
>
> **Addressing Questions**
>
> **Q1.** Proving propagation of chaos in KL divergence would be ideal; this is the approach taken in [1] as well, and is much more direct and simpler. We tried to emulate this framework at first, but quickly found out key concepts such as their $N$-particle proximal distribution cannot be applied. This is due to the dependence on the entire history and cannot be fixed by adding stronger assumptions. Thus we had to resort to more technically involved methods (uniform LLN, short-time perturbation analysis) leading to our $W_1$ result. While morally there seems to be no reason why the result would not hold in KL, our current tools are not enough to guarantee this.
>
> For the uniform LLN (Proposition 3.6) at least, the $O(1/\sqrt{N})$ rate is optimal. The worst-case LLN error is determined by the change in Lipschitz constant of the function $F(\mu_{\mathscr{X}}, \nu_{\mathscr{X}})$ when each particle is removed, which in turn is tightly bounded by the leave-one-out $W_1$ error due to W1-Lipschitzity of $F(\cdot,\cdot)$. Since the leave-one-out $W_2^2$ error is $O(1/N)$ (which is central in Chen's argument), the LLN error must be $O(1/\sqrt{N})$. Since we did not quite use the full power of the uniform LLN in subsequent analyses, however, it is possible that a different approach could circumvent this and achieve $O(1/N)$, which seems an interesting direction for future work. Nevertheless, we emphasize this is the first history-dependent propagation of chaos result with explicit rates in any setting. Existing related results in the SPDE literature are asymptotic and do not provide error bounds [3].
>
> **Q2.** $\text{argmax}\_j \mathcal{L}\_\lambda(\mu\_{\mathscr{X}^i},\nu\_{\mathscr{Y}^j})$ is not meant to approximate $\text{argmax}\_\nu \mathcal{L}\_\lambda(\mu\_{\mathscr{X}^i},\nu)$ per se but to measure relative optimality between the 3 solutions (this has now been clarified in the paper). Similarly to a payoff matrix, a nonzero relative NI error implies switching to another $\nu\_{\mathscr{Y}^j}$ will improve $\mathcal{L}\_\lambda(\mu_{\mathscr{X}^i}, \cdot)$, and same for $\mu\_{\mathscr{X}^i}$. Of course for a finite payoff matrix a pure Nash equilibrium does not always exist in which case it is hard to say which is better, but in all of our experiments 1 of the 3 algorithms exhibited zero or near-zero error.
>
> As you mentioned, $\text{argmax}\_\nu \mathcal{L}\_\lambda(\mu\_{\mathscr{X}^i},\nu)$ can be directly computed by running single-objective MFLD in principle. The problem is that this strongly biases the results towards MFL-ABR in practice, since decoupled MFLD is precisely what the inner loop is running. That is, MFL-ABR converges when MFLD w.r.t. $\mu\_k$ ($\nu\_k$) outputs $\nu\_k$ ($\mu\_k$), which is exactly when MFLD-based NI error will evaluate to zero, regardless of various empirical factors that make these processes diverge from theory.
>
> A third option is to compute the distance to the MNE $(\mu^*,\nu^*)$ if it can be determined. However, it is very difficult to find closed-form solutions for objectives with nontrivial interactions between $\mu,\nu$, and we cannot rely on the final output of the 3 proposed algorithms since we saw that they converge to different distributions in practice. These issues are what led us to use the indirect 3-point NI error.
>
> **Q3.** We have performed some rough computations; it is indeed possible to achieve the unregularized MNE using MFL-AG with an annealing rate $\lambda_t=O(1/\log t)$ as in Lu's work or [4]. This results in a proximal gap (Prop. 3.3) of $O(1/(\log t)^2)$ and NI error convergence rate (Theorem 3.4) of $O(1/\log t)$. Interestingly, this also absorbs the slow $O(1/t)$ rate of the regularized problem. A caveat is that we must assume $\delta\mathcal{L}/\delta\mu$ is bounded rather than Lipschitz as in Assumption 3 (a usually stronger condition) in order to avoid the use of Talagrand's inequality and instead use TV distance bounds as in Appendix D. For MFL-ABR, we can similarly anneal $\lambda_k$ (keeping it fixed in the inner loop) to achieve $O(1/\log k)$ in the outer loop.

---

> ### Author Response · Authors · 2023-11-17
>
> (continued)
>
> [1] Suzuki et al. Mean-field Langevin dynamics: Time-space discretization, stochastic gradient, and variance reduction. NeurIPS 2023.
>
> [2] Suzuki et al. Feature learning via mean-field Langevin dynamics:
> classifying sparse parities and beyond. NeurIPS 2023.
>
> [3] Wu et al. On a class of McKean-Vlasov stochastic functional differential equations with applications. 2023.
>
> [4] Chizat. Mean-field Langevin dynamics: exponential convergence and annealing. TMLR 2022.

---

> > ### Comment · Reviewer_q6kk · 2023-11-21
> >
> > I thank the authors for their detailed explanation of my questions and concerns. I believe most of my concerns are well-addressed.
> >
> > For the propagation of chaos results for the MFL-AG algorithm, the interpretation of  Theorem 3.7 as the bias of the " propagation of chaos " is valid to me, and I understand the dependence of dimensionality for variance term might be inevitable technically. However, I'm still a bit concerned about whether the propagation of chaos in $W_1$ is strong enough to interpret the utility of the mean-field minimax problem.
> >
> > For example,  consider the problem of training a two-layer network as discussed in [1, section 2.1 example 1], the propagation of chaos results will imply $ \mathbb{E}_{\mathscr{ X}\_k } [ | f(z;\mu\_{\mathscr{ X}\_k } ) -   f(z;\mu\_*) |^2 ] \leq \mathcal{O}(  \frac{1}{N} + e^{- \eta k} ), \quad \forall z$  as discussed in [1, section "Conversion to a Wasserstein distance bound"].  This gave a good interpretation that the output of the finite-width network is close to the optimal infinite-width neural network in expectation as $N$ is large and training sufficiently long. Thus, in this case, the propagation of chaos results in $W_2$ is good enough.
> >
> > In mean-field minimax problems, especially in mean-field game settings, it seems to me that people are more interested in interpreting the closeness to MNE by measuring $ NI( \mu\_{\mathscr{ X}\_k }, \nu\_{\mathscr{ X}\_k } ) $ for example. Since from Theorem 3.7, it is not obvious to see whether it will imply an upper bound on  $ NI( \mu\_{\mathscr{ X}\_k }, \nu\_{\mathscr{ X}\_k } ) $,  I'm still not convinced whether the propagation of chaos in $W_1^2$ is strong enough.
> >
> >
> > Nevertheless, now I believe that Theorem 3.7 is indeed a valid propagation of chaos results which is new in the literature and the techniques are novel from the explanations of the authors.

---

> > > ### Author Response · Authors · 2023-11-22
> > >
> > > Thank you for taking the time to read through our responses! We agree that the current statement is not ideal to present our results, however alternatives are somewhat unclear. $\text{NI}(\mu\_{\mathscr{X}\_k}, \nu\_{\mathscr{Y}\_k})$ is lower bounded by KL which is lower bounded by $W_2^2$, so it is impossible to avoid the curse of dimensionality for NI error. A compromise could be $\text{NI}(\mathbb{E}[\mu\_{\mathscr{X}\_k}], \mathbb{E}[\nu\_{\mathscr{Y}\_k}])$ or $\text{KL}(\mathbb{E}[\mu\_{\mathscr{X}\_k}]\Vert\mu^*)$, however entropic terms involving $\mathbb{E}[\mu\_{\mathscr{X}\_k}]$ cannot be controlled with only the uniform LLN.
> > >
> > > In fact, we can show that **unregularized** NI error $\text{NI}\_0(\mathbb{E}[\mu\_{\mathscr{X}\_k}], \mathbb{E}[\nu\_{\mathscr{Y}\_k}]) = \max\_{\nu'}\mathcal{L}(\mathbb{E}[\mu\_{\mathscr{X}\_k}],\nu')-\min\_{\mu'}\mathcal{L}(\mu',\mathbb{E}[\nu\_{\mathscr{Y}\_k}])$, if it is defined, can be upper bounded by $W_1$ as discussed above. This is the corresponding result to [1, Conversion to a Wasserstein distance bound], while this cannot be extended to regularized NI due to additional entropic terms.
> > >
> > > Define $\text{NI}\_0(\mu,\nu)=\max\_{\nu'}\mathcal{L}(\mu,\nu')-\min\_{\mu'}\mathcal{L}(\mu',\nu)$ and suppose $\mathcal{L}$ is strongly convex-concave so that it possesses a unique saddle point $(\mu\_0^*,\nu\_0^*)$. More precisely, suppose $\mathcal{L}(\mu^*,\nu)\geq \mathcal{L}(\mu\_0^*,\nu) + \kappa W_1^2(\mu^*,\mu\_0^*)$, then we have the chain of inequalities
> > >
> > > $$\mathcal{L}(\mu_0^*,\nu^*) + \kappa W_1^2(\mu^*,\mu_0^*) \leq \mathcal{L}(\mu^*,\nu^*)\leq \mathcal{L}(\mu^*,\nu^*)+\lambda\text{KL}(\mu^*\Vert\rho^\mu) \leq \mathcal{L}(\mu_0^*,\nu^*)+\lambda\text{KL}(\mu_0^*\Vert\rho^\mu)$$
> > >
> > > which shows that $W_1(\mu^*,\mu_0^*)=O(\sqrt{\lambda})$. Furthermore, we have $\max_{\nu'}\mathcal{L}(\mu,\nu')-\mathcal{L}(\mu_0^*,\nu_0^*)\leq \max_{\nu'} (\mathcal{L}(\mu,\nu')-\mathcal{L}(\mu_0^*,\nu'))\leq M_\mu W_1(\mu,\mu_0^*)$. Combining and adding with the corresponding result for $\nu$ gives that $\text{NI}_0(\mu,\nu)\leq O(W_1(\mu,\mu^*)+W_1(\nu,\nu^*)+\sqrt{\lambda})$.
> > >
> > > We hope this can clarify the reviewer's concerns.

---

### Official Review · Reviewer_ByeN · 2023-10-28

**Soundness:** 2 fair
**Presentation:** 2 fair
**Contribution:** 3 good
**Rating:** 6
**Confidence:** 3

**Summary:**

This paper presents an approach to minimize optimization over probability distributions by extending the principles of mean-field Langevin dynamics. They propose two methods: (MFL-AG),  a single-loop algorithm designed to achieve convergence towards the mixed Nash equilibrium, and (MFL-ABR), a two-loop algorithm that demonstrates linear convergence in the outer loop. Furthermore, they also investigate various discretization methods and offer a fresh analysis of how chaos spreads, taking into account its dependencies on past distributions. This research paves the way for more in-depth investigations into mean-field dynamics for numerous learning agents.

**Strengths:**

* The paper's contributions are supported by theoretical proofs
* It seems that the convergence of mean-field Langevin dynamics was still an open problem and the problem of this paper is well defined.

**Weaknesses:**

* Proposed methods could be well motivated. The two proposed methods themselves are convincing. However, is there any reasons or motivation why you have both methods at the same time: mean-field Langevin averaged gradient and mean-field Langevin anchored the best response? Any explicit relationship between these two settings?
* The description of the equation can be more comprehensive. Usually, we should describe all the notations of the equation at least right after it. For example, you do not mention what $\rho^{\mu}$ and $\rho^{\nu}$ are in equation 1.

**Questions:**

* Could you please explain more about what "SYMMETRIC" means in your proposed method and title? Is it mainly from the same temperature or regularization strength $\lambda$ for both players?
* Then how compatible your proposed method and proof can be extended to when there exists $\lambda_1$ and $\lambda_2$ for two players respectively? Since you mentioned that the problems you are looking for naturally arise for example solving robust learning or zero-sum games in reinforcement learning. Usually, in the robustness via adversarial learning, the regularization strength is relevant to the strength of the adversary. For example, if the adversary is attacked on soft actor-critic (SAC)[1], which originally had its regularization term (temperature), then it is not flexible to have the regularization terms for the target agent and the adversary the same values.
* Could you analyze why MFL-AG can have a lower NI error compared with MFL-ABR and does it mean that MFL-AG performs better than MFL-ABR in this experiment? Furthermore, please suggest what kind of numerical experiment is more suitable with MFL-ABR against MFL-AG, and vice versa.

[1] Haarnoja, T., Zhou, A., Abbeel, P., and Levine, S. Soft actor-critic: Off-policy maximum entropy deep reinforcement learning with a stochastic actor. In International Conference on Machine Learning (ICML), 2018

---

> ### Author Response · Authors · 2023-11-17
>
> Thank you for your helpful advice and comments! We are very grateful for your through review, and various parts of the manuscript have been clarified or updated to improve presentation.
>
> **Addressing Weaknesses**
>
> **W1.** Ideally in this line of research we ultimately want a single-loop Langevin algorithm with provable last-iterate convergence, however both MFL-AG and MFL-ABR only fulfill halves of this role. Hence we have chosen to present & compare both possibilities even though their philosophies are different: MFL-AG realizes fictitious play (optimizing based on historical behavior of the opponent) while MFL-ABR realizes best response (optimizing based on current behavior). The modes of convergence (average-iterate $O(1/t)$ convergence v.s. last-iterate exponential convergence in the outer loop) also provides a contrast between the two Langevin algorithms.
>
> **W2.** Thank you for your suggestions. The description of Equation 1 and other potentially confusing notational issues have been fixed.
>
> **Addressing Questions**
>
> **Q1.** Symmetry means that both $\mu,\nu$ have same learning rates (flow speeds). Until our paper, existing minimax MFLD results only studied imbalanced flows, that is $\partial_t\mu_t = (\text{equation for }\mu_t)$ and $\partial_t\nu_t = \eta\times(\text{equation for }\nu_t)$ for very large $\eta$ [1] or the static limit $\eta\to\infty$ [2]. In such regimes $\nu$ can be considered already nearly optimized for any $\mu$ and the difficulty of controlling two coupled flows are mostly removed. However, needless to say the setting is quite artificial. Symmetric updates are much more difficult to analyze and do not converge in general for even simple problems such as minmax$(x^\top y)$ for $x,y\in\mathbb{R}^d$ [3].
>
> **Q2.** Our method easily extends to different regularization strengths $\lambda_\mu,\lambda_\nu$ with minimal modifications on the derivations and bounds (this has been added to the first page footnote). Intuitively, the regularizations take place in their respective spaces $\mathcal{X}$ and $\mathcal{Y}$ so there is no need for the strength parameters to be equal. We only took equal $\lambda$ for simplicity and to adhere to preceding works.
>
> **Q3.** MFL-AG indeed performed better than MFL-ABR in this particular experiment, however we found empirically that this generally depended on the choice of hyperparameters. For example, speeding up the inner loop of MFL-ABR by decreasing iterations and increasing $\eta$, or speeding up the outer loop by increasing $\beta$ could accelerate convergence of MFL-ABR, but also led to instability in some runs. In contrast, MFL-AG always exhibited a "slow but steady" $O(1/t)$ convergence.
>
> In addition, MFL-ABR and MFL-DA performed better for problems where $Q(x,y)$ factored into a sum (or product) of separate potentials for $x,y$. In this case the dynamics decouples into two single-optimization problems which are known to converge exponentially. In contrast, MFL-AG was preferable for potentials with complex interactions between $x,y$. For example for the sigmoid potential in Section 6, $\mu$ pulls $\nu$ toward itself while $\nu$ pushes away, leading to a more complicated interacting dynamics that is better solved by the more stable MFL-AG.
>
> [1] Lu. Two-scale gradient descent ascent dynamics finds mixed Nash equilibria of
> continuous games: A mean-field perspective. ICML 2023.
>
> [2] Ma, Ying. Provably convergent quasistatic dynamics for mean-field two-player zero-sum games. ICLR 2022.
>
> [3] Daskalakis, Panageas. Last-iterate convergence: zero-sum games and constrained min-max optimization. Innovations in Theoretical Computer Science, 2019.

---

> > ### Comment · Reviewer_ByeN · 2023-11-22
> >
> > Thank you for all your detailed clarifications, which make this manuscript much clearer. I do not have any further questions, and I will keep my original score.

---

### Official Review · Reviewer_DQbp · 2023-10-31

**Soundness:** 2 fair
**Presentation:** 2 fair
**Contribution:** 4 excellent
**Rating:** 8
**Confidence:** 3

**Summary:**

The paper considers the solution of minmax problems with convex-concave objectives (plus entropic regularization terms) in the space of pairs of probability measures. Numerous problems can be viewed as particular cases of this.

There exists a natural Mckean-Vlasov type system of diffusions --  mean-field Langevin dynamics (MFLD) -- that should converge to the saddle point that solves the above problem. The paper shows such a result for a version of these diffusions where the drift term is averaged over past states. Moreover, particle approximations of the diffusions are shown to satisfy strong convergence rates: for instance, the approximation bounds are uniform in time.

Bounds for the authors' methods are given for "Markov games" assuming oracle access to certain functions and gradients related to the problem. A small simulation study suggests improved performance vis a vis MFLD.

**Strengths:**

The class of minmax optimization problems under consideration is quite natural. The result seems to be the first of its kind for this sort of problems. Similar results on convergence of McKean-Vlasov particle systems for convex optimization would be much easier to prove. The technical difficulties faced in the minimax scenario with the time averaging seem considerable.

**Weaknesses:**

The assumptions over the two measures used in entropic regularization are quite strong. The result of Theorem 3.7 has an expectation inside $W_1$, which is somewhat hard to interpret: it would seem to be saying that the bias of the procedure is small, but not the variance, so many repetitions of the particle dynamics would be needed to estimate the expectation of some function $f$. The outer-loop particle discretization error is not accounted for in the algorithm from Section 4.

I do not understand what is going on in Lemma C.7 (it might be because I do not understand $\Pi$ fully -- see below).

Some of the notation is unnecessarily hard to parse. Let me offer a few examples.

Page 5: *Denote ordered sets of $N$ particles* -- it would seem that the authors mean a vector in $\mathcal{X}^N$. Also on the same page: it seems that the authors introduce the subscript $1:k$ to identify particle positions at different times, but then don't use this notation consistently at the bottom of the page (and also elsewhere).

$L_\mu$ and $L_\nu$ are used to denote Lipschitz constants in the first/second variable, but also, $L$ is the functional and $(\mu,\nu)$ are used to denoted arbitrary elements of the space of probability measures.

The definition of $\Pi$ in page 6 as ``the average of the pushforward operators along the projections $X\mapsto X^i$ does not seem correct. How can you project to a higher-dimensional space?

**Questions:**

*(Q1)* Algorithm 1 and elsewhere - rathen than sample subsets of $\mathcal{X}_k,\mathcal{Y}_k$, maybe you could just use a weighted empirical measure where set $\mathcal{X}_k$ has weight proportional to $\beta_k$?

*(Q2)* Can you clarify what is $\Pi$ and explain specifically the proof of Lemma C7?

---

> ### Author Response · Authors · 2023-11-17
>
> Thank you for your positive feedback and detailed suggestions! We have updated the paper and clarified some important points accordingly.
>
> **Addressing Weaknesses**
>
> **Assumptions.** Our assumptions on the regularizer functions and functional $\mathcal{L}$ are standard in the mean-field literature, corresponding to L2 regularization in the case of two-layer neural networks [1,2]. In fact, they are more relaxed compared to some works which require super-quadratic growth of regularizers [3] or uniform boundedness of functional derivatives [4]. Lu and Domingo-Enrich's works do not have an L2 regularizer only due to the fact that they impose the strong assumptions that $\mathcal{X,Y}$ are *compact manifolds with Ricci curvature lower bounded*, which also guarantees an LSI constant.
>
> **Outer loop.** Please see our discussion with Reviewer q6kk on the outer loop discretization issue.
>
> **Expectation.** The expectation inside $W_1$ is purely due to the following fundamental fact: the expected Wasserstein distance between any $d$-dimensional distribution and a size $N$ sample is $O(N^{-1/d})$, so any attempt to directly control distance of $\widehat{\mu}$ to $\mu^*$ will immediately incur the curse of dimensionality even if $\widehat{\mu}$ is perfectly i.i.d. sampled from the true distribution. We thus carefully designed our approach to completely avoid this dependency. Nonetheless, if the expectation is outside we can still obtain a similar bound save for the CoD. We have added a new section (Appendix C.6) which discusses the effect of CoD and develops corresponding results and proofs.
>
> Of course, as you mentioned in practice we may also simply run the algorithm multiple ($M$) times and take the average of the outputs, which would also bypass the issue and yield the standard $1/\sqrt{M}$ convergence.
>
> **Notation.** The notation has been updated. The notation $(\mathscr{X},\mathscr{Y})\_{1:k}$ has been changed to $(\mathscr{X}\_{1:k},\mathscr{Y}\_{1:k})$ in the main text and reintroduced in the appendix. The functional has been changed from $L$ to $\mathcal{L}$, and the use of $\mu,\nu$ in sub/superscripts to denote problem quantities has been clarified on the first page.
>
> **Addressing Questions**
>
> **Q1.** It is indeed possible to weighted-concatenate the particles instead of random sampling in both Algorithm 1 and the outer loop of Algorithm 2. Since our discretization guarantee for MFL-AG is written in terms of expected distribution, it applies to both methods, and concatenation may help stabilize the output.
>
> The main issue is that the number of total particles $NK$ will then grow proportionally with the number of epochs $K$, so that with a per-epoch memory/compute constraint $C$ the user may have to set a smaller $N=O(C/K)$ to begin with. Whereas with random sampling the two algorithms only require 4 arrays of length $N=O(C)$ (as discussed in Section 5.1), so it is hard to say which is superior.
>
> If particle weighting is allowed, another natural consideration is to update the masses themselves and study the resulting discrete Wasserstein-Fisher-Rao gradient dynamics, which is an interesting direction for future work.
>
> **Q2.** We apologize for our mistake - the projection is supposed to be $\mathscr{X}\mapsto X^i$, that is the projection of the $N$-particle vector to its $i$th coordinate (averaged over all $i$). The corresponding pullback operator on functions is $\Pi^* f(\mathscr{X})=\frac{1}{N}\sum_{i=1}^N f(X^i)$. An explicit definition is given at the beginning of Appendix C.2. We hope this clarifies the role of Lemma C.7 as well.
>
> [1] Chen et al. Uniform-in-time propagation of chaos for mean
> field Langevin dynamics. 2022.
>
> [2] Mei et al. Mean-field theory of two-layers neural networks: dimension-free bounds and kernel limit. CoLT 2019.
>
> [3] Suzuki et al. Uniform-in-time propagation of chaos for the mean-field gradient Langevin dynamics. ICLR 2023.
>
> [4] Lascu et al. Entropic mean-field min-max problems via Best Response and Fisher-Rao flows. 2023.

---

### Official Review · Reviewer_bLee · 2023-11-01

**Soundness:** 3 good
**Presentation:** 3 good
**Contribution:** 3 good
**Rating:** 6
**Confidence:** 3

**Summary:**

The paper analyzes two approaches for simulating mean-field games, denoted by MFL-AG and MFL-ABR respectively.  The convergence of the algorithms (in idealized form, and finite particle form) is provided.

**Strengths:**

This appears to be the first algorithm which achieves a convergence in time + propagation of chaos result for mean-field games under transparent assumptions.

The proofs are well-written and contain analytical insight into this problem. In particular there is clear intuition on how averaging helps to regularize this problem.

Both algorithms are realizable and seem to have reasonable performance in practice.

**Weaknesses:**

I have some issues with the Assumptions in the paper.
	Firstly, the requirement of convexity combined with strictly bounded gradients is quite strict, as it essentially forces the problem to be roughly linear. I think this is a major restriction and the authors should do their best to circumvent this, or otherwise explain why this assumption was necessary in their opinion.

Secondly, it is unclear to me why the authors prefer to use $L^1$ Lipschitz-ness and obtain their result in Wasserstein-1. Isn’t it more natural to frame all results in W2, or was W1 chosen for a specific reason?

The dependencies on $N$ are worse when compared to standard mean-field results. The dependence on $\eta$ in the first term is also not ideal. For instance uniform-in-time propagation of chaos for the mean-field Langevin dynamics would look like $k^2/N^2$ (in $W_2^2$ under mild assumptions).

The results in Figure 1 are decent, but it is difficult to have any scale for comparison. It seems as well from the $W_1$ convergence that there is some degree of persistent bias for the choice of parameters. I would recommend further experiments if the authors are interested in highlighting the empirical performance of their approach.

In summary, I think this is a useful first result in this setting. However, there are numerous ways that it could be improved and I have some important questions regarding the results. I would be willing to raise my score if some of my questions could be addressed in more detail.

**Questions:**

Is it possible to write in this setting an analogue of the BBGKY hierarchy, and conduct the analysis that way? Would this at all sharpen the rates? It seems that under the convex-concavity assumption and bounded gradients assumptions at least, the analysis would be tractable using that approach.

Could Figure 1 be moved to the main text? It is a core part of the contribution and should not be placed in the appendix.

Could the dependency of $K$ on key problem parameters be clarified in the main text? It seems from the Appendices that it should be roughly $\epsilon^2$. Additionally, it would be good to highlight dimension dependence, which appears to be fairly mild.

Why in Theorem 4.1 should $k$ be taken as $1/\epsilon \log \epsilon$? Shouldn’t $\log 1/\epsilon$ suffice?

In Lemma C.2, no need to refer to Gronwall’s lemma (induction suffices).

A definition of “convexity” should be given in the Appendix. E.g. geodesic convexity in $W_2$, or convexity wrt TV distance.

---

> ### Author Response · Authors · 2023-11-17
>
> Thank you for your through review and suggestions, and raising a number of important issues! We have updated the manuscript to clarify technical details and improve presentation. We hope our discussion can convince the reviewer of the technical difficulty and generality of our work.
>
> **Addressing Weaknesses**
>
> **W1.** The functionals studied in even ordinary MFLD works (such as that for optimizing neural networks) are always convex; this is due to convexity of the loss function and an intended consequence of lifting to the space of measures, not due to simplicity of the underlying problem. Furthermore, minimax MFLD is often studied explicitly for bi-linear objectives, that is $\mathcal{L}(\mu,\nu)=\iint Q(x,y) d\mu(x)d\nu(y)$ where the underlying payoff $Q$ can be any nonconvex-nonconcave $C^1$ function. This covers all (2-player) zero-sum continuous games, which in turn generalizes all finite zero-sum games.
>
> In fact, **all** preceding minimax MFLD works [1,2,3] only considered bi-linear $\mathcal{L}$ (linear-convex in [4]), and even in this simple case it was unknown how to design a symmetric descent-ascent type algorithm until our paper. Hence, our work can actually be considered much more general.
>
> Furthermore, our assumptions in the gradient are implied by the assumptions on $Q$ in these works and are also generally considered weaker than sup norm control, which is also a common assumption in the MFLD literature. In fact, they are more relaxed compared to some works which require super-quadratic regularization or uniform boundedness of functional derivatives. (The works [1,2,3] avoid L2 regularization only because they impose the strong assumption that $\mathcal{X,Y}$ are compact manifolds without boundary, which automatically ensures various isoperimetric inequalities.)
>
> We agree that nonconvex MFLD is also of great interest, however extremely little is known for even the single-optimization case as of yet and should be considered as a separate direction of study.
>
> **W2.** Indeed, framing the result in $W_2$ as in [5] could be more natural. However unlike [5], for our dynamics we only have indirect control over the empirical-proximal distribution gap via Prop. 3.6, which yields a $W_1$ bound by Kantorovich duality when converting to a bound for the empirical measures. Our uniform LLN analysis (as opposed to entropy-based bounds) is a novel approach that allows us to overcome the challenges inherent in the extremely complex historically-dependent particle interactions. Please note that we have added a new section on bounding expected distance and curse of dimensionality (Appendix C.6); in this case, a stronger moment bound will allow strengthening to $W_2$.
>
> **W3.** The worse rates arise from controlling the full history dependence, not minimax dynamics a priori. For the uniform LLN (Proposition 3.6) at least, the $O(1/\sqrt{N})$ rate is optimal. The worst-case LLN error is determined by the change in Lipschitz constant of the function $F(\mu_{\mathscr{X}}, \nu_{\mathscr{X}})$ when each particle is removed, which in turn is tightly bounded by the leave-one-out $W_1$ error due to W1-Lipschitzity of $F(\cdot,\cdot)$. Since the leave-one-out $W_2^2$ error is $O(1/N)$, the LLN error must be $O(1/\sqrt{N})$.
>
> Since we did not quite use the full power of the uniform LLN in subsequent analyses, however, it is possible that a different approach could circumvent this and achieve better rates as well as getting rid of the dependency on $\eta$. Nevertheless, we emphasize this is the first history-dependent propagation of chaos result with explicit rates in any setting. The few existing results in the SPDE literature are asymptotic and do not provide error bounds [6].
>
> **W4.** The persistent bias in convergence is an artifact stemming from Langevin noise of fixed temperature, as particles are perturbed and empirical $W_1$ distance between each step retains some ground energy even at equilibrium. This decreases in MFL-AG due to replacing only a fraction of the particles. From a mean-field perspective, however, the distributions have converged to a solution.
>
> While our paper is focused on the regularized problem, it is also possible to implement a temperature annealing schedule such as $\lambda_t=\Theta(1/\log t)$ to find the MNE of the unregularized problem; for details, please see our response to Q3 of Reviewer q6kk. In this setting, the Langevin noise bias will gradually die down.
>
> Please also see our responses to Q2 of Reviewer q6kk for justification of the 3-point error as a metric for optimality, and Q3 of Reviewer ByeN for further discussion of experimental settings and comparisons of the algorithms.
>
> (continued below)

---

> ### Author Response · Authors · 2023-11-17
>
> (continued)
>
> **Addressing Questions**
>
> **Kinetic dynamics.** While not quite the same as BBGKY, a kinetic mean-field system has been considered in the context of machine learning optimization very recently in [7]. The studied underdamped MFLD corresponds to a momentum gradient descent, and exponential convergence and uniform-in-time propagation of chaos results are obtained. Although from an algorithmic perspective the use of momentum may potentially help to speed up the slow $O(1/t)$ rate of MFL-AG, it is unclear if the convergence rates (derived using entropic hypocoercivity) improves upon ordinary MFLD in general, and the propagation of chaos result requires some higher-order regularity assumptions. Adapting these techniques to deal with history dependence is also highly nontrivial (as was for ordinary MFLD for our paper). Nonetheless, applying kinetic MFLD to develop faster minimax algorithms is a very interesting and promising direction for future work.
>
> **Other edits.** We are grateful for your various suggestions which helped to improve important points of the manuscript.
>
> The dependency of $C_1$ on $\eta$ mentioned after Theorem 3.7 means that in order to bound all 3 terms as $\epsilon$, we need to take $\eta=O(\epsilon^2)$ which in turn roughly requires $K=O(\epsilon^{-1-1/\alpha})$ to assure $C_1/K<\epsilon$. This has been clarified in the main text.
>
> In Theorem 4.1, bounding discretization error (3rd term) requires $C\beta<\epsilon$ so that $k=\frac{1}{\epsilon}\log\frac{1}{\epsilon}$ is needed to ensure the outer loop convergence $\exp(-\beta k)<\epsilon$.
>
> The Figure has been moved to the main text.
>
> Dimension dependence only appears in the LSI constant as $\alpha = O(d)$ due to Lipschitz perturbations, and can be averted entirely if we assume bounded perturbations by the Holley-Stroock argument. We have added this in the paper. Also, the definition of convexity has been added and Lemma C.2 has been reworded as per your suggestions.
>
> [1] Domingo-Enrich et al. A mean-field analysis of two-player zero-sum games. NeurIPS 2020.
>
> [2] Ma, Ying. Provably convergent quasistatic dynamics for mean-field two-player zero-sum games. ICLR 2022.
>
> [3] Lu. Two-scale gradient descent ascent dynamics finds mixed Nash equilibria of continuous games: A mean-field perspective. ICML 2023.
>
> [4] Conforti et al. Game on random environment, mean-field Langevin system and neural networks. 2020.
>
> [5] Suzuki et al. Mean-field Langevin dynamics: Time-space discretization, stochastic gradient, and variance reduction. NeurIPS 2023.
>
> [6] Wu et al. On a class of McKean-Vlasov stochastic functional differential equations with applications. 2023.
>
> [7] Chen et al. Uniform-in-time propagation of chaos for kinetic
> mean field Langevin dynamics. 2023.

---

> > ### Comment · Reviewer_bLee · 2023-11-20
> > **Re: Response**
> >
> > I have read the authors response and agree with most of their points. The one issue for me in **W1** is to justify why strictly bounded gradients can be an interesting assumption, when compared with just Lipschitz gradients (which is more standard and covers the quadratic case). The authors addressed this somewhat but I would be keen to hear why this is a limiting assumption in practice.
> >
> > Nonetheless, I feel like most of my other points have been addressed and I have raised my score correspondingly. I thank the authors for their prompt and thorough response.

---

> ### Author Response · Authors · 2023-11-20
>
> Thank you for taking the time to consider our responses and raising your score! Your insightful comments and questions have helped greatly in improving our paper. Below, we discuss the gradient assumption in further detail. Recall (dropping the dependency on $\nu$ for simplicity):
>
> **Assumption 2.** $\nabla_x\frac{\delta\mathcal{L}}{\delta\mu}(\mu)(x)$ is uniformly bounded, and Lipschitz in $x,\mu$.
>
> We clarify that this is *not* a bound for the functional derivative $\frac{\delta\mathcal{L}}{\delta\mu}$ itself, only a regularity condition, and $\mathcal{L}$ could well be quadratic or higher order in $\mu$. For example, supposing $\mathcal{L}(\mu) = \iint f d\mu^{\otimes 2} = \iint f(x,z)\mu(dx)\mu(dz)$ for some function $f$, we have that $\nabla_x\frac{\delta\mathcal{L}}{\delta\mu}(\mu)(x) = \int \nabla_x f(x,z)\mu(dz)+\int \nabla_x f(z,x)\mu(dz)$, which is strictly bounded if $\nabla f$ is bounded. Moreover, Lipschitzity in $x$ and $W_1$-metric Lipschitzity in $\mu$ hold if $\nabla f$ is Lipschitz. Hence the assumption is conceptually a restriction on the *underlying potential* $f$, which is often well-behaved in practice.
>
> This is even more apparent in the two-layer neural network setting, which is nowadays the central application of MFLD. Denoting a neuron with parameter $\theta\in\Theta$ as $z\mapsto h_\theta(z)$, the mean-field network is $z\mapsto\int h_\theta(z)\mu(d\theta)$ for a distribution $\mu$ on the parameter space $\Theta$. Given a convex loss function $\ell(z,y)$, the training objective is then $\mathcal{L}(\mu)=\mathbb{E}[\ell(\int h_\theta(z)\mu(d\theta), y)]$. In this setting, Assumption 2 is verified if the derivative $\partial_z\ell$ of the loss is Lipschitz - this includes quadratic loss, logistic loss, etc. - and the neurons and their gradients are bounded, e.g. tanh activation. Moreover, for other applications such as MMD and KSD estimation, Assumption 2 is verified if the kernel is smooth and light-tailed such as the RBF kernel (see [5] above).
>
> Of course, bounded & smooth neurons are not ideal as ReLU does not satisfy both conditions. Clipping the output and smoothing near the origin are some ways to overcome this technicality, albeit with suboptimal results. The main point is that the *loss function* $\ell$ can be very general. In addition, for zero-sum games the underlying potential is the *payoff function* $Q$ which is likely to be suitably regular in applications.
>
> We reiterate that all current MFLD works require either this or a stronger assumption, and our technical settings mostly inherit from these single-optimization works. We put in much effort to develop the history-dependent propagation of chaos analysis with only Assumption 2, and indeed the proof of Theorem 3.7 is the most technically heavy result in our paper which could have been more simplified and optimized with stronger assumptions. However, our Assumption 3 for the MFL-ABR algorithm *does* unfortunately require boundedness of $\frac{\delta\mathcal{L}}{\delta\mu}$ (which is implied by a bounded payoff in the bilinear case). This is inherited from the original mean-field best-response paper; we could not find a way to weaken the assumption while also performing the additional perturbation analyses. Nevertheless, improving on these types of assumptions is definitely an important challenge for mean-field theory in general.

---

### Meta-Review · Area_Chair_GgNy · 2023-12-04

**Metareview:**

This paper introduces novel algorithms utilizing symmetric mean-field Langevin dynamics (MFLD) to address distributional minimax problems with convex-concave objectives. The authors offer a rigorous proof of convergence for two discrete MFLD algorithms, distinguished by single-loop and double-loop structures. The paper includes experimental validations affirming the proposed algorithms' convergence. The substantial contributions of this work are evident in both algorithmic design and theoretical analysis. All reviewers unanimously recommend acceptance. Congratulations on the excellent work!

**Justification For Why Not Higher Score:**

N/A

**Justification For Why Not Lower Score:**

This paper is well written, with new algorithms proposed and strong theoretical convergence analyses provided.

---

### Decision · Program_Chairs · 2024-01-16

Accept (spotlight)